# Interrogating theoretical models of neural computation with emergent property inference

**Sean R Bittner[1], Agostina Palmigiano[1], Alex T Piet[2,3,4], Chunyu A Duan[5], Carlos D Brody[2,3,6], Kenneth D Miller[1], John Cunningham[7]***

[1]Department of Neuroscience, Columbia University, New York, United States; [2]Princeton Neuroscience Institute, Princeton, United States; [3]Princeton University, Princeton, United States; [4]Allen Institute for Brain Science, Seattle, United States; [5]Institute of Neuroscience, Chinese Academy of Sciences, Shanghai, China; [6]Howard Hughes Medical Institute, Chevy Chase, United States; [7]Department of Statistics, Columbia University, New York, United States

**Abstract** A cornerstone of theoretical neuroscience is the circuit model: a system of equations that captures a hypothesized neural mechanism. Such models are valuable when they give rise to an experimentally observed phenomenon – whether behavioral or a pattern of neural activity – and thus can offer insights into neural computation. The operation of these circuits, like all models, critically depends on the choice of model parameters. A key step is then to identify the model parameters consistent with observed phenomena: to solve the inverse problem. In this work, we present a novel technique, emergent property inference (EPI), that brings the modern probabilistic modeling toolkit to theoretical neuroscience. When theorizing circuit models, theoreticians predominantly focus on reproducing computational properties rather than a particular dataset. Our method uses deep neural networks to learn parameter distributions with these computational properties. This methodology is introduced through a motivational example of parameter inference in the stomatogastric ganglion. EPI is then shown to allow precise control over the behavior of inferred parameters and to scale in parameter dimension better than alternative techniques. In the remainder of this work, we present novel theoretical findings in models of primary visual cortex and superior colliculus, which were gained through the examination of complex parametric structure captured by EPI. Beyond its scientific contribution, this work illustrates the variety of analyses possible once deep learning is harnessed towards solving theoretical inverse problems.

**\*For correspondence:**
jpc2181@columbia.edu

**Competing interests:** The authors declare that no competing interests exist.

## Introduction

The fundamental practice of theoretical neuroscience is to use a mathematical model to understand neural computation, whether that computation enables perception, action, or some intermediate processing. A neural circuit is systematized with a set of equations – the model – and these equations are motivated by biophysics, neurophysiology, and other conceptual considerations (*Kopell and Ermentrout, 1988*; *Marder, 1998*; *Abbott, 2008*; *Wang, 2010*; *O'Leary et al., 2015*). The function of this system is governed by the choice of model *parameters*, which when configured in a particular way, give rise to a measurable signature of a computation. The work of analyzing a model then requires solving the inverse problem: given a computation of interest, how can we reason about the distribution of parameters that give rise to it? The inverse problem is crucial for reasoning about likely parameter values, uniquenesses and degeneracies, and predictions made by the model (*Gutenkunst et al., 2007*; *Erguler and Stumpf, 2011*; *Mannakee et al., 2016*).

Ideally, one carefully designs a model and analytically derives how computational properties determine model parameters. Seminal examples of this gold standard include our field's understanding of memory capacity in associative neural networks (*Hopfield, 1982*), chaos and autocorrelation timescales in random neural networks (*Sompolinsky et al., 1988*), central pattern generation (*Olypher and Calabrese, 2007*), the paradoxical effect (*Tsodyks et al., 1997*), and decision making (*Wong and Wang, 2006*). Unfortunately, as circuit models include more biological realism, theory via analytical derivation becomes intractable. Absent this analysis, statistical inference offers a toolkit by which to solve the inverse problem by identifying, at least approximately, the distribution of parameters that produce computations in a biologically realistic model (*Foster et al., 1993*; *Prinz et al., 2004*; *Achard and De Schutter, 2006*; *Fisher et al., 2013*; *O'Leary et al., 2014*; *Alonso and Marder, 2019*).

Statistical inference, of course, requires quantification of the sometimes vague term *computation*. In neuroscience, two perspectives are dominant. First, often we directly use an *exemplar dataset*: a collection of samples that express the computation of interest, this data being gathered either experimentally in the lab or from a computer simulation. Although a natural choice given its connection to experiment (*Paninski and Cunningham, 2018*), some drawbacks exist: these data are well known to have features irrelevant to the computation of interest (*Niell and Stryker, 2010*; *Saleem et al., 2013*; *Musall et al., 2019*), confounding inferences made on such data. Related to this point, use of a conventional dataset encourages conventional data likelihoods or loss functions, which focus on some global metric like squared error or marginal evidence, rather than the computation itself.

Alternatively, researchers often quantify an *emergent property* (EP): a statistic of data that directly quantifies the computation of interest, wherein the dataset is implicit. While such a choice may seem esoteric, it is not: the above 'gold standard' examples (*Hopfield, 1982*; *Sompolinsky et al., 1988*; *Olypher and Calabrese, 2007*; *Tsodyks et al., 1997*; *Wong and Wang, 2006*) all quantify and focus on some derived feature of the data, rather than the data drawn from the model. An emergent property is of course a dataset by another name, but it suggests different approach to solving the same inverse problem: here, we directly specify the desired emergent property – a statistic of data drawn from the model – and the value we wish that property to have, and we set up an optimization program to find the distribution of parameters that produce this computation. This statistical framework is not new: it is intimately connected to the literature on approximate bayesian computation (*Beaumont et al., 2002*; *Marjoram et al., 2003*; *Sisson et al., 2007*), parameter sensitivity analyses (*Raue et al., 2009*; *Karlsson et al., 2012*; *Hines et al., 2014*; *Raman et al., 2017*), maximum entropy modeling (*Elsayed and Cunningham, 2017*; *Savin and Tkačik, 2017*; *Młynarski et al., 2020*), and approximate bayesian inference (*Tran et al., 2017*; *Gonçalves et al., 2019*); we detail these connections in Section 'Related approaches'.

The parameter distributions producing a computation may be curved or multimodal along various parameter axes and combinations. It is by quantifying this complex structure that emergent property inference offers scientific insight. Traditional approximation families (e.g. mean-field or mixture of gaussians) are limited in the distributional structure they may learn. To address such restrictions on expressivity, advances in machine learning have used deep probability distributions as flexible approximating families for such complicated distributions (*Rezende and Mohamed, 2015*; *Papamakarios et al., 2019a*) (see Section 'Deep probability distributions and normalizing flows'). However, the adaptation of deep probability distributions to the problem of theoretical circuit analysis requires recent developments in deep learning for constrained optimization (*Loaiza-Ganem et al., 2017*), and architectural choices for efficient and expressive deep generative modeling (*Dinh et al., 2017*; *Kingma and Dhariwal, 2018*). We detail our method, which we call emergent property inference (EPI) in Section 'Emergent property inference via deep generative models'.

Equipped with this method, we demonstrate the capabilities of EPI and present novel theoretical findings from its analysis. First, we show EPI's ability to handle biologically realistic circuit models using a five-neuron model of the stomatogastric ganglion (*Gutierrez et al., 2013*): a neural circuit whose parametric degeneracy is closely studied (*Goldman et al., 2001*). Then, we show EPI's scalability to high dimensional parameter distributions by inferring connectivities of recurrent neural networks that exhibit stable, yet amplified responses – a hallmark of neural responses throughout the brain (*Murphy and Miller, 2009*; *Hennequin et al., 2014*; *Bondanelli et al., 2019*). In a model of primary visual cortex (*Litwin-Kumar et al., 2016*; *Palmigiano et al., 2020*), EPI reveals how the

recurrent processing across different neuron-type populations shapes excitatory variability: a finding that we show is analytically intractable. Finally, we investigated the possible connectivities of a superior colliculus model that allow execution of different tasks on interleaved trials (*Duan et al., 2021*). EPI discovered a rich distribution containing two connectivity regimes with different solution classes. We queried the deep probability distribution learned by EPI to produce a mechanistic understanding of neural responses in each regime. Intriguingly, the inferred connectivities of each regime reproduced results from optogenetic inactivation experiments in markedly different ways. These theoretical insights afforded by EPI illustrate the value of deep inference for the interrogation of neural circuit models.

## Results

### Motivating emergent property inference of theoretical models

Consideration of the typical workflow of theoretical modeling clarifies the need for emergent property inference. First, one designs or chooses an existing circuit model that, it is hypothesized, captures the computation of interest. To ground this process in a well-known example, consider the stomatogastric ganglion (STG) of crustaceans, a small neural circuit which generates multiple rhythmic muscle activation patterns for digestion (*Marder and Thirumalai, 2002*). Despite full knowledge of STG connectivity and a precise characterization of its rhythmic pattern generation, biophysical models of the STG have complicated relationships between circuit parameters and computation (*Goldman et al., 2001*; *Prinz et al., 2004*).

A subcircuit model of the STG (*Gutierrez et al., 2013*) is shown schematically in *Figure 1A*. The fast population (f1 and f2) represents the subnetwork generating the pyloric rhythm and the slow population (s1 and s2) represents the subnetwork of the gastric mill rhythm. The two fast neurons mutually inhibit one another, and spike at a greater frequency than the mutually inhibiting slow neurons. The hub neuron couples with either the fast or slow population, or both depending on modulatory conditions. The jagged connections indicate electrical coupling having electrical conductance $g_{el}$, smooth connections in the diagram are inhibitory synaptic projections having strength $g_{synA}$ onto the hub neuron, and $g_{synB} = 5$ nS for mutual inhibitory connections. Note that the behavior of this model will be critically dependent on its parameterization – the choices of conductance parameters $\mathbf{z} = [g_{el}, g_{synA}]$.

Second, once the model is selected, one must specify what the model should produce. In this STG model, we are concerned with neural spiking frequency, which emerges from the dynamics of the circuit model (*Figure 1B*). An emergent property studied by Gutierrez et al. is the hub neuron firing at an intermediate frequency between the intrinsic spiking rates of the fast and slow populations. This emergent property (EP) is shown in *Figure 1C* at an average frequency of 0.55 Hz. To be precise, we define intermediate hub frequency not strictly as 0.55 Hz, but frequencies of moderate deviation from 0.55 Hz between the fast (.35Hz) and slow (.68Hz) frequencies.

Third, the model parameters producing the emergent property are inferred. By precisely quantifying the emergent property of interest as a statistical feature of the model, we use emergent property inference (EPI) to condition directly on this emergent property. Before presenting technical details (in the following section), let us understand emergent property inference schematically. EPI (*Figure 1D*) takes, as input, the model and the specified emergent property, and as its output, returns the parameter distribution (*Figure 1E*). This distribution – represented for clarity as samples from the distribution – is a parameter distribution constrained such that the circuit model produces the emergent property. Once EPI is run, the returned distribution can be used to efficiently generate additional parameter samples. Most importantly, the inferred distribution can be efficiently queried to quantify the parametric structure that it captures. By quantifying the parametric structure governing the emergent property, EPI informs the central question of this inverse problem: what aspects or combinations of model parameters have the desired emergent property?

### Emergent property inference via deep generative models

EPI formalizes the three-step procedure of the previous section with deep probability distributions (*Rezende and Mohamed, 2015*; *Papamakarios et al., 2019a*). First, as is typical, we consider the model as a coupled set of noisy differential equations. In this STG example, the model activity (or

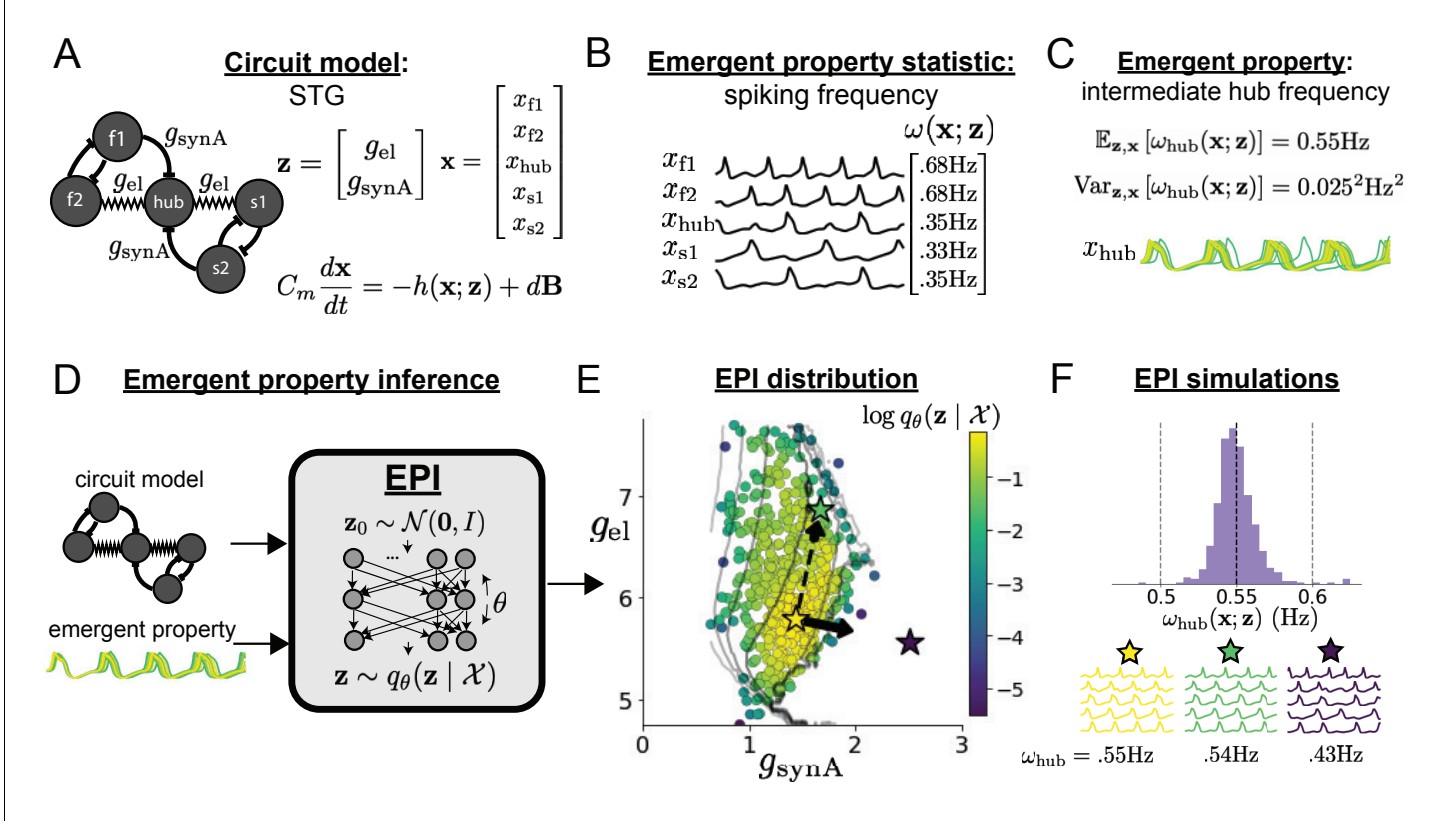

**Figure 1.** Emergent property inference in the stomatogastric ganglion. (A) Conductance-based subcircuit model of the STG. (B) Spiking frequency $\omega(\mathbf{x}; \mathbf{z})$ is an emergent property statistic. Simulated at $g_{el} = 4.5$ nS and $g_{synA} = 3$ nS. (C) The emergent property of intermediate hub frequency. Simulated activity traces are colored by log probability of generating parameters in the EPI distribution (Panel E). (D) For a choice of circuit model and emergent property, EPI learns a deep probability distribution of parameters $\mathbf{z}$. (E) The EPI distribution producing intermediate hub frequency. Samples are colored by log probability density. Contours of hub neuron frequency error are shown at levels of 0.525, 0.53, . . . 0.575 Hz (dark to light gray away from mean). Dimension of sensitivity $\mathbf{v_1}$ (solid arrow) and robustness $\mathbf{v_2}$ (dashed arrow). (F) (Top) The predictions of the EPI distribution. The black and gray dashed lines show the mean and two standard deviations according the emergent property. (Bottom) Simulations at the starred parameter values. The online version of this article includes the following figure supplement(s) for figure 1:

**Figure supplement 1.** Emergent property inference in a 2D linear dynamical system.

**Figure supplement 2.** Analytic contours of inferred EPI distribution.

**Figure supplement 3.** Sampled dynamical systems $\mathbf{z} \sim q_{\boldsymbol{\theta}}(\mathbf{z} \mid \mathcal{X})$ and their simulated activity from $\mathbf{x}(t = 0) = [\frac{\sqrt{2}}{2}, -\frac{\sqrt{2}}{2}]$ colored by log probability.

**Figure supplement 4.** EPI optimization of the STG model producing network syncing.

state) $\mathbf{x} = [x_{f1}, x_{f2}, x_{hub}, x_{s1}, x_{s2}]$ is the membrane potential for each neuron, which evolves according to the biophysical conductance-based equation:

$$C_m \frac{d\mathbf{x}(t)}{dt} = -h(\mathbf{x}(t); \mathbf{z}) + d\mathbf{B} \qquad (1)$$

where $C_m = 1$nF, and $\mathbf{h}$ is a sum of the leak, calcium, potassium, hyperpolarization, electrical, and synaptic currents, all of which have their own complicated dependence on activity $\mathbf{x}$ and parameters $\mathbf{z} = [g_{el}, g_{synA}]$, and $d\mathbf{B}$ is white gaussian noise (*Gutierrez et al., 2013*; see Section 'STG model' for more detail).

Second, we determine that our model should produce the emergent property of 'intermediate hub frequency' (*Figure 1C*). We stipulate that the hub neuron's spiking frequency – denoted by statistic $\omega_{hub}(\mathbf{x})$ – is close to a frequency of 0.55 Hz, between that of the slow and fast frequencies. Mathematically, we define this emergent property with two constraints: that the mean hub frequency is 0.55 Hz,

$$\mathbb{E}_{\mathbf{z},\mathbf{x}}[\omega_{\mathrm{hub}}(\mathbf{x};\mathbf{z})] = 0.55 \qquad (2)$$

and that the variance of the hub frequency is moderate

$$\mathrm{Var}_{\mathbf{z},\mathbf{x}}[\omega_{\mathrm{hub}}(\mathbf{x};\mathbf{z})] = 0.025^2. \qquad (3)$$

In the emergent property of intermediate hub frequency, the statistic of hub neuron frequency is an expectation over the distribution of parameters $\mathbf{z}$ and the distribution of the data $\mathbf{x}$ that those parameters produce. We define the emergent property $\mathcal{X}$ as the collection of these two constraints. In general, an emergent property is a collection of constraints on statistical moments that together define the computation of interest.

Third, we perform emergent property inference: we find a distribution over parameter configurations $\mathbf{z}$ of models that produce the emergent property; in other words, they satisfy the constraints introduced in *Equations 2 and 3*. This distribution will be chosen from a family of probability distributions $\mathcal{Q} = \{q_{\boldsymbol{\theta}}(\mathbf{z}) : \boldsymbol{\theta} \in \Theta\}$, defined by a deep neural network (*Rezende and Mohamed, 2015*; *Papamakarios et al., 2019a*; *Figure 1D*, EPI box). Deep probability distributions map a simple random variable $\mathbf{z}_0$ (e.g. an isotropic gaussian) through a deep neural network with weights and biases $\boldsymbol{\theta}$ to parameters $\mathbf{z} = g_{\boldsymbol{\theta}}(\mathbf{z}_0)$ of a suitably complicated distribution (see Section 'Deep probability distributions and normalizing flows' for more details). Many distributions in $\mathcal{Q}$ will respect the emergent property constraints, so we select the most random (highest entropy) distribution, which also means this approach is equivalent to bayesian variational inference (see Section 'EPI as variational inference'). In EPI optimization, stochastic gradient steps in $\boldsymbol{\theta}$ are taken such that entropy is maximized, and the emergent property $\mathcal{X}$ is produced (see Section 'Emergent property inference (EPI)'). We then denote the inferred EPI distribution as $q_{\boldsymbol{\theta}}(\mathbf{z} \mid \mathcal{X})$, since the structure of the learned parameter distribution is determined by weights and biases $\boldsymbol{\theta}$, and this distribution is conditioned upon emergent property $\mathcal{X}$.

The structure of the inferred parameter distributions of EPI can be analyzed to reveal key information about how the circuit model produces the emergent property. As probability in the EPI distribution decreases away from the mode of $q_{\boldsymbol{\theta}}(\mathbf{z} \mid \mathcal{X})$ (*Figure 1E* yellow star), the emergent property deteriorates. Perturbing $\mathbf{z}$ along a dimension in which $q_{\boldsymbol{\theta}}(\mathbf{z} \mid \mathcal{X})$ changes little will not disturb the emergent property, making this parameter combination *robust* with respect to the emergent property. In contrast, if $\mathbf{z}$ is perturbed along a dimension with strongly decreasing $q_{\boldsymbol{\theta}}(\mathbf{z} \mid \mathcal{X})$, that parameter combination is deemed *sensitive* (*Raue et al., 2009*; *Raman et al., 2017*). By querying the second-order derivative (Hessian) of $\log q_{\boldsymbol{\theta}}(\mathbf{z} \mid \mathcal{X})$ at a mode, we can quantitatively identify how sensitive (or robust) each eigenvector is by its eigenvalue; the more negative, the more sensitive and the closer to zero, the more robust (see Section 'Hessian sensitivity vectors'). Indeed, samples equidistant from the mode along these dimensions of sensitivity ($\mathbf{v}_1$, smaller eigenvalue) and robustness ($\mathbf{v}_2$, greater eigenvalue) (*Figure 1E*, arrows) agree with error contours (*Figure 1E* contours) and have diminished or preserved hub frequency, respectively (*Figure 1F* activity traces). The directionality of $\mathbf{v}_2$ suggests that changes in conductance along this parameter combination will most preserve hub neuron firing between the intrinsic rates of the pyloric and gastric mill rhythms. Importantly and unlike alternative techniques, once an EPI distribution has been learned, the modes and Hessians of the distribution can be measured with trivial computation (see Section 'Deep probability distributions and normalizing flows').

In the following sections, we demonstrate EPI on three neural circuit models across ranges of biological realism, neural system function, and network scale. First, we demonstrate the superior scalability of EPI compared to alternative techniques by inferring high-dimensional distributions of recurrent neural network connectivities that exhibit amplified, yet stable responses. Next, in a model of primary visual cortex (*Litwin-Kumar et al., 2016*; *Palmigiano et al., 2020*), we show how EPI discovers parametric degeneracy, revealing how input variability across neuron types affects the excitatory population. Finally, in a model of superior colliculus (*Duan et al., 2021*), we used EPI to capture multiple parametric regimes of task switching, and queried the dimensions of parameter sensitivity to characterize each regime.

## Scaling inference of recurrent neural network connectivity with EPI

To understand how EPI scales in comparison to existing techniques, we consider recurrent neural networks (RNNs). Transient amplification is a hallmark of neural activity throughout cortex and is often thought to be intrinsically generated by recurrent connectivity in the responding cortical area (*Murphy and Miller, 2009*; *Hennequin et al., 2014*; *Bondanelli et al., 2019*). It has been shown that to generate such amplified, yet stabilized responses, the connectivity of RNNs must be non-normal (*Goldman, 2009*; *Murphy and Miller, 2009*), and satisfy additional constraints (*Bondanelli and Ostojic, 2020*). In theoretical neuroscience, RNNs are optimized and then examined to show how dynamical systems could execute a given computation (*Sussillo, 2014*; *Barak, 2017*), but such biologically realistic constraints on connectivity (*Goldman, 2009*; *Murphy and Miller, 2009*; *Bondanelli and Ostojic, 2020*) are ignored for simplicity or because constrained optimization is difficult. In general, access to distributions of connectivity that produce theoretical criteria like stable amplification, chaotic fluctuations (*Sompolinsky et al., 1988*), or low tangling (*Russo et al., 2018*) would add scientific value to existing research with RNNs. Here, we use EPI to learn RNN connectivities producing stable amplification, and demonstrate the superior scalability and efficiency of EPI to alternative approaches.

We consider a rank-2 RNN with $N$ neurons having connectivity $W = UV^\top$ and dynamics

$$\tau \dot{\mathbf{x}} = -\mathbf{x} + W\mathbf{x}, \tag{4}$$

where $U = [\mathbf{U}_1 \ \mathbf{U}_2] + g\chi^{(U)}$, $V = [\mathbf{V}_1 \ \mathbf{V}_2] + g\chi^{(V)}$, $\mathbf{U}_1\mathbf{U}_2, \mathbf{V}_1, \mathbf{V}_2 \in [-1,1]^N$, and $\chi_{i,j}^{(U)}, \chi_{i,j}^{(V)} \sim \mathcal{N}(0,1)$. We infer connectivity parameters $\mathbf{z} = [\mathbf{U}_1, \mathbf{U}_2, \mathbf{V}_1, \mathbf{V}_2]$ that produce stable amplification. Two conditions are necessary and sufficient for RNNs to exhibit stable amplification (*Bondanelli and Ostojic, 2020*): $\mathrm{real}(\lambda_1) < 1$ and $\lambda_1^s > 1$, where $\lambda_1$ is the eigenvalue of $W$ with greatest real part and $\lambda^s$ is the maximum eigenvalue of $W^s = \frac{W+W^\top}{2}$. RNNs with $\mathrm{real}(\lambda_1) = 0.5 \pm 0.5$ and $\lambda_1^s = 1.5 \pm 0.5$ will be stable with modest decay rate ($\mathrm{real}(\lambda_1)$ close to its upper bound of 1) and exhibit modest amplification ($\lambda_1^s$ close to its lower bound of 1). EPI can naturally condition on this emergent property

$$
\begin{aligned}
\mathcal{X} : \mathbb{E}_{\mathbf{z},\mathbf{x}} \begin{bmatrix} \mathrm{real}(\lambda_1) \\ \lambda_1^s \end{bmatrix} &= \begin{bmatrix} 0.5 \\ 1.5 \end{bmatrix} \\
\mathrm{Var}_{\mathbf{z},\mathbf{x}} \begin{bmatrix} \mathrm{real}(\lambda_1) \\ \lambda_1^s \end{bmatrix} &= \begin{bmatrix} 0.25^2 \\ 0.25^2 \end{bmatrix}.
\end{aligned}
\tag{5}
$$

Variance constraints predicate that the majority of the distribution (within two standard deviations) are within the specified ranges.

For comparison, we infer the parameters $\mathbf{z}$ likely to produce stable amplification using two alternative simulation-based inference approaches. Sequential Monte Carlo approximate bayesian computation (SMC-ABC) (*Sisson et al., 2007*) is a rejection sampling approach that uses SMC techniques to improve efficiency, and sequential neural posterior estimation (SNPE) (*Gonçalves et al., 2019*) approximates posteriors with deep probability distributions (see Section 'Related approaches'). Unlike EPI, these statistical inference techniques do not constrain the predictions of the inferred distribution, so they were run by conditioning on an exemplar dataset $\mathbf{x}_0 = \boldsymbol{\mu}$, following standard practice with these methods (*Sisson et al., 2007*; *Gonçalves et al., 2019*). To compare the efficiency of these different techniques, we measured the time and number of simulations necessary for the distance of the predictive mean to be less than 0.5 from $\boldsymbol{\mu} = \mathbf{x}_0$ (see Section 'Scaling EPI for stable amplification in RNNs').

As the number of neurons $N$ in the RNN, and thus the dimension of the parameter space $\mathbf{z} \in [-1,1]^{4N}$, is scaled, we see that EPI converges at greater speed and at greater dimension than SMC-ABC and SNPE (*Figure 2A*). It also becomes most efficient to use EPI in terms of simulation count at $N = 50$ (*Figure 2B*). It is well known that ABC techniques struggle in parameter spaces of modest dimension (*Sisson et al., 2018*), yet we were careful to assess the scalability of SNPE, which is a more closely related methodology to EPI. Between EPI and SNPE, we closely controlled the number of parameters in deep probability distributions by dimensionality (*Figure 2—figure supplement 1*), and tested more aggressive SNPE hyperparameter choices when SNPE failed to converge (*Figure 2—figure supplement 2*). In this analysis, we see that deep inference techniques EPI and SNPE are far more amenable to inference of high dimensional RNN connectivities than rejection

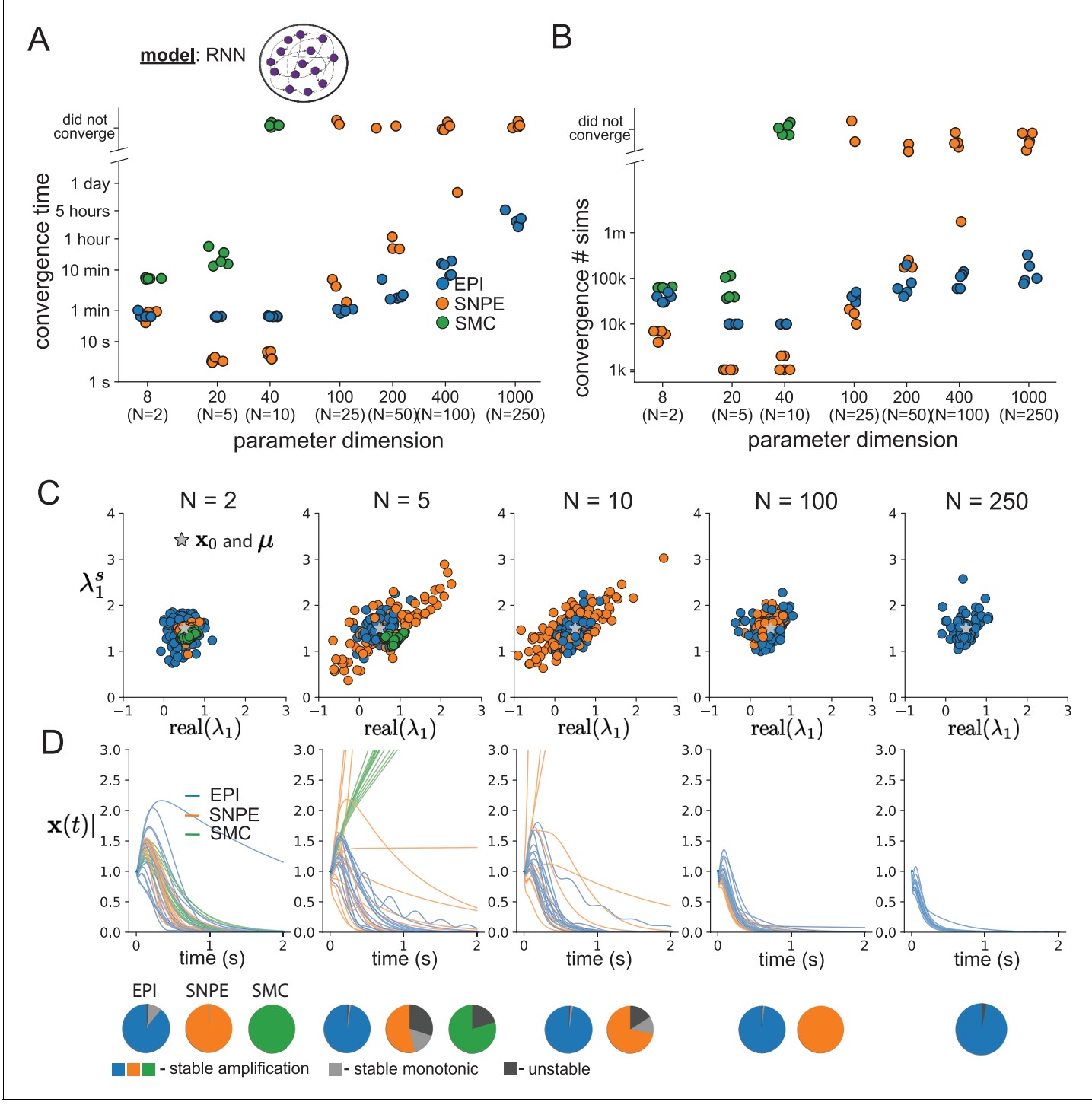

**Figure 2.** Inferring recurrent neural networks with stable amplification. (**A**) Wall time of EPI (blue), SNPE (orange), and SMC-ABC (green) to converge on RNN connectivities producing stable amplification. Each dot shows convergence time for an individual random seed. For reference, the mean wall time for EPI to achieve its full constraint convergence (means and variances) is shown (blue line). (**B**) Simulation count of each algorithm to achieve convergence. Same conventions as A. (**C**) The predictive distributions of connectivities inferred by EPI (blue), SNPE (orange), and SMC-ABC (green), with reference to $\mathbf{x}_0 = \boldsymbol{\mu}$ (gray star). (**D**) Simulations of networks inferred by each method ($\tau = 100ms$). Each trace (15 per algorithm) corresponds to simulation of one $z$. (Below) Ratio of obtained samples producing stable amplification, stable monotonic decay, and instability.

The online version of this article includes the following figure supplement(s) for figure 2:

**Figure supplement 1.** Architecture parameter comparison of EPI and SNPE.

**Figure supplement 2.** SNPE convergence was enabled by increasing $n_{\mathrm{round}}$, not $n_{\mathrm{atom}}$.

*Figure 2 continued on next page*

*Figure 2 continued*

**Figure supplement 3.** Model characteristics affect predictions of posteriors inferred by SNPE, while predictions of parameters inferred by EPI remain fixed.

sampling techniques like SMC-ABC, and that EPI outperforms SNPE in both wall time (elapsed real time) and simulation count.

No matter the number of neurons, EPI always produces connectivity distributions with mean and variance of $\mathrm{real}(\lambda_1)$ and $\lambda_1^s$ according to $\mathcal{X}$ (*Figure 2C*, blue). For the dimensionalities in which SMC-ABC is tractable, the inferred parameters are concentrated and offset from the exemplar dataset $\mathbf{x}_0$ (*Figure 2C*, green). When using SNPE, the predictions of the inferred parameters are highly concentrated at some RNN sizes and widely varied in others (*Figure 2C*, orange). We see these properties reflected in simulations from the inferred distributions: EPI produces a consistent variety of stable, amplified activity norms $|\mathbf{x}(t)|$, SMC-ABC produces a limited variety of responses, and the changing variety of responses from SNPE emphasizes the control of EPI on parameter predictions (*Figure 2D*). Even for moderate neuron counts, the predictions of the inferred distribution of SNPE are highly dependent on $N$ and $g$, while EPI maintains the emergent property across choices of RNN (see Section 'Effect of RNN parameters on EPI and SNPE inferred distributions').

To understand these differences, note that EPI outperforms SNPE in high dimensions by using gradient information (from $\nabla_{\mathbf{z}}[\mathrm{real}(\lambda_1), \lambda_1^s]^\top$). This choice agrees with recent speculation that such gradient information could improve the efficiency of simulation-based inference techniques (*Cranmer et al., 2020*), as well as reflecting the classic tradeoff between gradient-based and sampling-based estimators (scaling and speed versus generality). Since gradients of the emergent property are necessary in EPI optimization, gradient tractability is a key criteria when determining the suitability of a simulation-based inference technique. If the emergent property gradient is efficiently calculated, EPI is a clear choice for inferring high dimensional parameter distributions. In the next two sections, we use EPI for novel scientific insight by examining the structure of inferred distributions.

## EPI reveals how recurrence with multiple inhibitory subtypes governs excitatory variability in a V1 model

Dynamical models of excitatory (E) and inhibitory (I) populations with supralinear input-output function have succeeded in explaining a host of experimentally documented phenomena in primary visual cortex (V1). In a regime characterized by inhibitory stabilization of strong recurrent excitation, these models give rise to paradoxical responses (*Tsodyks et al., 1997*), selective amplification (*Goldman, 2009*; *Murphy and Miller, 2009*), surround suppression (*Ozeki et al., 2009*), and normalization (*Rubin et al., 2015*). Recent theoretical work (*Hennequin et al., 2018*) shows that stabilized E-I models reproduce the effect of variability suppression (*Churchland et al., 2010*). Furthermore, experimental evidence shows that inhibition is composed of distinct elements – parvalbumin (P), somatostatin (S), VIP (V) – composing 80% of GABAergic interneurons in V1 (*Markram et al., 2004*; *Rudy et al., 2011*; *Tremblay et al., 2016*), and that these inhibitory cell types follow specific connectivity patterns (*Figure 3A*; *Pfeffer et al., 2013*). Here, we use EPI on a model of V1 with biologically realistic connectivity to show how the structure of input across neuron types affects the variability of the excitatory population – the population largely responsible for projecting to other brain areas (*Felleman and Van Essen, 1991*).

We considered response variability of a nonlinear dynamical V1 circuit model (*Figure 3A*) with a state comprised of each neuron-type population's rate $\mathbf{x} = [x_E, x_P, x_S, x_V]^\top$. Each population receives recurrent input $W\mathbf{x}$, where $W$ is the effective connectivity matrix (see Section 'Primary visual cortex') and an external input with mean $\mathbf{h}$, which determines population rate via supralinear nonlinearity $\phi(\cdot) = [\cdot]_+^2$. The external input has an additive noisy component $\epsilon$ with variance $\sigma^2 = [\sigma_E^2, \sigma_P^2, \sigma_S^2, \sigma_V^2]$. This noise has a slower dynamical timescale $\tau_{\mathrm{noise}} > \tau$ than the population rate, allowing fluctuations around a stimulus-dependent steady-state (*Figure 3B*). This model is the stochastic stabilized supralinear network (SSSN) (*Hennequin et al., 2018*)

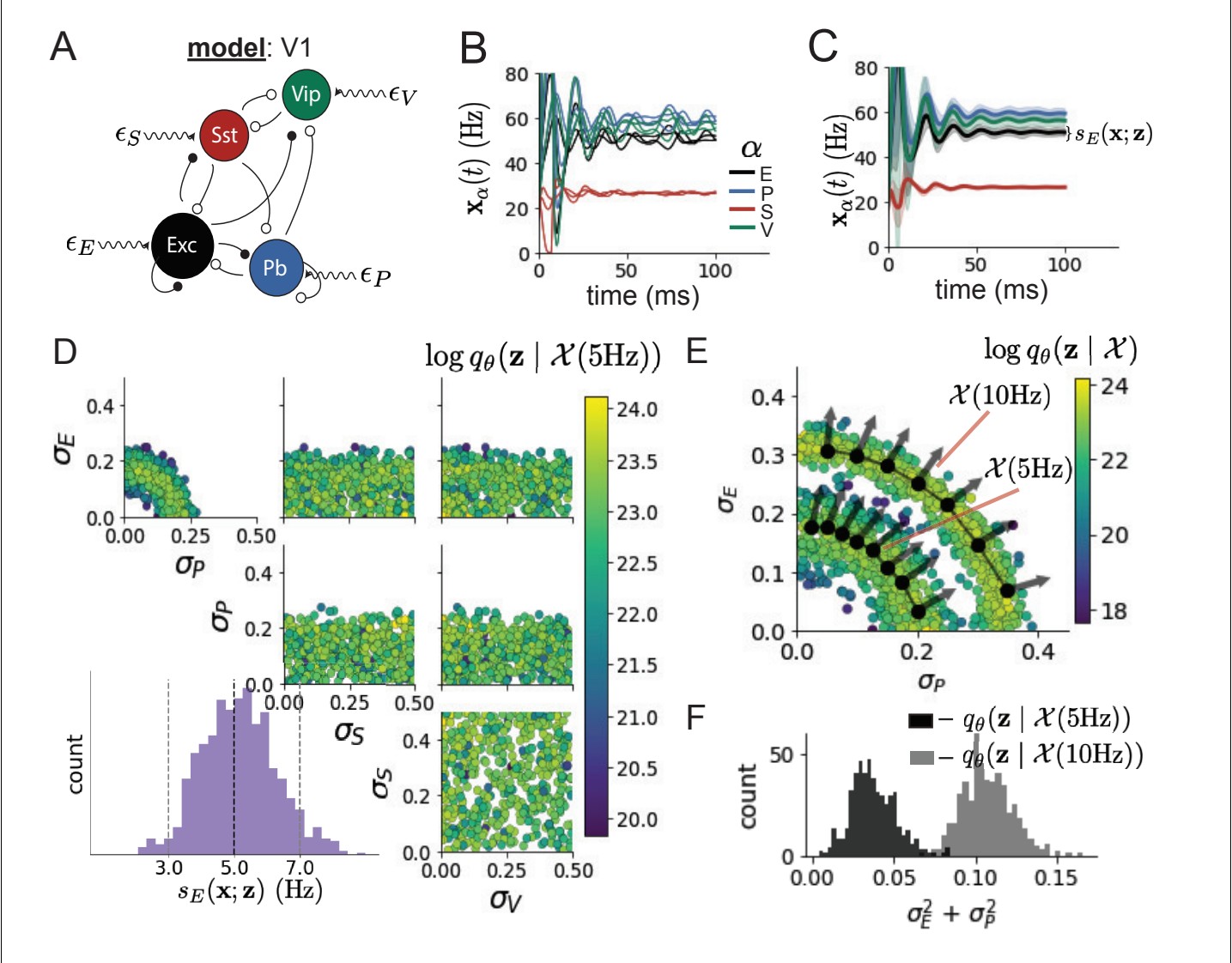

**Figure 3.** Emergent property inference in the stochastic stabilized supralinear network (SSSN). **(A)** Four-population model of primary visual cortex with excitatory (black), parvalbumin (blue), somatostatin (red), and VIP (green) neurons (excitatory and inhibitory projections filled and unfilled, respectively). Some neuron-types largely do not form synaptic projections to others ($|(W_{\alpha_1, \alpha_2})|<0.025$). Each neural population receives a baseline input $\mathbf{h}_b$, and the E- and P-populations also receive a contrast-dependent input $\mathbf{h}_c$. Additionally, each neural population receives a slow noisy input $\epsilon$. **(B)** Transient network responses of the SSSN model. Traces are independent trials with varying initialization $\mathbf{x}(0)$ and noise $\epsilon$. **(C)** Mean (solid line) and standard deviation $s_E(\mathbf{x}; \mathbf{z})$ (shading) across 100 trials. **(D)** EPI distribution of noise parameters $\mathbf{z}$ conditioned on E-population variability. The EPI predictive distribution of $s_E(\mathbf{x}; \mathbf{z})$ is show on the bottom-left. **(E)** (Top) Enlarged visualization of the $\sigma_E$-$\sigma_P$ marginal distribution of EPI $q_\theta(\mathbf{z} \mid \mathcal{X}(5\text{Hz}))$ and $q_\theta(\mathbf{z} \mid \mathcal{X}(10\text{Hz}))$. Each black dot shows the mode at each $\sigma_P$. The arrows show the most sensitive dimensions of the Hessian evaluated at these modes. **(F)** The predictive distributions of $\sigma_E^2 + \sigma_P^2$ of each inferred distribution $q_\theta(\mathbf{z} \mid \mathcal{X}(5\text{Hz}))$ and $q_\theta(\mathbf{z} \mid \mathcal{X}(10\text{Hz}))$.

The online version of this article includes the following figure supplement(s) for figure 3:

**Figure supplement 1.** EPI inferred distribution for $\mathcal{X}(10\text{Hz})$.

**Figure supplement 2.** EPI optimization.

**Figure supplement 3.** EPI predictive distributions of the sum of squares of each pair of noise parameters.

**Figure supplement 4.** SSSN simulations for small increases in neuron-type population input (left); average (solid) and standard deviation (shaded) of stochastic fluctuations of responses (right).

$$\tau \frac{d\mathbf{x}}{dt} = -\mathbf{x} + \phi(W\mathbf{x} + \mathbf{h} + \epsilon), \tag{6}$$

generalized to have multiple inhibitory neuron types. It introduces stochasticity to four neuron-type models of V1 (*Litwin-Kumar et al., 2016*). Stochasticity and inhibitory multiplicity introduce substantial complexity to the mathematical treatment of this problem (see Section 'Primary visual cortex: Mathematical intuition and challenges') motivating the analysis of this model with EPI. Here, we consider fixed weights $W$ and input $\mathbf{h}$ (*Palmigiano et al., 2020*), and study the effect of input variability $\mathbf{z} = [\sigma_E, \sigma_P, \sigma_S, \sigma_V]^\top$ on excitatory variability.

We quantify levels of E-population variability by studying two emergent properties

$$\begin{aligned}\mathcal{X}(5\text{Hz}):\quad &\mathbb{E}_{\mathbf{z},\mathbf{x}} s_E(\mathbf{x};\mathbf{z}) &&= 5\text{Hz} &\qquad \mathcal{X}(10\text{Hz}):\quad &\mathbb{E}_{\mathbf{z},\mathbf{x}} s_E(\mathbf{x};\mathbf{z}) &&= 10\text{Hz}\\ &\mathrm{Var}_{\mathbf{z},\mathbf{x}} s_E(\mathbf{x};\mathbf{z}) &&= 1\text{Hz}^2 & & \mathrm{Var}_{\mathbf{z},\mathbf{x}} s_E(\mathbf{x};\mathbf{z}) &&= 1\text{Hz}^2,\end{aligned} \tag{7}$$

where $s_E(\mathbf{x};\mathbf{z})$ is the standard deviation of the stochastic $E$-population response about its steady state (*Figure 3C*). In the following analyses, we select 1 Hz$^2$ variance such that the two emergent properties do not overlap in $s_E(\mathbf{z};\mathbf{x})$.

First, we ran EPI to obtain parameter distribution $q_\theta(\mathbf{z} \mid \mathcal{X}(5\text{Hz}))$ producing E-population variability around 5 Hz (*Figure 3D*). From the marginal distribution of $\sigma_E$ and $\sigma_P$ (*Figure 3D*, top-left), we can see that $s_E(\mathbf{x};\mathbf{z})$ is sensitive to various combinations of $\sigma_E$ and $\sigma_P$. Alternatively, both $\sigma_S$ and $\sigma_V$ are degenerate with respect to $s_E(\mathbf{x};\mathbf{z})$ evidenced by the unexpectedly high variability in those dimensions (*Figure 3D*, bottom-right). Together, these observations imply a curved path with respect to $s_E(\mathbf{x};\mathbf{z})$ of 5 Hz, which is indicated by the modes along $\sigma_P$ (*Figure 3E*).

*Figure 3E* suggests a quadratic relationship in E-population fluctuations and the standard deviation of E- and P-population input; as the square of either $\sigma_E$ or $\sigma_P$ increases, the other compensates by decreasing to preserve the level of $s_E(\mathbf{x};\mathbf{z})$. This quadratic relationship is preserved at greater level of E-population variability $\mathcal{X}(10\text{Hz})$ (*Figure 3E* and *Figure 3—figure supplement 1*). Indeed, the sum of squares of $\sigma_E$ and $\sigma_P$ is larger in $q_\theta(\mathbf{z} \mid \mathcal{X}(10\text{Hz}))$ than $q_\theta(\mathbf{z} \mid \mathcal{X}(5\text{Hz}))$ (*Figure 3F*, $p < 1 \times 10^{-10}$), while the sum of squares of $\sigma_S$ and $\sigma_V$ are not significantly different in the two EPI distributions (*Figure 3—figure supplement 3*, $p = .40$), in which parameters were bounded from 0 to 0.5. The strong interaction between E- and P-population input variability on excitatory variability is intriguing, since this circuit exhibits a paradoxical effect in the P-population (and no other inhibitory types) (*Figure 3—figure supplement 4*), meaning that the E-population is P-stabilized. Future research may uncover a link between the population of network stabilization and compensatory interactions governing excitatory variability.

EPI revealed the quadratic dependence of excitatory variability on input variability to the E- and P-populations, as well as its independence to input from the other two inhibitory populations. In a simplified model ($\tau = \tau_{\text{noise}}$), it can be shown that surfaces of equal variance are ellipsoids as a function of $\sigma$ (see Section 'Primary visual cortex: Mathematical intuition and challenges'). Nevertheless, the sensitive and degenerate parameters are intractable to predict mathematically, since the covariance matrix depends on the steady-state solution of the network (*Hennequin et al., 2018*; *Gardiner, 2009*), and terms in the covariance expression increase quadratically with each additional neuron-type population (see also Section 'Primary visual cortex: Mathematical intuition and challenges'). By pointing out this mathematical complexity, we emphasize the value of EPI for gaining understanding about theoretical models when mathematical analysis becomes onerous or impractical.

## EPI identifies two regimes of rapid task switching

It has been shown that rats can learn to switch from one behavioral task to the next on randomly interleaved trials (*Duan et al., 2015*), and an important question is what neural mechanisms produce this computation. In this experimental setup, rats were given an explicit task cue on each trial, either Pro or Anti. After a delay period, rats were shown a stimulus, and made a context (task) dependent response (*Figure 4A*). In the Pro task, rats were required to orient toward the stimulus, while in the Anti task, rats were required to orient away from the stimulus. Pharmacological inactivation of the SC impaired rat performance, and time-specific optogenetic inactivation revealed a crucial role for the SC on the cognitively demanding Anti trials (*Duan et al., 2021*). These results motivated a

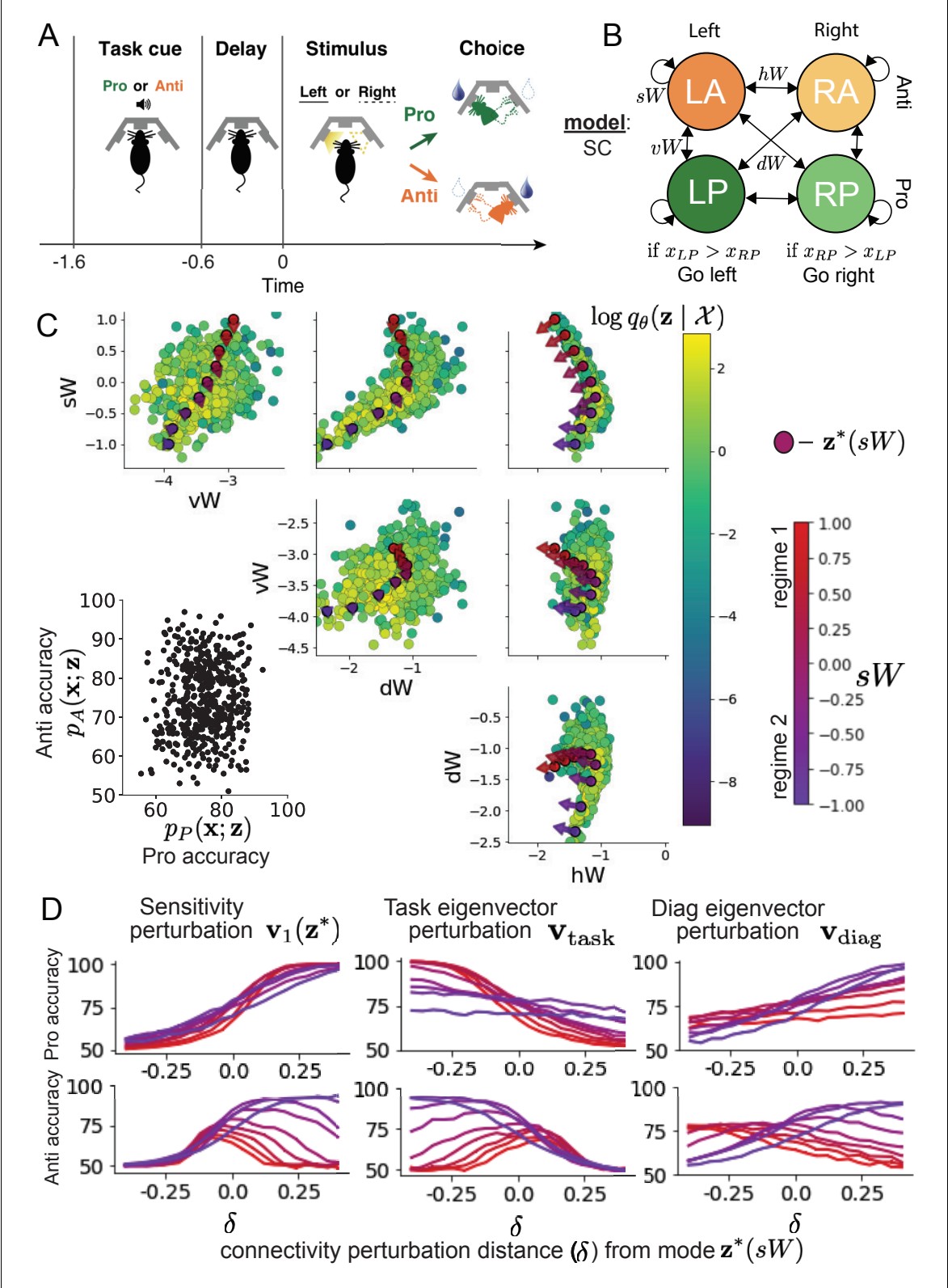

**Figure 4.** Inferring rapid task switching networks in superior colliculus. (A) Rapid task switching behavioral paradigm (see text). (B) Model of superior colliculus (SC). Neurons: LP - Left Pro, RP - Right Pro, LA - Left Anti, RA - Right Anti. Parameters: $sW$ - self, $hW$ - horizontal, $vW$ -vertical, $dW$ - diagonal weights. (C) The EPI inferred distribution of rapid task switching networks. Red/purple parameters indicate modes $\mathbf{z}^*(sW)$ colored by $sW$. Sensitivity

*Figure 4 continued on next page*

*Figure 4 continued*

vectors $\mathbf{v}_1(\mathbf{z}^*)$ are shown by arrows. (Bottom-left) EPI predictive distribution of task accuracies. (D) Mean and standard error ($N_{\text{test}}$ = 25, bars not visible) of accuracy in Pro (top) and Anti (bottom) tasks after perturbing connectivity away from mode along $\mathbf{v}_1(\mathbf{z}^*)$ (left), $\mathbf{v}_{\text{task}}$ (middle), and $\mathbf{v}_{\text{diag}}$ (right).

The online version of this article includes the following figure supplement(s) for figure 4:

**Figure supplement 1.** Task accuracy by EPI inferred SC network connectivity.
**Figure supplement 2.** SC network simulations by regime.
**Figure supplement 3.** Eigenmodes of SC connectivity.
**Figure supplement 4.** EPI optimization of the SC model producing rapid task switching.
**Figure supplement 5.** SC connectivities obtained through brute-force sampling.

nonlinear dynamical model of the SC containing four functionally defined neuron-type populations. In *Duan et al., 2021*, a computationally intensive procedure was used to obtain a set of 373 connectivity parameters that qualitatively reproduced these optogenetic inactivation results. To build upon the insights of this previous work, we use the probabilistic tools afforded by EPI to identify and characterize two linked, yet distinct regimes of rapid task switching connectivity.

In this SC model, there are Pro- and Anti-populations in each hemisphere (left (L) and right (R)) with activity variables $\mathbf{x} = [x_{LP}, x_{LA}, x_{RP}, x_{RA}]^\top$ (*Duan et al., 2021*). The connectivity of these populations is parameterized by self $sW$, vertical $vW$, diagonal $dW$ and horizontal $hW$ connections (*Figure 4B*). The input $\mathbf{h}$ is comprised of a positive cue-dependent signal to the Pro- or Anti-populations, a positive stimulus-dependent input to either the Left or Right populations, and a choice-period input to the entire network (see Section 'SC model'). Model responses are bounded from 0 to 1 as a function $\phi$ of an internal variable $\mathbf{u}$

$$\tau \frac{d\mathbf{u}}{dt} = -\mathbf{u} + W\mathbf{x} + \mathbf{h} + d\mathbf{B}$$
$$\mathbf{x} = \phi(\mathbf{u}). \tag{8}$$

The model responds to the side with greater Pro neuron activation; for example the response is left if $x_{LP} > x_{RP}$ at the end of the trial. Here, we use EPI to determine the network connectivity $\mathbf{z} = [sW, vW, dW, hW]^\top$ that produces rapid task switching.

Rapid task switching is formalized mathematically as an emergent property with two statistics: accuracy in the Pro task $p_P(\mathbf{x}; \mathbf{z})$ and Anti task $p_A(\mathbf{x}; \mathbf{z})$. We stipulate that accuracy be on average 0.75 in each task with variance $.075^2$

$$\mathcal{X}: \quad \mathbb{E}_{\mathbf{z}} \begin{bmatrix} p_P(\mathbf{x}; \mathbf{z}) \\ p_A(\mathbf{x}; \mathbf{z}) \end{bmatrix} = \begin{bmatrix} .75 \\ .75 \end{bmatrix}$$
$$\text{Var}_{\mathbf{z}} \begin{bmatrix} p_P(\mathbf{x}; \mathbf{z}) \\ p_A(\mathbf{x}; \mathbf{z}) \end{bmatrix} = \begin{bmatrix} .075^2 \\ .075^2 \end{bmatrix}. \tag{9}$$

Seventy-five percent accuracy is a realistic level of performance in each task, and with the chosen variance, inferred models will not exhibit fully random responses (50%), nor perfect performance (100%).

The EPI inferred distribution (*Figure 4C*) produces Pro- and Anti-task accuracies (*Figure 4C*, bottom-left) consistent with rapid task switching (*Equation 9*). This parameter distribution has rich structure that is not captured well by simple linear correlations (*Figure 4—figure supplement 1*). Specifically, the shape of the EPI distribution is sharply bent, matching ground truth structure indicated by brute-force sampling (*Figure 4—figure supplement 5*). This is most saliently observed in the marginal distribution of $sW$-$hW$ (*Figure 4C* top-right), where anticorrelation between $sW$ and $hW$ switches to correlation with decreasing $sW$. By identifying the modes of the EPI distribution $\mathbf{z}^*(sW)$ at different values of $sW$ (*Figure 4C* red/purple dots), we can quantify this change in distributional structure with the sensitivity dimension $\mathbf{v}_1(\mathbf{z})$ (*Figure 4C* red/purple arrows). Note that the directionality of these sensitivity dimensions at $\mathbf{z}^*(sW)$ changes distinctly with $sW$, and are perpendicular to the robust dimensions of the EPI distribution that preserve rapid task switching. These two directionalities of sensitivity motivate the distinction of connectivity into two regimes, which produce different types of responses in the Pro and Anti tasks (*Figure 4—figure supplement 2*).

When perturbing connectivity along the sensitivity dimension away from the modes

$$\mathbf{z} = \mathbf{z}^*(sW) + \delta\mathbf{v}_1(\mathbf{z}^*(sW)), \tag{10}$$

Pro-accuracy monotonically increases in both regimes (*Figure 4D*, top-left). However, there is a stark difference between regimes in Anti-accuracy. Anti-accuracy falls in either direction of $\mathbf{v}_1$ in regime 1, yet monotonically increases along with Pro accuracy in regime 2 (*Figure 4D*, bottom-left). The sharp change in local structure of the EPI distribution is therefore explained by distinct sensitivities: Anti-accuracy diminishes in only one or both directions of the sensitivity perturbation.

To understand the mechanisms differentiating the two regimes, we can make connectivity perturbations along dimensions that only modify a single eigenvalue of the connectivity matrix. These eigenvalues $\lambda_{\mathrm{all}}$, $\lambda_{\mathrm{side}}$, $\lambda_{\mathrm{task}}$, and $\lambda_{\mathrm{diag}}$ correspond to connectivity eigenmodes with intuitive roles in processing in this task (*Figure 4—figure supplement 3A*). For example, greater $\lambda_{\mathrm{task}}$ will strengthen internal representations of task, while greater $\lambda_{\mathrm{diag}}$ will amplify dominance of Pro and Anti pairs in opposite hemispheres (Section 'Connectivity eigendecomposition and processing modes'). Unlike the sensitivity dimension, the dimensions $\mathbf{v}_a$ that perturb isolated connectivity eigenvalues $\lambda_a$ for $a \in \{\mathrm{all}, \mathrm{side}, \mathrm{task}, \mathrm{diag}\}$ are independent of $\mathbf{z}^*(sW)$ (see Section 'Connectivity eigendecomposition and processing modes'), e.g.

$$\mathbf{z} = \mathbf{z}^*(sW) + \delta\mathbf{v}_{\mathrm{task}}. \tag{11}$$

Connectivity perturbation analyses reveal that decreasing $\lambda_{\mathrm{task}}$ has a very similar effect on Anti accuracy as perturbations along the sensitivity dimension (*Figure 4D*, middle). The similar effects of perturbations along the sensitivity dimension $\mathbf{v}_1(\mathbf{z}^*)$ and reduction of task eigenvalue (via perturbations along $-\mathbf{v}_{\mathrm{task}}$) suggest that there is a carefully tuned strength of task representation in connectivity regime 1, which if disturbed results in random Anti-trial responses. Finally, we recognize that increasing $\lambda_{\mathrm{diag}}$ has opposite effects on Anti-accuracy in each regime (*Figure 4D*, right). In the next section, we build on these mechanistic characterizations of each regime by examining their resilience to optogenetic inactivation.

## EPI inferred SC connectivities reproduce results from optogenetic inactivation experiments

During the delay period of this task, the circuit must prepare to execute the correct task according to the presented cue. The circuit must then maintain a representation of task throughout the delay period, which is important for correct execution of the Anti-task. Duan et al. found that bilateral optogenetic inactivation of SC during the delay period consistently decreased performance in the Anti-task, but had no effect on the Pro-task (*Figure 5A*; *Duan et al., 2021*). The distribution of connectivities inferred by EPI exhibited this same effect in simulation at high optogenetic strengths $\gamma$, which reduce the network activities $\mathbf{x}(t)$ by a factor $1 - \gamma$ (*Figure 5B*) (see Section 'Modeling optogenetic silencing').

To examine how connectivity affects response to delay period inactivation, we grouped connectivities of the EPI distribution along the continuum linking regimes 1 and 2 of Section 'EPI identifies two regimes of rapid task switching'. $Z(sW)$ is the set of EPI samples for which the closest mode was $\mathbf{z}^*(sW)$ (see Section 'Mode identification with EPI'). In the following analyses, we examine how error, and the influence of connectivity eigenvalue on Anti-error change along this continuum of connectivities. Obtaining the parameter samples for these analysis with the learned EPI distribution was more than 20,000 times faster than a brute force approach (see Section 'Sample grouping by mode').

The mean increase in Anti-error of the EPI distribution is closest to the experimentally measured value of 7% at $\gamma = 0.675$ (*Figure 5B*, black dot). At this level of optogenetic strength, regime 1 exhibits an increase in Anti-error with delay period silencing (*Figure 5C*, left), while regime 2 does not. In regime 1, greater $\lambda_{\mathrm{task}}$ and $\lambda_{\mathrm{diag}}$ decrease Anti-error (*Figure 5C*, right). In other words, stronger task representations and diagonal amplification make the SC model more resilient to delay period silencing in the Anti-task. This complements the finding from *Duan et al., 2021* (*Duan et al., 2021*) that $\lambda_{\mathrm{task}}$ and $\lambda_{\mathrm{diag}}$ improve Anti accuracy.

At roughly $\gamma = 0.85$ (*Figure 5B*, gray dot), the Anti-error saturates, while Pro-error remains at zero. Following delay period inactivation at this optogenetic strength, there are strong similarities in the responses of Pro- and Anti-trials during the choice period (*Figure 5D*, left). We interpreted these

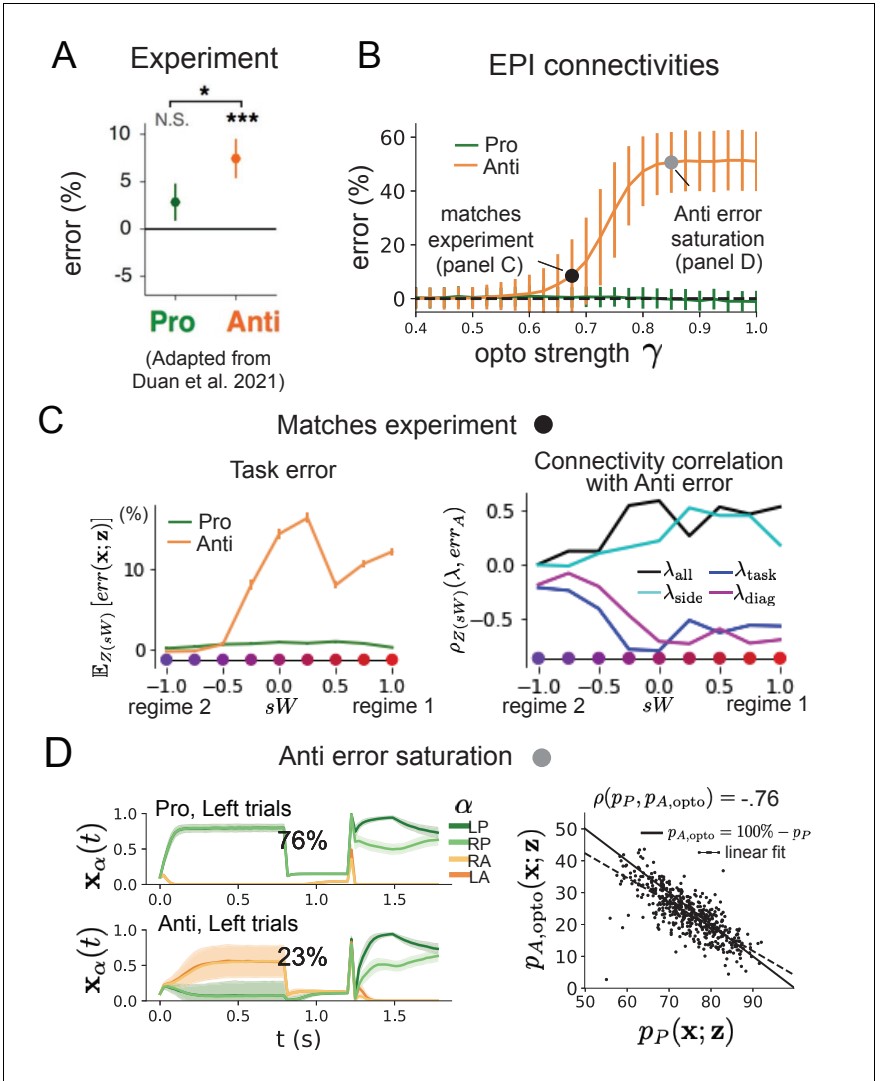

**Figure 5.** Responses to optogenetic perturbation by connectivity regime. (**A**) Mean and standard error (bars) across recording sessions of task error following delay period optogenetic inactivation in rats. (**B**) Mean and standard deviation (bars) of task error induced by delay period inactivation of varying optogenetic strength γ across the EPI distribution. (**C**) (Left) Mean and standard error of Pro and Anti error from regime 1 to regime 2 at $\gamma = 0.675$. (Right) Correlations of connectivity eigenvalues with Anti error from regime 1 to regime 2 at $\gamma = 0.675$. (**D**) (Left) Mean and standard deviation (shading) of responses of the SC model at the mode of the EPI distribution to delay period inactivation at $\gamma = 0.85$. Accuracy in Pro (top) and Anti (bottom) task is shown as a percentage. (Right) Anti-accuracy following delay period inactivation at $\gamma = 0.85$ versus accuracy in the Pro-task across connectivities in the EPI distribution.

The online version of this article includes the following figure supplement(s) for figure 5:

**Figure supplement 1.** SC responses to delay period inactivation at Anti error saturating levels.

**Figure supplement 2.** SC responses to delay period inactivation at experiment matching levels.

similarities to suggest that delay period inactivation at this saturated level flips the internal representation of task (from Anti to Pro) in the circuit model. A flipped task representation would explain why the Anti-error saturates at 50%: the average Anti-accuracy in EPI inferred connectivities is 75%, but average Anti accuracy would be 25% (100% - $\mathbb{E}_z[p_P]$) if the internal representation of task is flipped during the delay period.This hypothesis prescribes a model of Anti-accuracy during delay period silencing of $p_{A,\text{opto}} = 100\% - p_P$, which is fit closely across both regimes of the EPI inferred connectivities (**Figure 5D**, right). Similarities between Pro- and Anti-trial responses were not present

at the experiment-matching level of $\gamma = 0.675$ (*Figure 5—figure supplement 2* left) and neither was anticorrelation in $p_P$ and $p_{A,\text{opto}}$ (*Figure 5—figure supplement 2* right).

In summary, the connectivity inferred by EPI to perform rapid task switching replicated results from optogenetic silencing experiments. We found that at levels of optogenetic strength matching experimental levels of Anti-error, only one regime actually exhibited the effect. This connectivity regime is less resilient to optogenetic perturbation, and perhaps more biologically realistic. Finally, we characterized the pathology in Anti-error that occurs in both regimes when optogenetic strength is increased to high levels, leading to a mechanistic hypothesis that is experimentally testable. The probabilistic tools afforded by EPI yielded this insight: we identified two regimes and the continuum of connectivities between them by taking gradients of parameter probabilities in the EPI distribution, we identified sensitivity dimensions by measuring the Hessian of the EPI distribution, and we obtained many parameter samples at each step along the continuum at an efficient rate.

## Discussion

In neuroscience, machine learning has primarily been used to reveal structure in neural datasets (*Paninski and Cunningham, 2018*). Careful inference procedures are developed for these statistical models allowing precise, quantitative reasoning, which clarifies the way data informs beliefs about the model parameters. However, these statistical models often lack resemblance to the underlying biology, making it unclear how to go from the structure revealed by these methods, to the neural mechanisms giving rise to it. In contrast, theoretical neuroscience has primarily focused on careful models of neural circuits and the production of emergent properties of computation, rather than measuring structure in neural datasets. In this work, we improve upon parameter inference techniques in theoretical neuroscience with emergent property inference, harnessing deep learning towards parameter inference in neural circuit models (see Section 'Related approaches').

Methodology for statistical inference in circuit models has evolved considerably in recent years. Early work used rejection sampling techniques (*Beaumont et al., 2002*; *Marjoram et al., 2003*; *Sisson et al., 2007*), but EPI and another recently developed methodology (*Gonçalves et al., 2019*) employ deep learning to improve efficiency and provide flexible approximations. SNPE has been used for posterior inference of parameters in circuit models conditioned upon exemplar data used to represent computation, but it does not infer parameter distributions that only produce the computation of interest like EPI (see Section 'Scaling inference of recurrent neural network connectivity with EPI'). When strict control over the predictions of the inferred parameters is necessary, EPI uses a constrained optimization technique (*Loaiza-Ganem et al., 2017*) (see Section 'Augmented lagrangian optimization') to make inference conditioned on the emergent property possible.

A key difference between EPI and SNPE, is that EPI uses gradients of the emergent property throughout optimization. In Section 'Scaling inference of recurrent neural network connectivity with EPI', we showed that such gradients confer beneficial scaling properties, but a concern remains that emergent property gradients may be too computationally intensive. Even in a case of close biophysical realism with an expensive emergent property gradient, EPI was run successfully on intermediate hub frequency in a five-neuron subcircuit model of the STG (Section 'Motivating emergent property inference of theoretical models'). However, conditioning on the pyloric rhythm (*Marder and Selverston, 1992*) in a model of the pyloric subnetwork model (*Prinz et al., 2004*) proved to be prohibitive with EPI. The pyloric subnetwork requires many time steps for simulation and many key emergent property statistics (e.g. burst duration and phase gap) are not calculable or easily approximated with differentiable functions. In such cases, SNPE, which does not require differentiability of the emergent property, has proven useful (*Gonçalves et al., 2019*). In summary, choice of deep inference technique should consider emergent property complexity and differentiability, dimensionality of parameter space, and the importance of constraining the model behavior predicted by the inferred parameter distribution.

In this paper, we demonstrate the value of deep inference for parameter sensitivity analyses at both the local and global level. With these techniques, flexible deep probability distributions are optimized to capture global structure by approximating the full distribution of suitable parameters. Importantly, the local structure of this deep probability distribution can be quantified at any parameter choice, offering instant sensitivity measurements after fitting. For example, the global structure captured by EPI revealed two distinct parameter regimes, which had different local structure

quantified by the deep probability distribution (see Section 'Superior colliculus'). In comparison, bayesian MCMC is considered a popular approach for capturing global parameter structure (*Girolami and Calderhead, 2011*), but there is no variational approximation (the deep probability distribution in EPI), so sensitivity information is not queryable and sampling remains slow after convergence. Local sensitivity analyses (e.g. *Raue et al., 2009*) may be performed independently at individual parameter samples, but these methods alone do not capture the full picture in nonlinear, complex distributions. In contrast, deep inference yields a probability distribution that produces a wholistic assessment of parameter sensitivity at the local and global level, which we used in this study to make novel insights into a range of theoretical models. Together, the abilities to condition upon emergent properties, the efficient inference algorithm, and the capacity for parameter sensitivity analyses make EPI a useful method for addressing inverse problems in theoretical neuroscience.

## Code availability statement

All software written for this study is available at https://github.com/cunningham-lab/epi (copy archived at swh:1:rev:38febae7035ca921334a616b0f396b3767bf18d4), *Bittner, 2021*.

# Materials and methods

## Emergent property inference (EPI)

Solving inverse problems is an important part of theoretical neuroscience, since we must understand how neural circuit models and their parameter choices produce computations. Recently, research on machine learning methodology for neuroscience has focused on finding latent structure in large-scale neural datasets, while research in theoretical neuroscience generally focuses on developing precise neural circuit models that can produce computations of interest. By quantifying computation into an *emergent property* through statistics of the emergent activity of neural circuit models, we can adapt the modern technique of deep probabilistic inference towards solving inverse problems in theoretical neuroscience. Here, we introduce a novel method for statistical inference, which uses deep networks to learn parameter distributions constrained to produce emergent properties of computation.

Consider model parameterization $\mathbf{z}$, which is a collection of scientifically meaningful variables that govern the complex simulation of data $\mathbf{x}$. For example (see Section 'Motivating emergent property inference of theoretical models'), $\mathbf{z}$ may be the electrical conductance parameters of an STG subcircuit, and $\mathbf{x}$ the evolving membrane potentials of the five neurons. In terms of statistical modeling, this circuit model has an intractable likelihood $p(\mathbf{x} \mid \mathbf{z})$, which is predicated by the stochastic differential equations that define the model. From a theoretical perspective, we are less concerned about the likelihood of an exemplar dataset $\mathbf{x}$, but rather the emergent property of intermediate hub frequency (which implies a consistent dataset $\mathbf{x}$).

In this work, emergent properties $\mathcal{X}$ are defined through the choice of emergent property statistic $f(\mathbf{x}; \mathbf{z})$ (which is a vector of one or more statistics), and its means $\boldsymbol{\mu}$, and variances $\sigma^2$:

$$\mathcal{X} : \mathbb{E}_{\mathbf{z},\mathbf{x}}[f(\mathbf{x}; \mathbf{z})] = \boldsymbol{\mu}, \mathrm{Var}_{\mathbf{z},\mathbf{x}}[f(\mathbf{x}; \mathbf{z})] = \sigma^2. \tag{12}$$

In general, an emergent property may be a collection of first-, second-, or higher-order moments of a group of statistics, but this study focuses on the case written in *Equation 12*. In the STG example, intermediate hub frequency is defined by mean and variance constraints on the statistic of hub neuron frequency $\omega_{\mathrm{hub}}(\mathbf{x}; \mathbf{z})$ (*Equations 2 and 3*). Precisely, the emergent property statistics $f(\mathbf{x}; \mathbf{z})$ must have means $\boldsymbol{\mu}$ and variances $\sigma^2$ over the EPI distribution of parameters ($\mathbf{z} \sim q_{\boldsymbol{\theta}}(\mathbf{z})$) and the data produced by those parameters ($\mathbf{x} \sim p(\mathbf{x} \mid \mathbf{z})$), where the inferred parameter distribution $q_{\boldsymbol{\theta}}(\mathbf{z})$ itself is parameterized by deep network weights and biases $\boldsymbol{\theta}$.

In EPI, a deep probability distribution $q_{\boldsymbol{\theta}}(\mathbf{z})$ is optimized to approximate the parameter distribution producing the emergent property $\mathcal{X}$. In contrast to simpler classes of distributions like the gaussian or mixture of gaussians, deep probability distributions are far more flexible and capable of fitting rich structure (*Rezende and Mohamed, 2015*; *Papamakarios et al., 2019a*). In deep probability distributions, a simple random variable $\mathbf{z}_0 \sim q_0(\mathbf{z}_0)$ (we choose an isotropic gaussian) is mapped

deterministically via a sequence of deep neural network layers ($g_1, . g_l$) parameterized by weights and biases $\boldsymbol{\theta}$ to the support of the distribution of interest:

$$\mathbf{z} = g_{\boldsymbol{\theta}}(\mathbf{z}_0) = g_l(...g_1(\mathbf{z}_0)) \sim q_{\boldsymbol{\theta}}(\mathbf{z}). \qquad (13)$$

Such deep probability distributions embed the inferred distribution in a deep network. Once optimized, this deep network representation of a distribution has remarkably useful properties: fast sampling and probability evaluations. Importantly, fast probability evaluations confer fast gradient and Hessian calculations as well.

Given this choice of circuit model and emergent property $\mathcal{X}$, $q_{\boldsymbol{\theta}}(\mathbf{z})$ is optimized via the neural network parameters $\boldsymbol{\theta}$ to find a maximally entropic distribution $q_{\boldsymbol{\theta}}^*$ within the deep variational family $\mathcal{Q} = \{q_{\boldsymbol{\theta}}(\mathbf{z}) : \boldsymbol{\theta} \in \Theta\}$ that produces the emergent property $\mathcal{X}$:

$$\begin{aligned} q_{\boldsymbol{\theta}}(\mathbf{z} \mid \mathcal{X}) = q_{\boldsymbol{\theta}}^*(\mathbf{z}) \quad &\underset{q_{\boldsymbol{\theta}} \in \mathcal{Q}}{\operatorname{argmax}} \, H(q_{\boldsymbol{\theta}}(\mathbf{z})) \\ &\mathrm{s.t.} \, \mathcal{X} : \mathbb{E}_{\mathbf{z}, \mathbf{x}}[f(\mathbf{x}; \mathbf{z})] = \boldsymbol{\mu}, \mathrm{Var}_{\mathbf{z}, \mathbf{x}}[f(\mathbf{x}; \mathbf{z})] = \sigma^2, \end{aligned} \qquad (14)$$

where $H(q_{\boldsymbol{\theta}}(\mathbf{z})) = \mathbb{E}_{\mathbf{z}}[-\log q_{\boldsymbol{\theta}}(z)]$ is entropy. By maximizing the entropy of the inferred distribution $q_{\boldsymbol{\theta}}$, we select the most random distribution in family $\mathcal{Q}$ that satisfies the constraints of the emergent property. Since entropy is maximized in *Equation 14*, EPI is equivalent to bayesian variational inference (see Section 'EPI as variational inference'), which is why we specify the inferred distribution of EPI as conditioned upon emergent property $\mathcal{X}$ with the notation $q_{\boldsymbol{\theta}}(\mathbf{z} \mid \mathcal{X})$. To run this constrained optimization, we use an augmented lagrangian objective, which is the standard approach for constrained optimization (*Bertsekas, 2014*), and the approach taken to fit Maximum Entropy Flow Networks (MEFNs) (*Loaiza-Ganem et al., 2017*). This procedure is detailed in Section 'Augmented lagrangian optimization' and the pseudocode in Algorithm 'Augmented lagrangian optimization'.

In the remainder of Section 'Emergent property inference (EPI)', we will explain the finer details and motivation of the EPI method. First, we explain related approaches and what EPI introduces to this domain (Section 'Related approaches'). Second, we describe the special class of deep probability distributions used in EPI called normalizing flows (Section 'Deep probability distributions and normalizing flows'). Then, we establish the known relationship between maximum entropy distributions and exponential families (Section 'Maximum entropy distributions and exponential families'). Next, we explain the constrained optimization technique used to solve *Equation 14* (Section 'Augmented lagrangian optimization'). Then, we demonstrate the details of this optimization in a toy example (Section 'Example: 2D LDS'). Finally, we explain how EPI is equivalent to variational inference (Section 'EPI as variational inference').

## Related approaches

When bayesian inference problems lack conjugacy, scientists use approximate inference methods like variational inference (VI) (*Saul and Jordan, 1998*) and Markov chain Monte Carlo (MCMC) (*Metropolis et al., 1953*; *Hastings, 1970*). After optimization, variational methods return a parameterized posterior distribution, which we can analyze. Also, the variational approximation is often chosen such that it permits fast sampling. In contrast MCMC methods only produce samples from the approximated posterior distribution. No parameterized distribution is estimated, and additional samples are always generated with the same sampling complexity. Inference in models defined by systems of differential has been demonstrated with MCMC (*Girolami and Calderhead, 2011*), although this approach requires tractable likelihoods. Advancements have introduced sampling (*Calderhead and Girolami, 2011*), likelihood approximation (*Golightly and Wilkinson, 2011*), and uncertainty quantification techniques (*Chkrebtii et al., 2016*) to make MCMC approaches more efficient and expand the class of applicable models.

Simulation-based inference (*Cranmer et al., 2020*) is model parameter inference in the absence of a tractable likelihood function. The most prevalent approach to simulation-based inference is approximate bayesian computation (ABC) (*Beaumont et al., 2002*), in which satisfactory parameter samples are kept from random prior sampling according to a rejection heuristic. The obtained set of parameters do not have a probabilities, and further insight about the model must be gained from examination of the parameter set and their generated activity. Methodological advances to ABC methods have come through the use of Markov chain Monte Carlo (MCMC-ABC) (*Marjoram et al.,*

*2003*) and sequential Monte Carlo (SMC-ABC) (*Sisson et al., 2007*) sampling techniques. SMC-ABC is considered state-of-the-art ABC, yet this approach still struggles to scale in dimensionality (*Sisson et al., 2018*; *Figure 2*). Still, this method has enjoyed much success in systems biology (*Liepe et al., 2014*). Furthermore, once a parameter set has been obtained by SMC-ABC from a finite set of particles, the SMC-ABC algorithm must be run again from scratch with a new population of initialized particles to obtain additional samples.

For scientific model analysis, we seek a parameter distribution represented by an approximating distribution as in variational inference (*Saul and Jordan, 1998*): a variational approximation that once optimized yields fast analytic calculations and samples. For the reasons described above, ABC and MCMC techniques are not suitable, because they only produce a set of parameter samples lacking probabilities and have unchanging sampling rate. EPI infers parameters in circuit models using the MEFN (*Loaiza-Ganem et al., 2017*) algorithm with a deep variational approximation. The deep neural network of EPI (*Figure 1E*) defines the parametric form (with weights and biases as variational parameters $\theta$) of the variational approximation of the inferred parameter distribution $q_\theta(\mathbf{z} \mid \mathbf{x})$. The EPI optimization is enabled using stochastic gradient techniques in the spirit of likelihood-free variational inference (*Tran et al., 2017*). The analytic relationship between EPI and variational inference is explained in Section 'EPI as variational inference'.

We note that, during our preparation and early presentation of this work (*Bittner et al., 2019a*; *Bittner et al., 2019b*), another work has arisen with broadly similar goals: bringing statistical inference to mechanistic models of neural circuits (*Nonnenmacher et al., 2018*; *Michael et al., 2019*; *Gonçalves et al., 2019*). We are encouraged by this general problem being recognized by others in the community, and we emphasize that these works offer complementary neuroscientific contributions (different theoretical models of focus) and use different technical methodologies (ours is built on our prior work [*Loaiza-Ganem et al., 2017*], theirs similarly [*Lueckmann et al., 2017*]).

The method EPI differs from SNPE in some key ways. SNPE belongs to a 'sequential' class of recently developed simulation-based inference methods in which two neural networks are used for posterior inference. This first neural network is a deep probability distribution (normalizing flow) used to estimate the posterior $p(\mathbf{z} \mid \mathbf{x})$ (SNPE) or the likelihood $p(\mathbf{x} \mid \mathbf{z})$ (sequential neural likelihood (SNL) [*Papamakarios et al., 2019b*]). A recent approach uses an unconstrained neural network to estimate the likelihood ratio (sequential neural ratio estimation (SNRE) [*Hermans et al., 2020*]). In SNL and SNRE, MCMC sampling techniques are used to obtain samples from the approximated posterior. This contrasts with EPI and SNPE, which use deep probability distributions to model parameters, which facilitates immediate measurements of sample probability, gradient, or Hessian for system analysis. The second neural network in this sequential class of methods is the amortizer. This unconstrained deep network maps data $\mathbf{x}$ (or statistics $f(\mathbf{x}; \mathbf{z})$ or model parameters $\mathbf{z}$) to the weights and biases of the first neural network. These methods are optimized on a conditional density (or ratio) estimation objective. The data used to optimize this objective are generated via an adaptive procedure, in which training data pairs ($\mathbf{x}_i$, $\mathbf{z}_i$) become sequentially closer to the true data and posterior.

The approximating fidelity of the deep probability distribution in sequential approaches is optimized to generalize across the training distribution of the conditioning variable. This generalization property of the sequential methods can reduce the accuracy at the singular posterior of interest. Whereas in EPI, the entire expressivity of the deep probability distribution is dedicated to learning a single distribution as well as possible. The well-known inverse mapping problem of exponential families (*Wainwright and Jordan, 2008*) prohibits an amortization-based approach in EPI, since EPI learns an exponential family distribution parameterized by its mean (in contrast to its natural parameter, see Section 'Maximum entropy distributions and exponential families'). However, we have shown that the same two-network architecture of the sequential simulation-based inference methods can be used for amortized inference in intractable exponential family posteriors when using their natural parameterization (*Bittner and Cunningham, 2019*).

Finally, one important differentiating factor between EPI and sequential simulation-based inference methods is that EPI leverages gradients $\nabla_\mathbf{z} f(\mathbf{x}; \mathbf{z})$ during optimization. These gradients can improve convergence time and scalability, as we have shown on an example conditioning low-rank RNN connectivity on the property of stable amplification (see Section 'Scaling inference of recurrent neural network connectivity with EPI'). With EPI, we prove out the suggestion that a deep inference technique can improve efficiency by leveraging these emergent property gradients when they are

tractable. Sequential simulation-based inference techniques may be better suited for scientific problems where $\nabla_{\mathbf{z}}f(\mathbf{x};\mathbf{z})$ is intractable or unavailable, like when there is a nondifferentiable emergent property. However, the sequential simulation-based inference techniques cannot constrain the predictions of the inferred distribution in the manner of EPI.

Structural identifiability analysis involves the measurement of sensitivity and unidentifiabilities in scientific models. Around a single parameter choice, one can measure the Jacobian. One approach for this calculation that scales well is EAR (*Karlsson et al., 2012*). A popular efficient approach for systems of ODEs has been neural ODE adjoint (*Chen et al., 2018*) and its stochastic adaptation (*Li et al., 2020*). Casting identifiability as a statistical estimation problem, the profile likelihood works via iterated optimization while holding parameters fixed (*Raue et al., 2009*). An exciting recent method is capable of recovering the functional form of such unidentifiabilities away from a point by following degenerate dimensions of the fisher information matrix (*Raman et al., 2017*). Global structural non-identifiabilities can be found for models with polynomial or rational dynamics equations using DAISY (*Pia Saccomani et al., 2003*), or through mean optimal transformations (*Hengl et al., 2007*). With EPI, we have all the benefits given by a statistical inference method plus the ability to query the first- or second-order gradient of the probability of the inferred distribution at any chosen parameter value. The second-order gradient of the log probability (the Hessian), which is directly afforded by EPI distributions, produces quantified information about parametric sensitivity of the emergent property in parameter space (see Section 'Emergent property inference via deep generative models').

## Deep probability distributions and normalizing flows

Deep probability distributions are comprised of multiple layers of fully connected neural networks (*Equation 13*). When each neural network layer is restricted to be a bijective function, the sample density can be calculated using the change of variables formula at each layer of the network. For $\mathbf{z}_i = g_i(\mathbf{z}_{i-1})$,

$$p(\mathbf{z}_i) = p(g_i^{-1}(\mathbf{z}_i))\left|\det\frac{\partial g_i^{-1}(\mathbf{z}_i)}{\partial \mathbf{z}_i}\right| = p(\mathbf{z}_{i-1})\left|\det\frac{\partial g_i(\mathbf{z}_{i-1})}{\partial \mathbf{z}_{i-1}}\right|^{-1}. \tag{15}$$

However, this computation has cubic complexity in dimensionality for fully connected layers. By restricting our layers to normalizing flows (*Rezende and Mohamed, 2015*; *Papamakarios et al., 2019a*) – bijective functions with fast log determinant Jacobian computations, which confer a fast calculation of the sample log probability. Fast log probability calculation confers efficient optimization of the maximum entropy objective (see Section 'Augmented lagrangian optimization').

We use the real NVP (*Dinh et al., 2017*) normalizing flow class, because its coupling architecture confers both fast sampling (forward) and fast log probability evaluation (backward). Fast probability evaluation facilitates fast gradient and Hessian evaluation of log probability throughout parameter space. Glow permutations were used in between coupling stages (*Kingma and Dhariwal, 2018*). This is in contrast to autoregressive architectures (*Papamakarios et al., 2017*; *Kingma et al., 2016*), in which only one of the forward or backward passes can be efficient. In this work, normalizing flows are used as flexible parameter distribution approximations $q_{\boldsymbol{\theta}}(\mathbf{z})$ having weights and biases $\boldsymbol{\theta}$. We specify the architecture used in each application by the number of real NVP affine coupling stages, and the number of neural network layers and units per layer of the conditioning functions.

When calculating Hessians of log probabilities in deep probability distributions, it is important to consider the normalizing flow architecture. With autoregressive architectures (*Kingma et al., 2016*; *Papamakarios et al., 2017*), fast sampling and fast log probability evaluations are mutually exclusive. That makes these architectures undesirable for EPI, where efficient sampling is important for optimization, and log probability evaluation speed predicates the efficiency of gradient and Hessian calculations. With real NVP coupling architectures, we get both fast sampling and fast Hessians making both optimization and scientific analysis efficient.

## Maximum entropy distributions and exponential families

The inferred distribution of EPI is a maximum entropy distribution, which have fundamental links to exponential family distributions. A maximum entropy distribution of form:

$$p^*(\mathbf{z}) \quad = \underset{p \in \mathcal{P}}{\arg\max} \, H(p(\mathbf{z}))$$
$$\text{s.t.} \, \mathbb{E}_{\mathbf{z} \sim p}[T(\mathbf{z})] = \boldsymbol{\mu}_{\text{opt}}, \tag{16}$$

where $T(\mathbf{z})$ is the sufficient statistics vector and $\boldsymbol{\mu}_{\text{opt}}$ a vector of their mean values, will have probability density in the exponential family:

$$p^*(\mathbf{z}) \propto \exp(\boldsymbol{\eta}^\top T(\mathbf{z})). \tag{17}$$

The mappings between the mean parameterization $\boldsymbol{\mu}_{\text{opt}}$ and the natural parameterization $\eta$ are formally hard to identify except in special cases (*Wainwright and Jordan, 2008*).

In this manuscript, emergent properties are defined by statistics $f(\mathbf{x}; \mathbf{z})$ having a fixed mean $\boldsymbol{\mu}$ and variance $\sigma^2$ as in *Equation 12*. The variance constraint is a second moment constraint on $f(\mathbf{x}; \mathbf{z})$:

$$\text{Var}_{\mathbf{z}, \mathbf{x}}[f(\mathbf{x}; \mathbf{z})] = \mathbb{E}_{\mathbf{z}, \mathbf{x}} \left[ (f(\mathbf{x}; \mathbf{z}) - \boldsymbol{\mu})^2 \right]. \tag{18}$$

As a general maximum entropy distribution (*Equation 16*), the sufficient statistics vector contains both first and second order moments of $f(\mathbf{x}; \mathbf{z})$

$$T(\mathbf{z}) = \begin{bmatrix} \mathbb{E}_{\mathbf{x} \sim p(\mathbf{x} \mid \mathbf{z})}[f(\mathbf{x}; \mathbf{z})] \\ \mathbb{E}_{\mathbf{x} \sim p(\mathbf{x} \mid \mathbf{z})} \left[ (f(\mathbf{x}; \mathbf{z}) - \boldsymbol{\mu})^2 \right] \end{bmatrix}, \tag{19}$$

which are constrained to the chosen means and variances

$$\boldsymbol{\mu}_{\text{opt}} = \begin{bmatrix} \boldsymbol{\mu} \\ \sigma^2 \end{bmatrix}. \tag{20}$$

Thus, $\boldsymbol{\mu}_{\text{opt}}$ is used to denote the mean parameter of the maximum entropy distribution defined by the emergent property (all constraints), while $\boldsymbol{\mu}$ is only the mean of $f(\mathbf{x}; \mathbf{z})$. The subscript 'opt' of $\boldsymbol{\mu}_{\text{opt}}$ is chosen since it contains all the constraint values to which the EPI optimization algorithm must adhere.

## Augmented lagrangian optimization

To optimize $q_\theta(\mathbf{z})$ in *Equation 14*, the constrained maximum entropy optimization is executed using the augmented lagrangian method. The following objective is minimized:

$$L(\boldsymbol{\theta}; \eta_{\text{opt}}, c) = -H(q_{\boldsymbol{\theta}}) + \eta_{\text{opt}}^\top R(\boldsymbol{\theta}) + \frac{c}{2} ||R(\boldsymbol{\theta})||^2 \tag{21}$$

where there are average constraint violations

$$R(\boldsymbol{\theta}) = \mathbb{E}_{\mathbf{z} \sim q_{\boldsymbol{\theta}}(\mathbf{z})} \left[ T(\mathbf{z}) - \boldsymbol{\mu}_{\text{opt}} \right], \tag{22}$$

$\eta_{\text{opt}} \in \mathbb{R}^m$ are the lagrange multipliers where $m$ is the number of total constraints

$$m = |\boldsymbol{\mu}_{\text{opt}}| = |T(\mathbf{z})| = 2|f(\mathbf{x}; \mathbf{z})|, \tag{23}$$

and $c$ is the penalty coefficient. The mean parameter $\boldsymbol{\mu}_{\text{opt}}$ and sufficient statistics $T(\mathbf{z})$ are determined by the means $\boldsymbol{\mu}$ and variances $\sigma^2$ of the emergent property statistics $f(\mathbf{x}; \mathbf{z})$ defined in *Equation 14*. Specifically, $T(\mathbf{z})$ is a concatenation of the first and second moments (*Equation 19*) and $\boldsymbol{\mu}_{\text{opt}}$ is a concatenation of their constraints $\boldsymbol{\mu}$ and $\sigma^2$ (*Equation 20*). (Although, note that this algorithm is written for general $T(\mathbf{z})$ and $\boldsymbol{\mu}_{\text{opt}}$ to satisfy the more general class of emergent properties.) The lagrange multipliers $\eta_{\text{opt}}$ are closely related to the natural parameters $\eta$ of exponential families (see Section 'EPI as variational inference'). Weights and biases $\boldsymbol{\theta}$ of the deep probability distribution are optimized according to *Equation 21* using the Adam optimizer with learning rate $10^{-3}$ (*Kingma and Ba, 2015*).

The gradient with respect to entropy $H(q_{\boldsymbol{\theta}}(\mathbf{z}))$ can be expressed using the reparameterization trick as an expectation of the negative log density of parameter samples $\mathbf{z}$ over the randomness in the parameterless initial distribution $q_0(\mathbf{z}_0)$:

$$H(q_{\boldsymbol{\theta}}(\mathbf{z})) = \int -q_{\boldsymbol{\theta}}(\mathbf{z})\log(q_{\boldsymbol{\theta}}(\mathbf{z}))d\mathbf{z} = \mathbb{E}_{\mathbf{z} \sim q_{\boldsymbol{\theta}}}[-\log(q_{\boldsymbol{\theta}}(\mathbf{z}))] = \mathbb{E}_{\mathbf{z}_0 \sim q_0}[-\log(q_{\boldsymbol{\theta}}(g_{\boldsymbol{\theta}}(\mathbf{z}_0)))]. \qquad (24)$$

Thus, the gradient of the entropy of the deep probability distribution can be estimated as an average of gradients with respect to the base distribution $\mathbf{z}_0$:

$$\nabla_{\boldsymbol{\theta}}H(q_{\boldsymbol{\theta}}(\mathbf{z})) = \mathbb{E}_{\mathbf{z}_0 \sim q_0}[-\nabla_{\boldsymbol{\theta}}\log(q_{\boldsymbol{\theta}}(g_{\boldsymbol{\theta}}(\mathbf{z}_0)))]. \qquad (25)$$

The gradients of the log density of the deep probability distribution are tractable through the use of normalizing flows (see Section 'Deep probability distributions and normalizing flows').

The full EPI optimization algorithm is detailed in Algorithm 1. The lagrangian parameters $\eta_{\mathrm{opt}}$ are initialized to zero and adapted following each augmented lagrangian epoch, which is a period of optimization with fixed $(\eta_{\mathrm{opt}}, c)$ for a given number of stochastic gradient descent (SGD) iterations. A low value of $c$ is used initially, and conditionally increased after each epoch based on constraint error reduction. The penalty coefficient is updated based on the result of a hypothesis test regarding the reduction in constraint violation. The p-value of $\mathbb{E}[||R(\boldsymbol{\theta}_{k+1})||] > \gamma \mathbb{E}[||R(\boldsymbol{\theta}_k)||]$ is computed, and $c_{k+1}$ is updated to $\beta c_k$ with probability $1 - p$. The other update rule is $\eta_{\mathrm{opt},k+1} = \eta_{\mathrm{opt},k} + c_k \frac{1}{n}\sum_{i=1}^{n}(T(\mathbf{z}^{(i)}) - \boldsymbol{\mu}_{\mathrm{opt}})$ given a batch size $n$ and $\mathbf{z}^{(i)} \sim q_{\boldsymbol{\theta}}(\mathbf{z})$. Throughout the study, $\gamma = 0.25$, while $\beta$ was chosen to be either 2 or 4. The batch size of EPI also varied according to application.

---

**Algorithm 1. Emergent property inference**

---

1 initialize $\boldsymbol{\theta}$ by fitting $q_{\boldsymbol{\theta}}$ to an isotropic gaussian of mean $\boldsymbol{\mu}_{\mathrm{init}}$ and variance $\sigma^2_{\mathrm{init}}$
2 initialize $c_0 > 0$ and $\eta_{\mathrm{opt},0} = \mathbf{0}$.
3 for Augmented lagrangian epoch $k = 1, ..., k_{\max}$ do
4 for SGD iteration $i = 1, ..., i_{\max}$ do
5 Sample $\mathbf{z}_0^{(1)}, ..., \mathbf{z}_0^{(n)} \sim q_0$, get transformed variable $\mathbf{z}^{(j)} = g_{\boldsymbol{\theta}}(\mathbf{z}_0^{(j)})$, $j = 1, ..., n$
6 Update $\boldsymbol{\theta}$ by descending its stochastic gradient (using ADAM optimizer [***Kingma and Ba, 2015***]).

$$\nabla_{\boldsymbol{\theta}}L(\boldsymbol{\theta}; \eta_{\mathrm{opt},k}, c) = \frac{1}{n}\sum_{j=1}^{n}\nabla_{\boldsymbol{\theta}}\log q_{\boldsymbol{\theta}}(\mathbf{z}^{(j)}) + \frac{1}{n}\sum_{j=1}^{n}\nabla_{\boldsymbol{\theta}}\left(T\left(\mathbf{z}^{(j)}\right) - \boldsymbol{\mu}_{\mathrm{opt}}\right)\eta_{\mathrm{opt},k}$$

$$+ c_k \frac{2}{n}\sum_{j=1}^{\frac{n}{2}}\nabla_{\boldsymbol{\theta}}\left(T\left(\mathbf{z}^{(j)}\right) - \boldsymbol{\mu}_{\mathrm{opt}}\right) \cdot \frac{2}{n}\sum_{j=\frac{n}{2}+1}^{n}\left(T\left(\mathbf{z}^{(j)}\right) - \boldsymbol{\mu}_{\mathrm{opt}}\right)$$

7 end
8 Sample $\mathbf{z}_0^{(1)}, ..., \mathbf{z}_0^{(n)} \sim q_0$, get transformed variable $\mathbf{z}^{(j)} = g_{\boldsymbol{\theta}}(\mathbf{z}_0^{(j)})$, $j = 1, ..., n$
9 Update $\eta_{\mathrm{opt},k+1} = \eta_{\mathrm{opt},k} + c_k \frac{1}{n}\sum_{j=1}^{n}\left(T(\mathbf{z}^{(j)}) - \boldsymbol{\mu}_{\mathrm{opt}}\right)$.
10 Update $c_{k+1} > c_k$ (see text for detail).
11 end

---

In general, $c$ and $\eta_{\mathrm{opt}}$ should start at values encouraging entropic growth early in optimization. With each training epoch in which the update rule for $c$ is invoked, the constraint satisfaction terms are increasingly weighted, which generally results in decreased entropy (e.g. see ***Figure 1—figure supplement 1C***). This encourages the discovery of suitable regions of parameter space, and the subsequent refinement of the distribution to produce the emergent property. The momentum parameters of the Adam optimizer are reset at the end of each augmented lagrangian epoch, which proceeds for $i_{\max}$ iterations. In this work, we used a maximum number of augmented lagrangian epochs $k_{\max} >= 5$.

Rather than starting optimization from some $\boldsymbol{\theta}$ drawn from a randomized distribution, we found that initializing $q_{\boldsymbol{\theta}}(\mathbf{z})$ to approximate an isotropic gaussian distribution conferred more stable, consistent optimization. The parameters of the gaussian initialization were chosen on an application-specific basis. Throughout the study, we chose isotropic Gaussian initializations with mean $\boldsymbol{\mu}_{\mathrm{init}}$ at the center of the support of the distribution and some variance $\sigma^2_{\mathrm{init}}$, except for one case, where an initialization informed by random search was used (see Section 'Stomatogastric ganglion'). Deep probability distributions were fit to these gaussian initializations using 10,000 iterations of stochastic

gradient descent on the evidence lower bound (as in *Bittner and Cunningham, 2019*) with Adam optimizer and a learning rate of $10^{-3}$.

To assess whether the EPI distribution $q_{\boldsymbol{\theta}}(\mathbf{z})$ produces the emergent property, we assess whether each individual constraint on the means and variances of $f(\mathbf{x};\mathbf{z})$ is satisfied. We consider the EPI to have converged when a null hypothesis test of constraint violations $R(\boldsymbol{\theta})_i$ being zero is accepted for all constraints $i \in \{1, ..., m\}$ at a significance threshold $\alpha = 0.05$. This significance threshold is adjusted through Bonferroni correction according to the number of constraints $m$. The p-values for each constraint are calculated according to a two-tailed nonparametric test, where 200 estimations of the sample mean $R(\boldsymbol{\theta})^i$ are made using $N_{\text{test}}$ samples of $\mathbf{z} \sim q_{\boldsymbol{\theta}}(\mathbf{z})$ at the end of the augmented lagrangian epoch. Of all $k_{\max}$ augmented lagrangian epochs, we select the EPI inferred distribution as that which satisfies the convergence criteria and has greatest entropy.

When assessing the suitability of EPI for a particular modeling question, there are some important technical considerations. First and foremost, as in any optimization problem, the defined emergent property should always be appropriately conditioned (constraints should not have wildly different units). Furthermore, if the program is underconstrained (not enough constraints), the distribution grows (in entropy) unstably unless mapped to a finite support. If overconstrained, there is no parameter set producing the emergent property, and EPI optimization will fail (appropriately).

## Example: 2D LDS

To gain intuition for EPI, consider a two-dimensional linear dynamical system (2D LDS) model (*Figure 1—figure supplement 1A*):

$$\tau \frac{d\mathbf{x}}{dt} = A\mathbf{x} \tag{26}$$

with

$$A = \begin{bmatrix} a_{1,1} & a_{1,2} \\ a_{2,1} & a_{2,2} \end{bmatrix}. \tag{27}$$

To run EPI with the dynamics matrix elements as the free parameters $\mathbf{z} = [a_{1,1}, a_{1,2}, a_{2,1}, a_{2,2}]$ (fixing $\tau = 1$ s), the emergent property statistics $f(\mathbf{x};\mathbf{z})$ were chosen to contain parts of the primary eigenvalue of $A$, which predicate frequency, $\text{imag}(\lambda_1)$, and the growth/decay, $\text{real}(\lambda_1)$, of the system

$$f(\mathbf{x};\mathbf{z}) \triangleq \begin{bmatrix} \text{real}(\lambda_1)(\mathbf{x};\mathbf{z}) \\ \text{imag}(\lambda_1)(\mathbf{x};\mathbf{z}) \end{bmatrix} \tag{28}$$

$\lambda_1$ is the eigenvalue of greatest real part when the imaginary component is zero, and alternatively that of positive imaginary component when the eigenvalues are complex conjugate pairs. To learn the distribution of real entries of $A$ that produce a band of oscillating systems around 1 Hz, we formalized this emergent property as $\text{real}(\lambda_1)$ having mean zero with variance $0.25^2$, and the oscillation frequency $\frac{\text{imag}(\lambda_1)}{2\pi}$ having mean 1 Hz with variance 0.1 Hz$^2$:

$$
\mathcal{X}: \quad \mathbb{E}_{\mathbf{z},\mathbf{x}}[f(\mathbf{x};\mathbf{z})] \triangleq \mathbb{E}_{\mathbf{z},\mathbf{x}} \begin{bmatrix} \text{real}(\lambda_1)(\mathbf{x};\mathbf{z}) \\ \text{imag}(\lambda_1)(\mathbf{x};\mathbf{z}) \end{bmatrix} = \begin{bmatrix} 0 \\ 2\pi \end{bmatrix} \triangleq \boldsymbol{\mu}
$$

$$
\text{Var}_{\mathbf{z},\mathbf{x}}[f(\mathbf{x};\mathbf{z})] \triangleq \text{Var}_{\mathbf{z},\mathbf{x}} \begin{bmatrix} \text{real}(\lambda_1)(\mathbf{x};\mathbf{z}) \\ \text{imag}(\lambda_1)(\mathbf{x};\mathbf{z}) \end{bmatrix} = \begin{bmatrix} 0.25^2 \\ (\frac{\pi}{5})^2 \end{bmatrix} \triangleq \boldsymbol{\sigma}^2. \tag{29}
$$

To write the emergent property $\mathcal{X}$ in the form required for the augmented lagrangian optimization (Section 'Augmented lagrangian optimization'), we concatenate these first and second moment constraints into a vector of sufficient statistics $T(\mathbf{z})$ and constraint values $\boldsymbol{\mu}_{\text{opt}}$.

$$
\mathbb{E}_{\mathbf{z}}[T(\mathbf{z})] \triangleq \mathbb{E}_{\mathbf{z}} \begin{bmatrix} \mathbb{E}_{\mathbf{x} \sim p(\mathbf{x}\,|\,\mathbf{z})}[\text{real}(\lambda_1)(\mathbf{x};\mathbf{z})] \\ \mathbb{E}_{\mathbf{x} \sim p(\mathbf{x}\,|\,\mathbf{z})}[\text{imag}(\lambda_1)(\mathbf{x};\mathbf{z})] \\ \mathbb{E}_{\mathbf{x} \sim p(\mathbf{x}\,|\,\mathbf{z})}\left[(\text{real}(\lambda_1)(\mathbf{x};\mathbf{z}) - 0)^2\right] \\ \mathbb{E}_{\mathbf{x} \sim p(\mathbf{x}\,|\,\mathbf{z})}\left[(\text{imag}(\lambda_1)(\mathbf{x};\mathbf{z}) - 2\pi)^2\right] \end{bmatrix} = \begin{bmatrix} 0 \\ 2\pi \\ 0.25^2 \\ (\frac{\pi}{5})^2 \end{bmatrix} \triangleq \boldsymbol{\mu}_{\text{opt}}. \tag{30}
$$

From now on in all scientific applications (Sections 'Stomatogastric ganglion', 'Scaling EPI for stable amplification in RNNs', 'Primary visual cortex', 'Superior colliculus'), we specify how the EPI optimization was setup by specifying $f(\mathbf{x}; \mathbf{z})$, $\boldsymbol{\mu}$, and $\sigma^2$.

Unlike the models we presented in the main text, this model admits an analytical form for the mean emergent property statistics given parameter $\mathbf{z}$, since the eigenvalues can be calculated using the quadratic formula:

$$\lambda = \frac{\left(\frac{a_{1,1}+a_{2,2}}{\tau}\right) \pm \sqrt{\left(\frac{a_{1,1}+a_{2,2}}{\tau}\right)^2 + 4\left(\frac{a_{1,2}a_{2,1}-a_{1,1}a_{2,2}}{\tau}\right)}}{2}. \tag{31}$$

We study this example, because the inferred distribution is curved and multimodal, and we can compare the result of EPI to analytically derived contours of the emergent property statistics.

Despite the simple analytic form of the emergent property statistics, the EPI distribution in this example is not simply determined. Although $\mathbb{E}_{\mathbf{z}}[T(\mathbf{z})]$ is calculable directly via a closed form function, the distribution $q_{\boldsymbol{\theta}}^*(\mathbf{z} \mid \mathcal{X})$ cannot be derived directly. This fact is due to the formally hard problem of the backward mapping: finding the natural parameters $\eta$ from the mean parameters $\boldsymbol{\mu}$ of an exponential family distribution (*Wainwright and Jordan, 2008*). Instead, we used EPI to approximate this distribution (*Figure 1—figure supplement 1B*). We used a real NVP normalizing flow architecture three coupling layers and two-layer neural networks of 50 units per layer, mapped onto a support of $z_i \in [-10, 10]$. (see Section 'Deep probability distributions and normalizing flows').

Even this relatively simple system has nontrivial (although intuitively sensible) structure in the parameter distribution. To validate our method, we analytically derived the contours of the probability density from the emergent property statistics and values. In the $a_{1,1}$-$a_{2,2}$ plane, the black line at $\text{real}(\lambda_1) = \frac{a_{1,1}+a_{2,2}}{2} = 0$, dashed black line at the standard deviation $\text{real}(\lambda_1) = \frac{a_{1,1}+a_{2,2}}{2} \pm 0.25$, and the dashed gray line at twice the standard deviation $\text{real}(\lambda_1) = \frac{a_{1,1}+a_{2,2}}{2} \pm 0.5$ follow the contour of probability density of the samples (*Figure 1—figure supplement 2A*). The distribution precisely reflects the desired statistical constraints and model degeneracy in the sum of $a_{1,1}$ and $a_{2,2}$. Intuitively, the parameters equivalent with respect to emergent property statistic $\text{real}(\lambda_1)$ have similar log densities.

To explain the bimodality of the EPI distribution, we examined the imaginary component of $\lambda_1$. When $\text{real}(\lambda_1) = a_{1,1} + a_{2,2} = 0$ (which is the case on average in $\mathcal{X}$), we have

$$\text{imag}(\lambda_1) = \begin{cases} \sqrt{\frac{a_{1,1}a_{2,2}-a_{1,2}a_{2,1}}{\tau}}, & \text{if } a1,1a2,2 < a1,2a2,1 \\ 0 & \text{otherwise} \end{cases}. \tag{32}$$

In *Figure 1—figure supplement 2B*, we plot the contours of $\text{imag}(\lambda_1)$ where $a_{1,1}a_{2,2}$ is fixed to 0 at one standard deviation ($\frac{\pi}{5}$, black dashed) and two standard deviations ($\frac{2\pi}{5}$, gray dashed) from the mean of $2\pi$. This validates the curved multimodal structure of the inferred distribution learned through EPI. Subtler combinations of model and emergent property will have more complexity, further motivating the use of EPI for understanding these systems. As we expect, the distribution results in samples of two-dimensional linear systems oscillating near 1 Hz (*Figure 1—figure supplement 3*).

## EPI as variational inference

In variational inference, a posterior approximation $q_{\boldsymbol{\theta}}^*$ is chosen from within some variational family $\mathcal{Q}$ to be as close as possible to the posterior under the KL divergence criteria

$$q_{\boldsymbol{\theta}}^*(\mathbf{z}) = \underset{q_{\boldsymbol{\theta}} \in \mathcal{Q}}{\text{argmax}} \, KL(q_{\boldsymbol{\theta}}(\mathbf{z}) \mid\mid p(\mathbf{z} \mid \mathbf{x})). \tag{33}$$

This KL divergence can be written in terms of entropy of the variational approximation:

$$KL(q_{\boldsymbol{\theta}}(\mathbf{z}) \mid\mid p(\mathbf{z} \mid \mathbf{x})) = \mathbb{E}_{\mathbf{z} \sim q_{\boldsymbol{\theta}}}[\log(q_{\boldsymbol{\theta}}(\mathbf{z}))] - \mathbb{E}_{\mathbf{z} \sim q_{\boldsymbol{\theta}}}[\log(p(\mathbf{z} \mid \mathbf{x}))] \tag{34}$$

$$= -H(q_{\boldsymbol{\theta}}) - \mathbb{E}_{\mathbf{z} \sim q_{\boldsymbol{\theta}}}[\log(p(\mathbf{x} \mid \mathbf{z})) + \log(p(\mathbf{z})) - \log(p(\mathbf{x}))] \tag{35}$$

Since the marginal distribution of the data $p(\mathbf{x})$ (or 'evidence') is independent of $\boldsymbol{\theta}$, variational

inference is executed by optimizing the remaining expression. This is usually framed as maximizing the evidence lower bound (ELBO)

$$\underset{q_{\boldsymbol{\theta}} \in \mathcal{Q}}{\operatorname{argmax}} KL(q_{\boldsymbol{\theta}} \mid \mid p(\mathbf{z} \mid \mathbf{x})) = \underset{q_{\boldsymbol{\theta}} \in \mathcal{Q}}{\operatorname{argmax}} H(q_{\boldsymbol{\theta}}) + \mathbb{E}_{\mathbf{z} \sim q_{\boldsymbol{\theta}}}[\log(p(\mathbf{x} \mid \mathbf{z})) + \log(p(\mathbf{z}))]. \tag{36}$$

Now, we will show how the maximum entropy problem of EPI is equivalent to variational inference. In general, a maximum entropy problem (as in *Equation 16*) has an equivalent lagrange dual form:

$$\begin{aligned} \underset{q \in \mathcal{Q}}{\operatorname{argmax}} \; H(q(\mathbf{z})) \quad &\Longleftrightarrow \underset{q \in \mathcal{Q}}{\operatorname{argmax}} \; H(q(\mathbf{z})) + \boldsymbol{\eta}^{*\top} \mathbb{E}_{\mathbf{z} \sim q}[T(\mathbf{z})], \\ \text{s.t.} \mathbb{E}_{\mathbf{z} \sim q}[T(\mathbf{z})] &= \mathbf{0} \end{aligned} \tag{37}$$

with lagrange multipliers $\boldsymbol{\eta}^*$. By moving the lagrange multipliers within the expectation

$$q^* = \underset{q \in \mathcal{Q}}{\operatorname{argmax}} H(q(\mathbf{z})) + \mathbb{E}_{\mathbf{z} \sim q}\left[\boldsymbol{\eta}^{*\top} T(\mathbf{z})\right], \tag{38}$$

inserting a $\log \exp(\cdot)$ within the expectation,

$$q^* = \underset{q \in \mathcal{Q}}{\operatorname{argmax}} H(q(\mathbf{z})) + \mathbb{E}_{\mathbf{z} \sim q}\left[\log \exp\left(\boldsymbol{\eta}^{*\top} T(\mathbf{z})\right)\right], \tag{39}$$

and finally choosing $T(\cdot)$ to be likelihood averaged statistics as in EPI

$$q^* = \underset{q \in \mathcal{Q}}{\operatorname{argmax}} H(q(\mathbf{z})) + \mathbb{E}_{\mathbf{z} \sim q}\left[\log \exp\left(\boldsymbol{\eta}^{*\top} \begin{bmatrix} \mathbb{E}_{\mathbf{x} \sim p(\mathbf{x} \mid \mathbf{z})}[\phi_1(\mathbf{x}; \mathbf{z})] \\ \dots \\ \mathbb{E}_{\mathbf{x} \sim p(\mathbf{x} \mid \mathbf{z})}[\phi_m(\mathbf{x}; \mathbf{z})] \end{bmatrix}\right)\right], \tag{40}$$

we can compare directly to the objective used in variational inference (*Equation 36*). We see that EPI is exactly variational inference with an exponential family likelihood defined by sufficient statistics $T(\mathbf{z}) = \mathbb{E}_{\mathbf{x} \sim p(\mathbf{x} \mid \mathbf{z})}[\phi(\mathbf{x}; \mathbf{z})]$, and where the natural parameter $\boldsymbol{\eta}^*$ is predicated by the mean parameter $\boldsymbol{\mu}_{\mathrm{opt}}$. *Equation 40* implies that EPI uses an improper (or uniform) prior, which is easily changed.

This derivation of the equivalence between EPI and variational inference emphasizes why defining a statistical inference program by its mean parameterization $\boldsymbol{\mu}_{\mathrm{opt}}$ is so useful. With EPI, one can clearly define the emergent property $\mathcal{X}$ that the model of interest should produce through intuitive selection of $\boldsymbol{\mu}_{\mathrm{opt}}$ for a given $T(\mathbf{z})$. Alternatively, figuring out the correct natural parameters $\boldsymbol{\eta}^*$ for the same $T(\mathbf{z})$ that produces $\mathcal{X}$ is a formally hard problem.

## Stomatogastric ganglion

In Section 'Motivating emergent property inference of theoretical models' and 'Emergent property inference via deep generative models', we used EPI to infer conductance parameters in a model of the stomatogastric ganglion (STG) (*Gutierrez et al., 2013*). This five-neuron circuit model represents two subcircuits: that generating the pyloric rhythm (fast population) and that generating the gastric mill rhythm (slow population). The additional neuron (the IC neuron of the STG) receives inhibitory synaptic input from both subcircuits, and can couple to either rhythm dependent on modulatory conditions. There is also a parametric regime in which this neuron fires at an intermediate frequency between that of the fast and slow populations (*Gutierrez et al., 2013*), which we infer with EPI as a motivational example. This model is not to be confused with an STG subcircuit model of the pyloric rhythm (*Marder and Selverston, 1992*), which has been statistically inferred in other studies (*Prinz et al., 2004*; *Gonçalves et al., 2019*).

### STG model

We analyze how the parameters $\mathbf{z} = [g_{\mathrm{el}}, g_{\mathrm{synA}}]$ govern the emergent phenomena of intermediate hub frequency in a model of the stomatogastric ganglion (STG) (*Gutierrez et al., 2013*) shown in *Figure 1A* with activity $\mathbf{x} = [x_{\mathrm{f1}}, x_{\mathrm{f2}}, x_{\mathrm{hub}}, x_{\mathrm{s1}}, x_{\mathrm{s2}}]$, using the same hyperparameter choices as Gutierrez et al. Each neuron's membrane potential $x_\alpha(t)$ for $\alpha \in \{\mathrm{f1}, \mathrm{f2}, \mathrm{hub}, \mathrm{s1}, \mathrm{s2}\}$ is the solution of the following stochastic differential equation:

$$C_m \frac{dx_\alpha}{dt} = -\left[h_{leak}(\mathbf{x};\mathbf{z}) + h_{Ca}(\mathbf{x};\mathbf{z}) + h_K(\mathbf{x};\mathbf{z}) + h_{hyp}(\mathbf{x};\mathbf{z}) + h_{elec}(\mathbf{x};\mathbf{z}) + h_{syn}(\mathbf{x};\mathbf{z})\right] + dB. \tag{41}$$

The input current of each neuron is the sum of the leak, calcium, potassium, hyperpolarization, electrical and synaptic currents. Each current component is a function of all membrane potentials and the conductance parameters $\mathbf{z}$. Finally, we include gaussian noise $dB$ to the model of Gutierrez et al. so that the model stochastic, although this is not required by EPI.

The capacitance of the cell membrane was set to $C_m = 1nF$. Specifically, the currents are the difference in the neuron's membrane potential and that current type's reversal potential multiplied by a conductance:

$$h_{leak}(\mathbf{x};\mathbf{z}) = g_{leak}(x_\alpha - V_{leak}) \tag{42}$$

$$h_{elec}(\mathbf{x};\mathbf{z}) = g_{el}(x_\alpha^{post} - x_\alpha^{pre}) \tag{43}$$

$$h_{syn}(\mathbf{x};\mathbf{z}) = g_{syn}S_\infty^{pre}(x_\alpha^{post} - V_{syn}) \tag{44}$$

$$h_{Ca}(\mathbf{x};\mathbf{z}) = g_{Ca}M_\infty(x_\alpha - V_{Ca}) \tag{45}$$

$$h_K(\mathbf{x};\mathbf{z}) = g_K N(x_\alpha - V_K) \tag{46}$$

$$h_{hyp}(\mathbf{x};\mathbf{z}) = g_h H(x_\alpha - V_{hyp}). \tag{47}$$

The reversal potentials were set to $V_{leak} = -40mV$, $V_{Ca} = 100mV$, $V_K = -80mV$, $V_{hyp} = -20mV$, and $V_{syn} = -75mV$. The other conductance parameters were fixed to $g_{leak} = 1 \times 10^{-4} \mu S$. $g_{Ca}$, $g_K$, and $g_{hyp}$ had different values based on fast, intermediate (hub) or slow neuron. The fast conductances had values $g_{Ca} = 1.9 \times 10^{-2}$, $g_K = 3.9 \times 10^{-2}$, and $g_{hyp} = 2.5 \times 10^{-2}$. The intermediate conductances had values $g_{Ca} = 1.7 \times 10^{-2}$, $g_K = 1.9 \times 10^{-2}$, and $g_{hyp} = 8.0 \times 10^{-3}$. Finally, the slow conductances had values $g_{Ca} = 8.5 \times 10^{-3}$, $g_K = 1.5 \times 10^{-2}$, and $g_{hyp} = 1.0 \times 10^{-2}$.

Furthermore, the Calcium, Potassium, and hyperpolarization channels have time-dependent gating dynamics dependent on steady-state gating variables $M_\infty$, $N_\infty$ and $H_\infty$, respectively:

$$M_\infty = 0.5\left(1 + \tanh\left(\frac{x_\alpha - v_1}{v_2}\right)\right) \tag{48}$$

$$\frac{dN}{dt} = \lambda_N(N_\infty - N) \tag{49}$$

$$N_\infty = 0.5\left(1 + \tanh\left(\frac{x_\alpha - v_3}{v_4}\right)\right) \tag{50}$$

$$\lambda_N = \phi_N \cosh\left(\frac{x_\alpha - v_3}{2v_4}\right) \tag{51}$$

$$\frac{dH}{dt} = \frac{(H_\infty - H)}{\tau_h} \tag{52}$$

$$H_\infty = \frac{1}{1 + \exp\left(\frac{x_\alpha + v_5}{v_6}\right)} \tag{53}$$

$$\tau_h = 272 - \left( \frac{-1499}{1 + \exp\left(\frac{-x_\alpha + v_7}{v_8}\right)} \right).$$ (54)

where we set $v_1 = 0mV$, $v_2 = 20mV$, $v_3 = 0mV$, $v_4 = 15mV$, $v_5 = 78.3mV$, $v_6 = 10.5mV$, $v_7 = -42.2mV$, $v_8 = 87.3mV$, $v_9 = 5mV$, and $v_{th} = -25mV$.

Finally, there is a synaptic gating variable as well:

$$S_\infty = \frac{1}{1 + \exp\left(\frac{v_{th} - x_\alpha}{v_9}\right)}.$$ (55)

When the dynamic gating variables are considered, this is actually a 15-dimensional nonlinear dynamical system. The gaussian noise $d\mathbf{B}$ has variance $(1 \times 10^{-12})^2$ $A^2$, and introduces variability in frequency at each parameterization $\mathbf{z}$.

## Hub frequency calculation

In order to measure the frequency of the hub neuron during EPI, the STG model was simulated for $T = 300$ time steps of $dt = 25$ms. The chosen $dt$ and $T$ were the most computationally convenient choices yielding accurate frequency measurement. We used a basis of complex exponentials with frequencies from 0.0 to 1.0 Hz at 0.01 Hz resolution to measure frequency from simulated time series

$$\Phi = [0.0, 0.01, ..., 1.0]^\top.$$ (56)

To measure spiking frequency, we processed simulated membrane potentials with a relu (spike extraction) and low-pass filter with averaging window of size 20, then took the frequency with the maximum absolute value of the complex exponential basis coefficients of the processed time-series. The first 20 temporal samples of the simulation are ignored to account for initial transients.

To differentiate through the maximum frequency identification, we used a soft-argmax Let $X_\alpha \in \mathcal{C}^{|\Phi|}$ be the complex exponential filter bank dot products with the signal $x_\alpha \in \mathbb{R}^N$, where $\alpha \in \{\mathrm{f1}, \mathrm{f2}, \mathrm{hub}, \mathrm{s1}, \mathrm{s2}\}$. The soft-argmax is then calculated using temperature parameter $\beta_\psi = 100$

$$\psi_\alpha = \mathrm{softmax}(\beta_\psi |X_\alpha| \odot i),$$ (57)

where $i = [0, 1, ..., 100]$. The frequency is then calculated as

$$\omega_\alpha = 0.01 \psi_\alpha \mathrm{Hz}.$$ (58)

Intermediate hub frequency, like all other emergent properties in this work, is defined by the mean and variance of the emergent property statistics. In this case, we have one statistic, hub neuron frequency, where the mean was chosen to be 0.55 Hz,(*Equation 2*) and variance was chosen to be $0.025^2$ Hz$^2$ (*Equation 3*).

## EPI details for the STG model

EPI was run for the STG model using

$$f(\mathbf{x}; \mathbf{z}) = \omega_{\mathrm{hub}}(\mathbf{x}; \mathbf{z}),$$ (59)

$$\boldsymbol{\mu} = [0.55],$$ (60)

and

$$\sigma^2 = \left[ 0.025^2 \right]$$ (61)

(see Sections 'Maximum entropy distributions and exponential families', 'Augmented lagrangian optimization', and example in Section 'Example: 2D LDS'). Throughout optimization, the augmented lagrangian parameters η and $c$, were updated after each epoch of $i_{\max} = 5,000$ iterations (see Section

'Augmented lagrangian optimization'). The optimization converged after five epochs (*Figure 1—figure supplement 4*).

For EPI in *Figure 1E*, we used a real NVP architecture with three coupling layers and two-layer neural networks of 25 units per layer. The normalizing flow architecture mapped $\mathbf{z}_0 \sim \mathcal{N}(\mathbf{0}, I)$ to a support of $\mathbf{z} = [g_{\mathrm{el}}, g_{\mathrm{synA}}] \in [4, 8] \times [0.01, 4]$, initialized to a gaussian approximation of samples returned by a preliminary ABC search. We did not include $g_{\mathrm{synA}} < 0.01$, for numerical stability. EPI optimization was run with an augmented lagrangian coefficient of $c_0 = 10^5$, hyperparameter $\beta = 2$, a batch size $n = 400$, and we simulated one $\mathbf{x}^{(i)}$ per $\mathbf{z}^{(i)}$. The architecture converged with criteria $N_{\mathrm{test}} = 100$.

### Hessian sensitivity vectors

To quantify the second-order structure of the EPI distribution, we evaluated the Hessian of the log probability $\frac{\partial^2 \log q(\mathbf{z} \mid \mathcal{X})}{\partial \mathbf{z}\mathbf{z}^\top}$. The eigenvector of this Hessian with most negative eigenvalue is defined as the sensitivity dimension $\mathbf{v}_1$, and all subsequent eigenvectors are ordered by increasing eigenvalue. These eigenvalues are quantifications of how fast the emergent property deteriorates via the parameter combination of their associated eigenvector. In *Figure 1D*, the sensitivity dimension $v_1$ (solid) and the second eigenvector of the Hessian $v_2$ (dashed) are shown evaluated at the mode of the distribution. Since the Hessian eigenvectors have sign degeneracy, the visualized directions in 2-D parameter space were chosen to have positive $g_{\mathrm{synA}}$. The length of the arrows is inversely proportional to the square root of the absolute value of their eigenvalues $\lambda_1 = -10.7$ and $\lambda_2 = -3.22$. For the same magnitude perturbation away from the mode, intermediate hub frequency only diminishes along the sensitivity dimension $\mathbf{v}_1$ (*Figure 1E–F*).

## Scaling EPI for stable amplification in RNNs

### Rank-2 RNN model

We examined the scaling properties of EPI by learning connectivities of RNNs of increasing size that exhibit stable amplification. Rank-2 RNN connectivity was modeled as $W = UV^\top$, where $U = [\mathbf{U}_1 \quad \mathbf{U}_2] + g\chi^{(W)}$, $V = [\mathbf{V}_1 \quad \mathbf{V}_2] + g\chi^{(V)}$, and $\chi_{i,j}^{(W)}, \chi_{i,j}^{(V)} \sim \mathcal{N}(0,1)$. This RNN model has dynamics

$$\tau\dot{\mathbf{x}} = -\mathbf{x} + W\mathbf{x}. \tag{62}$$

In this analysis, we inferred connectivity parameterizations $\mathbf{z} = [\mathbf{U}_1^\top, \mathbf{U}_2^\top, \mathbf{V}_1^\top, \mathbf{V}_2^\top]^\top \in [-1, 1]^{(4N)}$ that produced stable amplification using EPI, SMC-ABC (*Sisson et al., 2007*), and SNPE (*Gonçalves et al., 2019*) (see Section Related methods).

### Stable amplification

For this RNN model to be stable, all real eigenvalues of $W$ must be less than 1: $\mathrm{real}(\lambda_1) < 1$, where $\lambda_1$ denotes the greatest real eigenvalue of $W$. For a stable RNN to amplify at least one input pattern, the symmetric connectivity $W^s = \frac{W + W^\top}{2}$ must have an eigenvalue greater than 1: $\lambda_1^s > 1$, where $\lambda^s$ is the maximum eigenvalue of $W^s$. These two conditions are necessary and sufficient for stable amplification in RNNs (*Bondanelli and Ostojic, 2020*).

### EPI details for RNNs

We defined the emergent property of stable amplification with means of these eigenvalues (0.5 and 1.5, respectively) that satisfy these conditions. To complete the emergent property definition, we chose variances ($0.25^2$) about those means such that samples rarely violate the eigenvalue constraints. To write the emergent property of *Equation 5* in terms of the EPI optimization, we have

$$f(\mathbf{x}; \mathbf{z}) = \begin{bmatrix} \mathrm{real}(\lambda_1)(\mathbf{x}; \mathbf{z}) \\ \lambda_1^s(\mathbf{x}; \mathbf{z}) \end{bmatrix}, \tag{63}$$

$$\boldsymbol{\mu} = \begin{bmatrix} 0.5 \\ 1.5 \end{bmatrix}, \tag{64}$$

and

$$\sigma^2 = \begin{bmatrix} 0.25^2 \\ 0.25^2 \end{bmatrix} \tag{65}$$

(see Sections 'Maximum entropy distributions and exponential families', 'Augmented lagrangian optimization', and example in Section 'Example: 2D LDS'). Gradients of maximum eigenvalues of Hermitian matrices like $W^s$ are available with modern automatic differentiation tools. To differentiate through the $\mathrm{real}(\lambda_1)$, we solved the following equation for eigenvalues of rank-2 matrices using the rank reduced matrix $W^r = V^\top U$

$$\lambda_\pm = \frac{\mathrm{Tr}(W^r) \pm \sqrt{\mathrm{Tr}(W^r)^2 - 4\mathrm{Det}(W^r)}}{2}. \tag{66}$$

For EPI in *Figure 2*, we used a real NVP architecture with three coupling layers of affine transformations parameterized by two-layer neural networks of 100 units per layer. The initial distribution was a standard isotropic gaussian $\mathbf{z}_0 \sim \mathcal{N}(\mathbf{0}, I)$ mapped to the support of $\mathbf{z}_i \in [-1, 1]$. We used an augmented lagrangian coefficient of $c_0 = 10^3$, a batch size $n = 200$, $\beta = 4$, and we simulated one $\mathbf{W}^{(i)}$ per $\mathbf{z}^{(i)}$. We chose to use $i_{\max} = 500$ iterations per augmented lagrangian epoch and emergent property constraint convergence was evaluated at $N_{\text{test}} = 200$ (*Figure 2B* blue line, and *Figure 2C–D* blue). It was fastest to initialize the EPI distribution on a Tesla V100 GPU, and then subsequently optimize it on a CPU with 32 cores. EPI timing measurements accounted for this initialization period.

## Methodological comparison

We compared EPI to two alternative simulation-based inference techniques, since the likelihood of these eigenvalues given $\mathbf{z}$ is not available. Approximate bayesian computation (ABC) (*Beaumont et al., 2002*) is a rejection sampling technique for obtaining sets of parameters $\mathbf{z}$ that produce activity $\mathbf{x}$ close to some observed data $\mathbf{x}_0$. Sequential Monte Carlo approximate bayesian computation (SMC-ABC) is the state-of-the-art ABC method, which leverages SMC techniques to improve sampling speed. We ran SMC-ABC with the pyABC package (*Klinger et al., 2018*) to infer RNNs with stable amplification: connectivities having eigenvalues within an $\epsilon$-defined $l-2$ distance of

$$\mathbf{x}_0 = \begin{bmatrix} \mathrm{real}(\lambda_1) \\ \lambda_1^s \end{bmatrix} = \begin{bmatrix} 0.5 \\ 1.5 \end{bmatrix}. \tag{67}$$

SMC-ABC was run with a uniform prior over $\mathbf{z} \in [-1, 1]^{(4N)}$, a population size of 1000 particles with simulations parallelized over 32 cores, and a multivariate normal transition model.

SNPE, the next approach in our comparison, is far more similar to EPI. Like EPI, SNPE treats parameters in mechanistic models with deep probability distributions, yet the two learning algorithms are categorically different. SNPE uses a two-network architecture to approximate the posterior distribution of the model conditioned on observed data $\mathbf{x}_0$. The amortizing network maps observations $\mathbf{x}_i$ to the parameters of the deep probability distribution. The weights and biases of the parameter network are optimized by sequentially augmenting the training data with additional pairs $(\mathbf{z}_i, \mathbf{x}_i)$ based on the most recent posterior approximation. This sequential procedure is important to get training data $\mathbf{z}_i$ to be closer to the true posterior, and $\mathbf{x}_i$ to be closer to the observed data. For the deep probability distribution architecture, we chose a masked autoregressive flow with affine couplings (the default choice), three transforms, 50 hidden units, and a normalizing flow mapping to the support as in EPI. This architectural choice closely tracked the size of the architecture used by EPI (*Figure 2—figure supplement 1*). As in SMC-ABC, we ran SNPE with $\mathbf{x}_0 = \mu$. All SNPE optimizations were run for a limit of 1.5 days, or until two consecutive rounds resulted in a validation log probability lower than the maximum observed for that random seed. It was always faster to run SNPE on a CPU with 32 cores rather than on a Tesla V100 GPU.

To compare the efficiency of these algorithms for inferring RNN connectivity distributions producing stable amplification, we develop a convergence criteria that can be used across methods. While EPI has its own hypothesis testing convergence criteria for the emergent property, it would not make sense to use this criteria on SNPE and SMC-ABC which do not constrain the means and variances of their predictions. Instead, we consider EPI and SNPE to have converged after completing its

most recent optimization epoch (EPI) or round (SNPE) in which the distance $|\mathbb{E}_{\mathbf{z},\mathbf{x}}[f(\mathbf{x};\mathbf{z})] - \boldsymbol{\mu}|_2$ is less than 0.5. We consider SMC-ABC to have converged once the population produces samples within the $\epsilon = 0.5$ ball ensuring stable amplification.

When assessing the scalability of SNPE, it is important to check that alternative hyperparamterizations could not yield better performance. Key hyperparameters of the SNPE optimization are the number of simulations per round $n_{\text{round}}$, the number of atoms used in the atomic proposals of the SNPE-C algorithm (**Greenberg, 2019**), and the batch size $n$. To match EPI, we used a batch size of $n = 200$ for $N <= 25$, however we found $n = 1,000$ to be helpful for SNPE in higher dimensions. While $n_{\text{round}} = 1,000$ yielded SNPE convergence for $N < = 25$, we found that a substantial increase to $n_{\text{round}} = 25,000$ yielded more consistent convergence at $N = 50$ (**Figure 2—figure supplement 2A**). By increasing $n_{\text{round}}$, we also necessarily increase the duration of each round. At $N = 100$, we tried two hyperparameter modifications. As suggested in **Greenberg, 2019**, we increased $n_{\text{atom}}$ by an order of magnitude to improve gradient quality, but this had little effect on the optimization (much overlap between same random seeds) (**Figure 2—figure supplement 2B**). Finally, we increased $n_{\text{round}}$ by an order of magnitude, which yielded convergence in one case, but no others. We found no way to improve the convergence rate of SNPE without making more aggressive hyperparameter choices requiring high numbers of simulations. In **Figure 2C–D**, we show samples from the random seed resulting in emergent property convergence at greatest entropy (EPI), the random seed resulting in greatest validation log probability (SNPE), and the result of all converged random seeds (SMC).

## Effect of RNN parameters on EPI and SNPE inferred distributions

To clarify the difference in objectives of EPI and SNPE, we show their results on RNN models with different numbers of neurons $N$ and random strength $g$. The parameters inferred by EPI consistently produces the same mean and variance of $\text{real}(\lambda_1)$ and $\lambda_1^s$, while those inferred by SNPE change according to the model definition (**Figure 2—figure supplement 3A**). For $N = 2$ and $g = 0.01$, the SNPE posterior has greater concentration in eigenvalues around $\mathbf{x}_0$ than at $g = 0.1$, where the model has greater randomness (**Figure 2—figure supplement 3B** top, orange). At both levels of $g$ when $N = 2$, the posterior of SNPE has lower entropy than EPI at convergence (**Figure 2—figure supplement 3B** top). However at $N = 10$, SNPE results in a predictive distribution of more widely dispersed eigenvalues (**Figure 2—figure supplement 3A** bottom), and an inferred posterior with greater entropy than EPI (**Figure 2—figure supplement 3B** bottom). We highlight these differences not to focus on an insightful trend, but to emphasize that these methods optimize different objectives with different implications.

Note that SNPE converges when it's validation log probability has saturated after several rounds of optimization (**Figure 2—figure supplement 3C**), and that EPI converges after several epochs of its own optimization to enforce the emergent property constraints (**Figure 2—figure supplement 3D** blue). Importantly, as SNPE optimizes its posterior approximation, the predictive means change, and at convergence may be different than $\mathbf{x}_0$ (**Figure 2—figure supplement 3D** orange, left). It is sensible to assume that predictions of a well-approximated SNPE posterior should closely reflect the data on average (especially given a uniform prior and a low degree of stochasticity); however, this is not a given. Furthermore, no aspect of the SNPE optimization controls the variance of the predictions (**Figure 2—figure supplement 3D** orange, right).

## Primary visual cortex
### V1 model
E-I circuit models, rely on the assumption that inhibition can be studied as an indivisible unit, despite ample experimental evidence showing that inhibition is instead composed of distinct elements (**Tremblay et al., 2016**). In particular three types of genetically identified inhibitory cell-types – parvalbumin (P), somatostatin (S), VIP (V) – compose 80% of GABAergic interneurons in V1 (**Markram et al., 2004**; **Rudy et al., 2011**; **Tremblay et al., 2016**), and follow specific connectivity patterns (**Figure 3A**; **Pfeffer et al., 2013**), which lead to cell-type-specific computations (**Mossing et al., 2021**; **Palmigiano et al., 2020**). Currently, how the subdivision of inhibitory cell-types, shapes correlated variability by reconfiguring recurrent network dynamics is not understood.

In the stochastic stabilized supralinear network (*Hennequin et al., 2018*), population rate responses $\mathbf{x}$ to mean input $\mathbf{h}$, recurrent input $W\mathbf{x}$ and slow noise $\epsilon$ are governed by

$$\tau \frac{d\mathbf{x}}{dt} = -\mathbf{x} + \phi(W\mathbf{x} + \mathbf{h} + \epsilon), \tag{68}$$

where $\phi(\cdot) = [\cdot]_+^2$, and the noise is an Ornstein-Uhlenbeck process $\epsilon \sim OU(\tau_{\mathrm{noise}}, \sigma)$

$$\tau_{\mathrm{noise}} d\epsilon_\alpha = -\epsilon_\alpha dt + \sqrt{2\tau_{\mathrm{noise}}}\tilde{\sigma}_\alpha dB \tag{69}$$

with $\tau_{\mathrm{noise}} = 5\mathrm{ms} > \tau = 1\mathrm{ms}$. The noisy process is parameterized as

$$\tilde{\sigma}_\alpha = \sigma_\alpha \sqrt{1 + \frac{\tau}{\tau_{\mathrm{noise}}}}, \tag{70}$$

so that $\sigma$ parameterizes the variance of the noisy input in the absence of recurrent connectivity ($W = \boldsymbol{0}$). As contrast $c \in [0, 1]$ increases, input to the E- and P-populations increases relative to a baseline input $\mathbf{h} = \mathbf{h}_b + c\mathbf{h}_c$. Connectivity ($W_{\mathrm{fit}}$) and input ($\mathbf{h}_{b,\mathrm{fit}}$ and $\mathbf{h}_{c,\mathrm{fit}}$) parameters were fit using the deterministic V1 circuit model (*Palmigiano et al., 2020*)

$$W_{\mathrm{fit}} = \begin{bmatrix} W_{EE} & W_{EP} & W_{ES} & W_{EV} \\ W_{PE} & W_{PP} & W_{PS} & W_{PV} \\ W_{SE} & W_{SP} & W_{SS} & W_{SV} \\ W_{VE} & W_{VP} & W_{VS} & W_{VV} \end{bmatrix} = \begin{bmatrix} 2.18 & -1.19 & -.594 & -.229 \\ 1.66 & -.651 & -.680 & -.242 \\ .895 & -5.22 \times 10^{-3} & -1.51 \times 10^{-4} & -.761 \\ 3.34 & -2.31 & -.254 & -2.52 \times 10^{-4} \end{bmatrix}, \tag{71}$$

$$\mathbf{h}_{b,\mathrm{fit}} = \begin{bmatrix} .416 \\ .429 \\ .491 \\ .486 \end{bmatrix}, \tag{72}$$

and

$$\mathbf{h}_{c,\mathrm{fit}} = \begin{bmatrix} .359 \\ .403 \\ 0 \\ 0 \end{bmatrix}. \tag{73}$$

To obtain rates on a realistic scale (100-fold greater), we map these fitted parameters to an equivalence class

$$W = \begin{bmatrix} W_{EE} & W_{EP} & W_{ES} & W_{EV} \\ W_{PE} & W_{PP} & W_{PS} & W_{PV} \\ W_{SE} & W_{SP} & W_{SS} & W_{SV} \\ W_{VE} & W_{VP} & W_{VS} & W_{VV} \end{bmatrix} = \begin{bmatrix} .218 & -.119 & -.0594 & -.0229 \\ .166 & -.0651 & -.068 & -.0242 \\ .0895 & -5.22 \times 10^{-4} & -1.51 \times 10^{-5} & -.0761 \\ .334 & -.231 & -.0254 & -2.52 \times 10^{-5} \end{bmatrix}, \tag{74}$$

$$\mathbf{h}_b = \begin{bmatrix} h_{b,E} \\ h_{b,P} \\ h_{b,S} \\ h_{b,V} \end{bmatrix} = \begin{bmatrix} 4.16 \\ 4.29 \\ 4.91 \\ 4.86 \end{bmatrix}, \tag{75}$$

and

$$\mathbf{h}_c = \begin{bmatrix} h_{c,E} \\ h_{c,P} \\ h_{c,S} \\ h_{c,V} \end{bmatrix} = \begin{bmatrix} 3.59 \\ 4.03 \\ 0 \\ 0 \end{bmatrix}. \tag{76}$$

Circuit responses are simulated using $T = 200$ time steps at $dt = 0.5\mathrm{ms}$ from an initial condition

drawn from $\mathbf{x}(0) \sim U[10\text{Hz}, 25\text{Hz}]$. Standard deviation of the E-population $s_E(\mathbf{x}; \mathbf{z})$ is calculated as the square root of the temporal variance from $t_{ss} = 75\text{ms}$ to $Tdt = 100\text{ms}$

$$s_E(\mathbf{x}; \mathbf{z}) = \sqrt{\mathbb{E}_{t > t_{ss}}\left[\left(x_E(t) - \mathbb{E}_{t > t_{ss}}[x_E(t)]\right)^2\right]}. \tag{77}$$

## EPI details for the V1 model

To write the emergent properties of *Equation 7* in terms of the EPI optimization, we have

$$f(\mathbf{x}; \mathbf{z}) = s_E(\mathbf{x}; \mathbf{z}), \tag{78}$$

$$\boldsymbol{\mu} = [5] \tag{79}$$

(or $\boldsymbol{\mu} = [10]$), and

$$\sigma^2 = [1^2] \tag{80}$$

(see Sections 'Maximum entropy distributions and exponential families', 'Augmented lagrangian optimization', and example in Section 'Example: 2D LDS').

For EPI in *Figure 3D–E* and *Figure 3—figure supplement 1*, we used a real NVP architecture with three coupling layers and two-layer neural networks of 50 units per layer. The normalizing flow architecture mapped $z_0 \sim \mathcal{N}(\mathbf{0}, I)$ to a support of $\mathbf{z} = [\sigma_E, \sigma_P, \sigma_S, \sigma_V] \in [0.0, 0.5]^4$. EPI optimization was run using three different random seeds for architecture initialization $\boldsymbol{\theta}$ with an augmented lagrangian coefficient of $c_0 = 10^{-1}$, $\beta = 2$, a batch size $n = 100$, and simulated 100 trials to calculate average $s_E(\mathbf{x}; \mathbf{z})$ for each $\mathbf{z}^{(i)}$. We used $i_{\max} = 2,000$ iterations per epoch. The distributions shown are those of the architectures converging with criteria $N_{\text{test}} = 100$ at greatest entropy across three random seeds. Optimization details are shown in *Figure 3—figure supplement 2*. The sums of squares of each pair of parameters are shown for each EPI distribution in *Figure 3—figure supplement 3*. The plots are histograms of 500 samples from each EPI distribution from which the significance p-values of Section 'EPI reveals how recurrence with multiple inhibitory subtypes governs excitatory variability in a V1 model' are determined.

## Sensitivity analyses

In *Figure 3E*, we visualize the modes of $q_{\boldsymbol{\theta}}(\mathbf{z} \mid \mathcal{X})$ throughout the $\sigma_E$-$\sigma_P$ marginal. At each local mode $\mathbf{z}^*(\sigma_P)$, where $\sigma_P$ is fixed, we calculated the Hessian and visualized the sensitivity dimension in the direction of positive $\sigma_E$.

## Testing for the paradoxical effect

The paradoxical effect occurs when a populations steady state rate is decreased (or increased) when an increase (decrease) in current is applied to that population (*Tsodyks et al., 1997*). To see which, if any, populations exhibited a paradoxical effect, we examined responses to changes in input to individual neuron-type populations, where the initial condition was the steady state response to $\mathbf{h}$ (*Figure 3—figure supplement 4*). Input magnitudes were chosen so that the effect is salient (0.002 for E and P, but 0.02 for S and V). Only the P-population exhibited the paradoxical effect at this connectivity $W$ and input $\mathbf{h}$.

## Primary visual cortex: Mathematical intuition and challenges

We write the original *Equations 68 and 69* in the following way:

$$\begin{aligned} dx &= \frac{1}{\tau}(-x + f(Wx + h + \epsilon))dt \\ d\epsilon &= -\frac{dt}{\tau_{\text{noise}}}\epsilon + \frac{\sqrt{2}}{\sqrt{\tau_{\text{noise}}}}\Sigma_\epsilon dW \end{aligned} \tag{81}$$

where in this paper we chose $\Sigma_\epsilon$, the covariance of the noise to be

$$\Sigma_\epsilon = \tau_{\text{noise}} \begin{bmatrix} \tilde{\sigma}_E & 0 & 0 & 0 \\ 0 & \tilde{\sigma}_P & 0 & 0 \\ 0 & 0 & \tilde{\sigma}_S & 0 \\ 0 & 0 & 0 & \tilde{\sigma}_V \end{bmatrix} \tag{82}$$

where $\tilde{\sigma}_\alpha$ is the reparameterized standard deviation of the noise for population $\alpha$ from *Equation 70*.

We are interested in computing the covariance of the activity. For that, first we define $v = \omega x + h + \epsilon$, the total input to each cell type, and the matrix $S$, the negative Jacobian $S = I - \omega f'(v)$. Then, *Equation 81* can be written as an 8-dimensional system. Linearizing around the fixed point of the system without fluctuations, we find the equations that describe the fluctuations of the input to each cell type:

$$d\begin{pmatrix} \delta v \\ \epsilon \end{pmatrix} = -\begin{pmatrix} S & -\frac{\tau_{\text{noise}} - \tau}{\tau \tau_{\text{noise}}} I \\ 0 & \frac{1}{\tau_{\text{noise}}} I \end{pmatrix}\begin{pmatrix} \delta v \\ \epsilon \end{pmatrix} dt + \begin{pmatrix} 0 & \frac{\sqrt{2}}{\sqrt{\tau_{\text{noise}}}}\Sigma_\epsilon \\ 0 & \frac{\sqrt{2}}{\sqrt{\tau_{\text{noise}}}}\Sigma_\epsilon \end{pmatrix} d\mathbf{W} \tag{83}$$

where $d\mathbf{W}$ is a vector with the private noise of each variable. The $d\mathbf{W}$ term is multiplied by a non-diagonal matrix, because the noise that the voltage receives is the exact same as the one that comes from the OU process and not another process. The covariance of the inputs $\Lambda_v = \langle \delta v \delta v^T \rangle$ can be found as the solution the following Lyapunov equation (*Hennequin et al., 2018*; *Gardiner, 2009*):

$$\begin{pmatrix} S & -\frac{\tau_{\text{noise}} - \tau}{\tau \tau_{\text{noise}}} I \\ 0 & \frac{1}{\tau_{\text{noise}}} I \end{pmatrix}\begin{pmatrix} \Lambda_v & \Lambda_c \\ \Lambda_c^T & \Lambda_\epsilon \end{pmatrix} + \begin{pmatrix} \Lambda_v & \Lambda_c \\ \Lambda_c^T & \Lambda_\epsilon \end{pmatrix}\begin{pmatrix} S^T & 0 \\ -\frac{\tau_{\text{noise}} - \tau}{\tau \tau_{\text{noise}}} I & \frac{1}{\tau_{\text{noise}}} I \end{pmatrix} = \begin{pmatrix} \frac{2}{\tau_{\text{noise}}}\Lambda_\epsilon & \frac{2}{\tau_{\text{noise}}}\Lambda_\epsilon \\ \frac{2}{\tau_{\text{noise}}}\Lambda_\epsilon & \frac{2}{\tau_{\text{noise}}}\Lambda_\epsilon \end{pmatrix} \tag{84}$$

Where $\Lambda_c = \langle \delta \mathbf{v} \delta \epsilon^T \rangle$ can be eliminated by solving this block matrix multiplication:

$$S\Lambda_v + \Lambda_v S^T = \frac{2\Lambda_\epsilon}{\tau_{\text{noise}}} + \frac{\tau_{\text{noise}}^2 - \tau^2}{(\tau \tau_{\text{noise}})^2}\left( (\frac{1}{\tau_{\text{noise}}} I + S)^{-1}\Lambda_\epsilon + \Lambda_\epsilon(\frac{1}{\tau_{\text{noise}}} I + S^T)^{-1} \right) \tag{85}$$

The equation above is another Lyapunov Equation, now in 4 dimensions. In the simplest case in which $\tau_{\text{noise}} = \tau$, the voltage is directly driven by white noise, and $\Lambda_v$ can be expressed in powers of $S$ and $S^T$. Because $S$ satisfies its own polynomial equation (Cayley Hamilton theorem), there will be four coefficients for the expansion of $S$ and four for $S^T$, resulting in 16 coefficients that define $\Lambda_v$ for a given $S$. Due to symmetry arguments (*Gardiner, 2009*), in this case the diagonal elements of the covariance matrix of the voltage will have the form:

$$\Lambda_{v_{ii}} = \sum_{i=\{E,P,S,V\}} g_i(S)\sigma_{ii}^2 \tag{86}$$

These coefficients $g_i(S)$ are complicated functions of the Jacobian of the system. Although expressions for these coefficients can be found explicitly, only numerical evaluation of those expressions determine which components of the noisy input are going to strongly influence the variability of excitatory population. Showing the generality of this dependence in more complicated noise scenarios (e.g. $\tau_{\text{noise}} > \tau$ as in Section 'EPI reveals how recurrence with multiple inhibitory subtypes governs excitatory variability in a V1 model'), is the focus of current research.

## Superior colliculus
### SC model
The ability to switch between two separate tasks throughout randomly interleaved trials, or 'rapid task switching,' has been studied in rats, and midbrain superior colliculus (SC) has been show to play an important in this computation (*Duan et al., 2015*). Neural recordings in SC exhibited two populations of neurons that simultaneously represented both task context (Pro or Anti) and motor response (contralateral or ipsilateral to the recorded side), which led to the distinction of two functional classes: the Pro/Contra and Anti/Ipsi neurons (*Duan et al., 2021*). Given this evidence, Duan et al. proposed a model with four functionally-defined neuron-type populations: two in each hemisphere corresponding to the Pro/Contra and Anti/Ipsi populations. We study how the connectivity of this neural circuit governs rapid task switching ability.

The four populations of this model are denoted as left Pro (LP), left Anti (LA), right Pro (RP) and right Anti (RA). Each unit has an activity ($x_\alpha$) and internal variable ($u_\alpha$) related by

$$x_\alpha = \phi(u_\alpha) = \left(\frac{1}{2}\tanh\left(\frac{u_\alpha - a}{b}\right) + \frac{1}{2}\right),\tag{87}$$

where $\alpha \in \{LP, LA, RA, RP\}$, $a = 0.05$ and $b = 0.5$ control the position and shape of the nonlinearity. We order the neural populations of $x$ and $u$ in the following manner

$$\mathbf{x} = \begin{bmatrix} x_{LP} \\ x_{LA} \\ x_{RP} \\ x_{RA} \end{bmatrix} \qquad \mathbf{u} = \begin{bmatrix} u_{LP} \\ u_{LA} \\ u_{RP} \\ u_{RA} \end{bmatrix},\tag{88}$$

which evolve according to

$$\tau \frac{d\mathbf{u}}{dt} = -\mathbf{u} + W\mathbf{x} + \mathbf{h} + d\mathbf{B}.\tag{89}$$

with time constant $\tau = 0.09s$, step size 24 ms and Gaussian noise $d\mathbf{B}$ of variance $0.2^2$. These hyperparameter values are motivated by modeling choices and results from *Duan et al., 2021*.

The weight matrix has four parameters for self $sW$, vertical $vW$, horizontal $hW$, and diagonal $dW$ connections:

$$W = \begin{bmatrix} sW & vW & hW & dW \\ vW & sW & dW & hW \\ hW & dW & sW & vW \\ dW & hW & vW & sW \end{bmatrix}.\tag{90}$$

We study the role of parameters $\mathbf{z} = [sW, vW, hW, dW]^\top$ in rapid task switching.

The circuit receives four different inputs throughout each trial, which has a total length of 1.8 s.

$$\mathbf{h} = \mathbf{h}_{\text{constant}} + \mathbf{h}_{\text{P,bias}} + \mathbf{h}_{\text{rule}} + \mathbf{h}_{\text{choice-period}} + \mathbf{h}_{\text{light}}.\tag{91}$$

There is a constant input to every population,

$$\mathbf{h}_{\text{constant}} = I_{\text{constant}}[1, 1, 1, 1]\top,\tag{92}$$

a bias to the Pro populations

$$\mathbf{h}_{\text{P,bias}} = I_{\text{P,bias}}[1, 0, 1, 0]\top,\tag{93}$$

rule-based input depending on the condition

$$\mathbf{h}_{\text{P,rule}}(t) = \begin{cases} I_{\text{P,rule}}[1, 0, 1, 0]^\top, & \text{if } t \leq 1.2s \\ 0, & \text{otherwise} \end{cases}\tag{94}$$

$$\mathbf{h}_{\text{A,rule}}(t) = \begin{cases} I_{\text{A,rule}}[0, 1, 0, 1]^\top, & \text{if } t \leq 1.2s \\ 0, & \text{otherwise} \end{cases},\tag{95}$$

a choice-period input

$$\mathbf{h}_{\text{choice}}(t) = \begin{cases} I_{\text{choice}}[1, 1, 1, 1]^\top, & \text{if } t > 1.2s \\ 0, & \text{otherwise} \end{cases},\tag{96}$$

and an input to the right or left-side depending on where the light stimulus is delivered

$$\mathbf{h}_{\text{light}}(t) = \begin{cases} I_{\text{light}}[1,1,0,0]^\top, & \text{if } 1.2s < t < 1.5s \text{ and Left} \\ I_{\text{light}}[0,0,1,1]^\top, & \text{if } 1.2s < t < 1.5s \text{ and Right} \\ 0, & \text{otherwise} \end{cases} \tag{97}$$

The input parameterization was fixed to $I_{\text{constant}} = 0.75$, $I_{\text{P,bias}} = 0.5$, $I_{\text{P,rule}} = 0.6$, $I_{\text{A,rule}} = 0.6$, $I_{\text{choice}} = 0.25$, and $I_{\text{light}} = 0.5$.

## Task accuracy calculation

The accuracies of the Pro- and Anti-tasks are calculated as

$$p_P(\mathbf{x}; \mathbf{z}) = \mathbb{E}_{\mathbf{x} \sim p(\mathbf{x} \mid \mathbf{z})}[d_P(\mathbf{x}; \mathbf{z})] \tag{98}$$

and

$$p_A(\mathbf{x}; \mathbf{z}) = \mathbb{E}_{\mathbf{x} \sim p(\mathbf{x} \mid \mathbf{z})}[d_A(\mathbf{x}; \mathbf{z})] \tag{99}$$

where $d_P(\mathbf{x}; \mathbf{z})$ and $d_A(\mathbf{x}; \mathbf{z})$ calculate the decision made in each trial (approximately 1 for correct and 0 for incorrect choices). Specifically,

$$d_P(\mathbf{x}; \mathbf{z}) = \Theta[x_{LP}(t = 1.8s) - x_{RP}(t = 1.8s)] \tag{100}$$

in Pro-trials where the stimulus is on the left side, and $\Theta$ approximates the Heaviside step function. Similarly,

$$d_A(\mathbf{x}; \mathbf{z}) = \Theta[x_{RP}(t = 1.8s) - x_{LP}(t = 1.8s)] \tag{101}$$

in Anti-trials where the stimulus was on the left side. Our accuracy calculation only considers one stimulus presentation (Left), since the model is left-right symmetric. The accuracy is averaged over 200 independent trials, and the Heaviside step function is approximated as

$$\Theta(\mathbf{x}) = \text{sigmoid}(\beta_\Theta \mathbf{x}), \tag{102}$$

where $\beta_\Theta = 100$.

## EPI details for the SC model

To write the emergent properties of *Equation 9* in terms of the EPI optimization, we have

$$f(\mathbf{x}; \mathbf{z}) = \begin{bmatrix} d_P(\mathbf{x}; \mathbf{z}) \\ d_A(\mathbf{x}; \mathbf{z}) \end{bmatrix} \tag{103}$$

$$\boldsymbol{\mu} = \begin{bmatrix} .75 \\ .75 \end{bmatrix}, \tag{104}$$

and

$$\sigma^2 = \begin{bmatrix} .075^2 \\ .075^2 \end{bmatrix} \tag{105}$$

(see Sections 'Maximum entropy distributions and exponential families', 'Augmented lagrangian optimization', and example in Section 'Example: 2D LDS').

Throughout optimization, the augmented lagrangian parameters $\eta$ and $c$, were updated after each epoch of $i_{\max} = 2,000$ iterations (see Section 'Augmented lagrangian optimization'). The optimization converged after ten epochs (*Figure 4—figure supplement 4*).

For EPI in *Figure 4C*, we used a real NVP architecture with three coupling layers of affine transformations parameterized by two-layer neural networks of 50 units per layer. The initial distribution was a standard isotropic gaussian $\mathbf{z}_0 \sim \mathcal{N}(\mathbf{0}, I)$ mapped to a support of $\mathbf{z}_i \in [-5, 5]$. We used an augmented lagrangian coefficient of $c_0 = 10^2$, a batch size $n = 100$, and $\beta = 2$. The distribution was the greatest EPI distribution to converge across five random seeds with criteria $N_{\text{test}} = 25$.

The bend in the EPI distribution is not a spurious result of the EPI optimization. The structure discovered by EPI matches the shape of the set of points returned from brute-force random sampling (*Figure 4—figure supplement 5A*) These connectivities were sampled from a uniform distribution over the range of each connectivity parameter, and all parameters producing accuracy in each task within the range of 60% to 90% were kept. This set of connectivities will not match the distribution of EPI exactly, since it is not conditioned on the emergent property. For example, the parameter set returned by the brute-force search is biased toward lower accuracies (*Figure 4—figure supplement 5B*).

## Mode identification with EPI

We found one mode of the EPI distribution for fixed values of $sW$ from 1 to $-1$ in steps of 0.25. To begin, we chose an initial parameter value from 500 parameter samples $\mathbf{z} \sim q_\theta(\mathbf{z} \mid \mathcal{X})$ that had closest $sW$ value to 1. We then optimized this estimate of the mode (for fixed $sW$) using probability gradients of the deep probability distribution for 500 steps of gradient ascent with a learning rate of $5 \times 10^{-3}$. The next mode (at $sW = 0.75$) was found using the previous mode as the initialization. This and all subsequent optimizations used 200 steps of gradient ascent with a learning rate of $1 \times 10^{-3}$, except at $sW = -1$ where a learning rate of $5 \times 10^{-4}$ was used. During all mode identification optimizations, the learning rate was reduced by half (decay = 0.5) after every 100 iterations.

## Sample grouping by mode

For the analyses in *Figure 5C* and *Figure 5—figure supplement 1*, we obtained parameters for each step along the continuum between regimes 1 and 2 by sampling from the EPI distribution. Each sample was assigned to the closest mode $\mathbf{z}^*(sW)$. Sampling continued until 500 samples were assigned to each mode, which took 2.67 s (5.34 ms/sample-per-mode). It took 9.59 min to obtain just five samples for each mode with brute force sampling requiring accuracies between 60% and 90% in each task (115 s/sample-per-mode). This corresponds to a sampling speed increase of roughly 21,500 once the EPI distribution has been learned.

## Sensitivity analysis

At each mode, we measure the sensitivity dimension (that of most negative eigenvalue in the Hessian of the EPI distribution) $\mathbf{v}_1(\mathbf{z}^*)$. To resolve sign degeneracy in eigenvectors, we chose $\mathbf{v}_1(\mathbf{z}^*)$ to have negative element in $hW$. This tells us what parameter combination rapid task switching is most sensitive to at this parameter choice in the regime.

## Connectivity eigendecomposition and processing modes

To understand the connectivity mechanisms governing task accuracy, we took the eigendecomposition of the connectivity matrices $W = Q\Lambda Q^{-1}$, which results in the same eigenmodes $\mathbf{q}_i$ for all $W$ parameterized by $\mathbf{z}$ (*Figure 4—figure supplement 3A*). These eigenvectors are always the same, because the connectivity matrix is symmetric and the model also assumes symmetry across hemispheres, but the eigenvalues of connectivity (or degree of eigenmode amplification) change with $\mathbf{z}$. These basis vectors have intuitive roles in processing for this task, and are accordingly named the *all* eigenmode - all neurons co-fluctuate, *side* eigenmode - one side dominates the other, *task* eigenmode - the Pro or Anti-populations dominate the other, and *diag* mode - Pro- and Anti-populations of opposite hemispheres dominate the opposite pair. Due to the parametric structure of the connectivity matrix, the parameters $\mathbf{z}$ are a linear function of the eigenvalues $\lambda = [\lambda_{\text{all}}, \lambda_{\text{side}}, \lambda_{\text{task}} \lambda_{\text{diag}}]^\top$ associated with these eigenmodes.

$$\mathbf{z} = A\lambda \tag{106}$$

$$A = \frac{1}{4} \begin{bmatrix} 1 & 1 & 1 & 1 \\ 1 & -1 & -1 & 1 \\ 1 & 1 & -1 & -1 \\ 1 & -1 & 1 & -1 \end{bmatrix}. \tag{107}$$

We are interested in the effect of raising or lowering the amplification of each eigenmode in the

connectivity matrix by perturbing individual eigenvalues $\lambda$. To test this, we calculate the unit vector of changes in the connectivity $\mathbf{z}$ that result from a change in the associated eigenvalues

$$\mathbf{v}_a = \frac{\frac{\partial \mathbf{z}}{\partial \lambda_a}}{\left| \frac{\partial \mathbf{z}}{\partial \lambda_a} \right|_2}, \tag{108}$$

where

$$\frac{\partial \mathbf{z}}{\partial \lambda_a} = A\mathbf{e}_a, \tag{109}$$

and for example $\mathbf{e}_{\mathrm{all}} = [1,0,0,0]^\top$. So $\mathbf{v}_a$ is the normalized column of A corresponding to eigenmode $a$. The parameter dimension $\mathbf{v}_a$ ($a \in \{\mathrm{all}, \mathrm{side}, \mathrm{task}, \mathrm{and\,diag}\}$) that increases the eigenvalue of connectivity $\lambda_a$ is $\mathbf{z}$-invariant (*Equation 109*) and $\mathbf{v}_a \perp \mathbf{v}_{b\neq a}$. By perturbing $\mathbf{z}$ along $\mathbf{v}_a$, we can examine how model function changes by directly modulating the connectivity amplification of specific eigenmodes, which have interpretable roles in processing in each task.

## Modeling optogenetic silencing

We tested whether the inferred SC model connectivities could reproduce experimental effects of optogenetic inactivation in rats (*Duan et al., 2021*). During periods of simulated optogenetic inactivation, activity was decreased proportional to the optogenetic strength $\gamma \in [0,1]$

$$x_\alpha = (1 - \gamma)\phi(u_\alpha). \tag{110}$$

Delay period inactivation was from $0.8 < t < 1.2$.

## Acknowledgements

This work was funded by NSF Graduate Research Fellowship, DGE-1644869, McKnight Endowment Fund, NIH NINDS 5R01NS100066, Simons Foundation 542963, NSF NeuroNex Award, DBI-1707398, The Gatsby Charitable Foundation, Simons Collaboration on the Global Brain Postdoctoral Fellowship, Chinese Postdoctoral Science Foundation, and International Exchange Program Fellowship. We also acknowledge the Marine Biological Laboratory Methods in Computational Neuroscience Course, where this work was discussed and explored in its early stages. Helpful conversations were had with Larry Abbott, Stephen Baccus, James Fitzgerald, Gabrielle Gutierrez, Francesca Mastrogiuseppe, Srdjan Ostojic, Liam Paninski, and Dhruva Raman.

## Additional information

### Funding

| Funder | Grant reference number | Author |
| --- | --- | --- |
| National Science Foundation | DGE-1644869 | Sean R Bittner |
| NINDS | 5R01NS100066 | Sean R Bittner<br>John Cunningham |
| McKnight Endowment Fund for Neuroscience | Faculty Award | John Cunningham |
| Gatsby Charitable Foundation | GAT3708 | Sean R Bittner<br>Agostina Palmigiano<br>Kenneth D Miller<br>John Cunningham |
| Simons Foundation | 542963 | Sean R Bittner<br>John Cunningham |
| National Science Foundation | 1707398 | Sean R Bittner<br>Agostina Palmigiano<br>Kenneth D Miller<br>John Cunningham |

| | | |
|---|---|---|
| NIH | 5U19NS107613 | Agostina Palmigiano<br>Kenneth D Miller |
| Simons Foundation | Postdoctoral Fellowship | Chunyu A Duan |
| NIH | U01-NS108683 | Agostina Palmigiano<br>Kenneth D Miller |
| NIH | R01-EY029999 | Agostina Palmigiano |
| Grossman Charitable Foundation | | Sean R Bittner<br>John Cunningham |
| Simons Foundation | 553017 | Kenneth D Miller |
| Howard Hughes Medical Institute | | Carlos D Brody |

The funders had no role in study design, data collection and interpretation, or the decision to submit the work for publication.

### Author contributions

Sean R Bittner, Conceptualization, Software, Formal analysis, Funding acquisition, Validation, Investigation, Visualization, Methodology, Writing - original draft, Writing - review and editing; Agostina Palmigiano, Alex T Piet, Chunyu A Duan, Carlos D Brody, Kenneth D Miller, Conceptualization, Methodology, Writing - review and editing; John Cunningham, Conceptualization, Resources, Supervision, Funding acquisition, Methodology, Writing - original draft, Project administration, Writing - review and editing

### Author ORCIDs

Sean R Bittner https://orcid.org/0000-0002-6773-5402
Alex T Piet https://orcid.org/0000-0002-6529-1414
Chunyu A Duan http://orcid.org/0000-0002-3095-8653
Carlos D Brody http://orcid.org/0000-0002-4201-561X

### Decision letter and Author response

Decision letter https://doi.org/10.7554/eLife.56265.sa1
Author response https://doi.org/10.7554/eLife.56265.sa2

# Additional files

### Supplementary files

• Transparent reporting form

### Data availability

All software and scripts for emergent property inference and the analyses in this manuscript can be found on the Cunningham Lab github at this link: https://github.com/cunningham-lab/epi (copy archived at https://archive.softwareheritage.org/swh:1:rev:38febae7035ca921334a616b0f396b3767bf18d4).

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
