## [Decision Letter]

**Acceptance summary:**

Progress in understanding cellular and circuit mechanisms that underpin neural function depend on us being able to reconstitute key computations and high level functions from models that consist of simpler, biologically motivated building blocks. This requires choosing parameters in a model, yet even in the simplest models there tends to be a degenerate relationship between model parameters and its overall emergent function. Bittner and colleagues present a computationally tractable method for inferring parameter distributions that are consistent with an emergent property of interest and demonstrate its use in a variety of well-known circuit models, and show that the method compares favourably with state of the art sampling and parameter identification methods.

**Decision letter after peer review:**

Thank you for submitting your article "Interrogating theoretical models of neural computation with deep inference" for consideration by *eLife*. Your article has been reviewed by 3 peer reviewers, and the evaluation has been overseen by a Reviewing Editor and John Huguenard as the Senior Editor. The following individual involved in review of your submission has agreed to reveal their identity: Mark S. Goldman (Reviewer #3).

The reviewers have discussed the reviews with one another and the Reviewing Editor has drafted this decision to help you prepare a revised submission.

The editors have judged that your manuscript is of potential interest, but as described below significant additional work is required before it can be considered for publication. Adequate revisions may require extensive work and we ordinarily avoid passing a manuscript to a second round of review on this basis. However, we have made changes in our revision policy in response to COVID-19 (https://elifesciences.org/articles/57162). First, because many researchers have temporarily lost access to the labs, we will give authors as much time as they need to submit revised manuscripts. We are also offering, if you choose, to post the manuscript to bioRxiv (if it is not already there) along with this decision letter and a formal designation that the manuscript is "in revision at *eLife*". Please let us know if you would like to pursue this option. (If your work is more suitable for medRxiv, you will need to post the preprint yourself, as the mechanisms for us to do so are still in development.)

Summary:

Bittner and colleagues introduce a machine learning framework for maximum entropy inference of model parameter distributions that are consistent with certain emergent model properties, specified by the investigator. The approach is illustrated on several models of potential interest.

Reviewers were broadly enthusiastic about the potential usefulness of this methodology. However, the reviews and ensuing discussion revealed several points of concern about the manuscript and the approach. The full reviewer comments are included below. The main concerns are summarized as follows.

Essential revisions:

1. The methodology is not adequately explained. Both the body text and methods section present a somewhat selective description that is very hard to follow in places and should be checked and rewritten for clarity, completeness, notational consistency and correctness.

2. The computational resources required to use this method are not adequately benchmarked. For example, the cosubmission (Macke et al) reported wall clock time, required hardware and iterations required to produce results by directly comparing to existing methods (approximate bayesian computation, naive sampling, etc.) Without transparent benchmarks it is not possible to assess the advance offered by this method.

3. The extent to which this method is generally/straightforwardly applicable was in doubt. It seemed as though a significant amount of computation was required to do inference on one specified property and that the computation would need to be run afresh to query a new property. The methodology in the cosubmission (Macke) made clear that computation required for successive inferences is 'amortized' during training on random parameters. Moreover, EPI seemed less flexible than the cosubmission's approach in that it required a differentiable loss function. The complementarity and advantages of this approach as opposed to the cosubmission's approach are therefore unclear.

4. Some examples lack depth in their treatment (see reviewer comments) and in some cases the presentation is somewhat misleading. The STG example is not in fact a model of the STG. The cosubmission (Macke) uses a model close to the original Prinz et al. model, which is a model of the pyloric subnetwork. It would be instructive to benchmark against this same model, including computation time/resources required. Secondly, the subsequent example (input-responsivity in a nonlinear sensory system) appeared to imply that EPI permits 'generation' and testing of hypotheses in a way that other methods do not. All the method really does is estimate a joint distribution of feasible parameters in a specific model which is manually inspected to propose hypotheses. Any other method (including brute force sampling) could be used in a similar way, so any claim that this is an integral advantage of EPI would be spurious. Indeed, one reviewer was confused about the origin of these hypotheses. While it is helpful to illustrate how EPI (and other methods) could be used, the writing needs to be far clearer in general and should clarify that EPI does not offer any new specific means of generating or evaluating hypotheses.

5. There is a substantial literature on parameter sensitivity analysis and inference in systems biology, applied dynamical systems and control that has been neglected in this manuscript. The manuscript needs to acknowledge, draw parallels and explain distinctions from current methods (ABC, profile likelihood, deep learning approaches, gaussian processes, etc). The under-referencing of this literature deepened concerns about whether this approach represented an advance. DOIs for a small subset of potentially relevant papers include:

https://doi.org/10.1038/nprot.2014.025

http://doi.org/10.1085/jgp.201311116

http://doi.org/10.1016/j.tcs.2008.07.005

http://doi.org/10.3182/20120711-3-BE-2027.00381

http://doi.org/10.1093/bioinformatics/btm382

http://doi.org/10.1111/j.1467-9868.2010.00765.x

http://doi.org/10.1098/rsfs.2011.0051

https://doi.org/10.1098/rsfs.2011.0047

http://doi.org/10.1214/16-BA1017

https://doi.org/10.1039/C0MB00107D

6. One of the reviewers expressed concern that the work might have had significant input from a senior colleague during its early stages, and that it might be worth discussing with the senior colleague whether their contribution was worthy of authorship. The authors may take this concern into account in revising the manuscript.

7. Finally, please also address specific points raised by the reviewers, included below.

*Reviewer #1:*

General Assessment: The authors introduce a machine learning framework for maximum entropy inference of model parameters which are consistent with certain emergent properties, which they call 'emergent property inference'. I think this is an interesting and direction, and this looks like a useful step towards this program. I think the paper could be improved with a more thorough discussion both of the broad principles their black box approach seeks to optimize, as well as the details of its implementation. I also think the detailed examples should be more self-contained. Finally I find this work to be somewhat misrepresented as a key to all of theoretical neuroscience. This approach may have some things to offer to the interesting problem of finding parameter regions of models, but this is not the entirety of, nor really a major part of theoretical neuroscience as I see it.

Other concerns:

(1) Maximizing the entropy of the distribution is not a reparameterization invariant exercise. That is, results depend on whether the model parameters contains rates or time constants, for example. I wonder if this approach is attempting to use a 'flat prior' in some sense which has the same reparameterization issue? Can the authors comment?

(2) I don't think this is a criticism of the work, but instead of the writing about it: I find the introductory paragraphs to give a rather limited overview of theory as finding parameters of models which contain the right phenomenology.

(3) I am somewhat familiar with the stomatognathic circuit model, and so that is where I think I understand what they have done best. I don't understand what I should take away from their paper with regards to this model. Are there any findings that hadn't been appreciated before? What does this method tell us about the system and or its model?

(4) I don't follow the other examples. Ideally more details should be given so that readers like myself who don't already know these systems can understand what's been done.

(5) In figure 2C, the difference between the confidence interval between linear and nonlinear predictions is huge! How much of this is due to nonlinearities, and how much is due to differences in the way these models are being evaluated?

*Reviewer #2:*

This is a very interesting approach to an extremely important question in theoretical neuroscience, and the mathematics and algorithms appear to be very rigorous. The complexities in using this in practice make me wonder if it will find wide application though: setting up the objective to be differentiable, tweaking hyperparameters for training, and interpreting the results; all seem to require a lot from the user. On the other hand, the authors are to be congratulated on providing high quality open source code including clear tutorials on how to use it.

1. Training deep networks is hard. Indeed the authors devote a substantial amount of the manuscript to techniques for training them, and note that different hyperparameters were necessary for each of the different studies. Can the authors be confident that they have found the network which gives maximum entropy or close to it? If so, how. If not, how does that affect the conclusions?

2. Interpreting the results still seems to require quite a lot of work. For example, from inspecting Figure 2 the authors extract four hypotheses. Why these four? Are there other hypotheses that could be extracted and if not how do we know there aren't? Could something systematic be said here?

3. Scalability. The authors state that the method should in principle be scalable, but does that apply to interpreting the results? For example, for the V1 model it seems that you need to look at 48 figures for 4 variables, and I believe this would scale as O(n^2^) with n variables. This seems to require an unsustainable amount of manual work?

4. There are some very particular choices made in the applications and I wonder how general the conclusions are as a consequence. For example, in Equation 5 the authors choose an arbitrary amount of variance 0.01^2^ – why? In the same example, why look at y=0.1 and 0.5?

*Reviewer #3:*

This paper addresses a major issue in fitting neuroscience models: how to identify the often degenerate, or nearly degenerate, set of parameters that can underlie a set of experimental observations. Whereas previous techniques often depended upon brute force explorations or special parametric forms or local linearizations to generate sets of parameters consistent with measured properties, the authors take advantage of deep generative networks to address this problem. Overall, I think this paper and the complementary submission have the potential to be transformative contributions to model fitting efforts in neuroscience. That being said, since the primary contribution is the methodology, I think the paper requires more systematic comparisons to ground truth examples to demonstrate potential strengths and weaknesses, and more focus on methodology rather than applications.

1) The authors only have a single ground-truth example where they compare to a known result (a 2x2 linear dynamical system). It would be good to show how well this method compares to results from, for example, a direct brute force grid search of a system with a strongly non-elliptical (e.g. sharply bent) shaped parameter regime and a reasonably large (e.g. 5?) number of parameters corresponding to a particular property, to see how well the derived probability distribution overlaps the brute force grid search parameters (perhaps shown via several 2-D projections).

2) It was not obvious whether EPI actually scales well to higher dimensions and how much computation it would take (there is one claim that it 'should scale reasonably'). While I agree that examples with a small number of parameters is nice for illustration, a major issue is how to develop techniques that can handle large numbers of parameters (brute force, while inelegant, inefficient, and not producing an explicit probability distribution can do a reasonable job for small #'s of parameters). The authors should show some example of extending to larger number of parameters and do some checks to show that it appears to work. As a methodological contribution, the authors should also give some sense of how computationally intensive the method is and some sense of how it scales with size. This seems particularly relevant to, for example, trying to infer uncertainties in a large weight matrix or a non-parametric description of spatial or temporal responses or a sensory neuron (which I'm assuming this technique is not appropriate for? See point #4 below).

3) For the STG-like example, this was done for a very simple model that was motivated by the STG but isn't based on experimental recordings. Most of the brute force models of the STG seek to fit various waveform properties of neurons and relative phases. Could the model handle these types of analyses, or would it run into problems due to either needing to specify too many properties or because properties like "number of spikes per burst" are discrete rather than continuous? This isn't fatal, but would be good to consider and/or note explicitly.

4) The discussion should be expanded to be more specific about what problems the authors think the model is, or is not, appropriate for. Comparisons to the Goncalves article would also be helpful since users will want to know the comparative advantages/disadvantages of each method. (if the authors could coordinate running their methods on a common illustrative example, that would be cool, but not required).

5) Given that the paper is heavily a (very valuable!) methods paper for a general audience, the method should be better explained both in the main text and the supplement. Some specific ones are below, but the authors should more generally send the paper to naïve readers to check what is/is not well explained.

– Figure 1 is somewhat opaque and also has notational issues (e.g. omega is the frequency but also appears to be the random input sample).

– For the general audience of *eLife*, panels C and D are not well described individually or well connected to each other and don't illustrate or describe all of the relevant variables (including what q0 is and what x is).

– In Equation 2 (and also in the same equation in the supplement), it was not immediately obvious what the expectation was taken over.

– The authors don't specific the distribution of w (it's referred to only as 'a simple random variable', which is not clear).

– It was also sometimes hard to quickly find in the text basic, important quantities like what z was for a given simulation.

– The augmented Lagrangian optimization was not well explained or motivated. There is a reference to m=absolute value(mu) but I didn't see m in the above equation.

– Using mu to describe a vector that includes means and variances is confusing notation since mu often denotes means.

– It would be helpful to have a pseudo-code 'Algorithm' figure or section of the text.

---

## [Author Response]

Essential revisions:1. The methodology is not adequately explained. Both the body text and methods section present a somewhat selective description that is very hard to follow in places and should be checked and rewritten for clarity, completeness, notational consistency and correctness.

Agreed. In the introduction, we present our method by focusing on the key aspect of EPI that differentiates it from other approaches to inverse problems. Specifically, we explain how current methodology infers the parameters producing computation by conditioning on *exemplar datasets* (real or simulated), whereas in EPI we condition directly on the *emergent properties* that define the computation. We also note (and have an appendix detailing) how EPI is in fact doing variational bayesian inference, but that the parameterization of the problem as we solve it is more natural given the goal of reproducing a computation. In other words, the methodology is mathematically sound and directly connected to well-worn techniques, but the specific form we solve is the appropriate choice for the motivating problem.

Lines 44-52

“Statistical inference, of course, requires quantification of the sometimes vague term *computation*. […] Related to this point, use of a conventional dataset encourages conventional data likelihoods or loss functions, which focus on some global metric like squared error or marginal evidence, rather than the computation itself.”

We now frame the method within the more general context of parameter inference techniques in neuroscience, rather than the context of recent advancements in machine learning. We have made concentrated efforts to simplify language and to relate to all relevant existing methodology.

Lines 53-63

“Alternatively, researchers often quantify an *emergent property* (EP): a statistic of data that directly quantifies the computation of interest, wherein the dataset is implicit. […] This statistical framework is not new: it is intimately connected to the literature on approximate bayesian computation [24, 25, 26], parameter sensitivity analyses [27, 28, 29, 30], maximum entropy modeling [31, 32, 33], and approximate bayesian inference [34, 35]; we detail these connections in Section 5.1.1.”

To improve clarity, we have overhauled and simplified the notation and presentation of emergent properties. Emergent properties are now denoted with X: rather than a vector of first- and second-moment constraints, we present the emergent property more readably (and equivalently) as mean and variance constraints on emergent property statistics:

Line 742

“X:Ez,x[f(x;z)]=μ,Varz,x[f(x;z)]=σ2 (12)”

Emergent properties and EPI are introduced and explained with this revised notation in Section 'Emergent property inference via deep generative models', and we have completely redone the Methods Section 'Emergent property inference (EPI)' to improve clarity, precisely explain EPI optimization (Section 'Augmented lagrangian optimization'), and derive equivalence to established bayesian inference techniques (Section 'EPI as variational inference').

In our revised writing, we deemphasize the role of maximum entropy in the EPI algorithm, because this has largely served as a distraction in our experience. We show in Section 5.1.6 that EPI is exactly variational inference (this was mentioned in the previous draft but not as a point of connection and focus, as it should be and now is). Therefore, it makes sense to present EPI in the main text as a statistical inference technique that constrains the predictions of the inferred parameters to be an emergent property, and leave the details of the maximum entropy to the technically proficient in Section 5.1.

Lines 152-154

“Many distributions in Q will respect the emergent property constraints, so we select the most random (highest entropy) distribution, which also means this approach is equivalent to bayesian variational inference (see Section 5.1.6).”

Figure 1 has been largely redone to reflect the modified presentation, and improve the pictorial representation of the method. In Section 3.3, we now show that EPI is the only simulation-based inference method that controls the predictions of its inferred distribution (Figure 2C-D).

Finally, we emphasize the utility of this deep inference technique for scientific inquiry in a new paragraph at the end of Section 3.2.

Lines 159-175

“The structure of the inferred parameter distributions of EPI can be analyzed to reveal key information about how the circuit model produces the emergent property.

[…] Importantly and unlike alternative techniques, once an EPI distribution has been learned, the modes and Hessians of the distribution can be measured with trivial computation (see Section 5.1.2).”

2. The computational resources required to use this method are not adequately benchmarked. For example, the cosubmission (Macke et al) reported wall clock time, required hardware and iterations required to produce results by directly comparing to existing methods (approximate bayesian computation, naive sampling, etc.) Without transparent benchmarks it is not possible to assess the advance offered by this method.

We thank the reviewers for emphasizing the importance of this methodological comparison. In Section 3.3, we provide a direct comparison of EPI to alternative simulation-based inference techniques SMC-ABC and SNPE by inferring RNN connectivities that exhibit stable amplification. These comparisons evaluate both wall time (Figure 2A) and simulation count (Figure 2B), and we explain how each algorithm was run in its preferred hardware setting in Section 5.3.4.

In this analysis, we demonstrate the improved scalability of deep inference techniques (EPI and SNPE) with respect to the state-of-the-art approximate bayesian computation technique (SMC-ABC). While controlling for architecture size (Figure 2—figure supplement 1), we push the limits of SNPE through targeted hyperparameter modifications (Figure 2—figure supplement 2), and show that EPI scales to higher dimensional RNN connectivities producing stable amplification than SNPE. Furthermore, we emphasize that SNPE does not constrain the properties of the inferred parameters; many connectivities inferred by SNPE result in unstable or nonamplified models (Figure 2C, Figure 2—figure supplement 3).

3. The extent to which this method is generally/straightforwardly applicable was in doubt. It seemed as though a significant amount of computation was required to do inference on one specified property and that the computation would need to be run afresh to query a new property. The methodology in the cosubmission (Macke) made clear that computation required for successive inferences is 'amortized' during training on random parameters. Moreover, EPI seemed less flexible than the cosubmission's approach in that it required a differentiable loss function. The complementarity and advantages of this approach as opposed to the cosubmission's approach are therefore unclear.

We respectfully disagree with this characterization, but we take responsibility here: we certainly needed to improve our exposition and comparisons to clarify the general applicability of EPI. We have done so, and what results is a meaningful comparison with the cosubmission (SNPE, Macke et al., now published) and other methods. Of course, there is a tradeoff (no free lunch, as usual), and these new analyses clarify when EPI is preferable to SNPE, and vice versa.

First, unlike SNPE, EPI leverages gradients of the emergent property throughout optimization, which leads to better efficiency and scalability (Section 3.3). This is the usual tradeoff of gradient-based vs sampling-based methods. The emergent properties of many models in neuroscience are tractably differentiable, four of which we analyze in this manuscript ranging across levels of biological realism, network size, and computational function. This tradeoff is now explained in the Discussion (Section 4).

Second, of course the reviewers are right to point out EPI does not amortize across emergent properties, and it requires differentiability of the emergent properties of the model. SNPE is more suitable for inference with nondifferentiable mechanistic models and scientific problems requiring many inferred parameter distributions. However, these relative drawbacks of EPI with respect to SNPE can be considered choices made in a tradeoff between simulation-based inference approaches.

On the balance, these two methods occupy different areas of use, and both are equivalently “generally/straightforwardly applicable”; we believe the new analyses and discussion in the manuscript clarify that, as it is a key point for practitioners going forward.

Lines 435-447

“A key difference between EPI and SNPE, is that EPI uses gradients of the emergent property throughout optimization. […] In summary, choice of deep inference technique should consider emergent property complexity and differentiability, dimensionality of parameter space, and the importance of constraining the model behavior predicted by the inferred parameter distribution.”

Furthermore, EPI focuses the entire expressivity of the approximating deep probability distribution on a single distribution, rather than spreading this expressivity to some uncharacterized degree across the chosen training distribution of amortized posteriors in SNPE (Section 5.1.1 Related Approaches).

Lines 832-838

“The approximating fidelity of the deep probability distribution in sequential approaches is optimized to generalize across the training distribution of the conditioning variable. […] The well-known inverse mapping problem of exponential families [85] prohibits an amortization-based approach in EPI, since EPI learns an exponential family distribution parameterized by its mean (in contrast to its natural parameter, see Section 5.1.3).”

Finally, we emphasize that EPI does something fundamental that SNPE and other inference techniques cannot. EPI learns parameter distributions whose predictions are constrained to produce the emergent property. We show in Figure 2 and Supplementary Figure 2—figure supplement 3 that SNPE does not control the statistical properties of its predictions, resulting in the inference of many parameters that are not consistent with the desired emergent property.

Lines 228-238

“No matter the number of neurons, EPI always produces connectivity distributions with mean and variance of real(*λ*_1_) and λ1s according to X (Figure 2C, blue). […] Even for moderate neuron counts, the predictions of the inferred distribution of SNPE are highly dependent on *N* and *g*, while EPI maintains the emergent property across choices of RNN (see Section 5.3.5).”

4. Some examples lack depth in their treatment (see reviewer comments) and in some cases the presentation is somewhat misleading. The STG example is not in fact a model of the STG. The cosubmission (Macke) uses a model close to the original Prinz et al. model, which is a model of the pyloric subnetwork. It would be instructive to benchmark against this same model, including computation time/resources required. Secondly, the subsequent example (input-responsivity in a nonlinear sensory system) appeared to imply that EPI permits 'generation' and testing of hypotheses in a way that other methods do not. All the method really does is estimate a joint distribution of feasible parameters in a specific model which is manually inspected to propose hypotheses. Any other method (including brute force sampling) could be used in a similar way, so any claim that this is an integral advantage of EPI would be spurious. Indeed, one reviewer was confused about the origin of these hypotheses. While it is helpful to illustrate how EPI (and other methods) could be used, the writing needs to be far clearer in general and should clarify that EPI does not offer any new specific means of generating or evaluating hypotheses.

We thank the reviewers for explaining how they found some of the presentation misleading. We have taken serious care in this manuscript to clarify (a) what is novel, appreciable scientific insight provided by EPI, as well as (b) which scientific analyses are made possible by EPI.

(a) In the revised manuscript we clarify that novel theoretical insights are not being made into the STG subcircuit model or the recurrent neural network models. The STG subcircuit serves as a motivational example to explain how EPI works, and we use RNNs exhibiting stable amplification as a substrate for scalability analyses.

Lines 75-79

“First, we show EPI’s ability to handle biologically realistic circuit models using a five-neuron model of the stomatogastric ganglion [41]: a neural circuit whose parametric degeneracy is closely studied [42]. Then, we show EPI’s scalability to high dimensional parameter distributions by inferring connectivities of recurrent neural networks that exhibit stable, yet amplified responses – a hallmark of neural responses throughout the brain [43, 44, 45].”

We do produce strong theoretical insights into a model of primary visual cortex (Section 3.4) and superior colliculus (Section 3.5). These analyses have substantially more depth than the previous manuscript.

Lines 79-87

“In a model of primary visual cortex [46, 47], EPI reveals how the recurrent processing across different neuron-type populations shapes excitatory variability: a finding that we show is analytically intractable. […] Intriguingly, the inferred connectivities of each regime reproduced results from optogenetic inactivation experiments in markedly different ways.”

(b) The ability to infer a flexible approximation to a probability distribution constrained to produce an emergent property is novel in its own right (Figure 2). The deep probability distribution fit by EPI facilitates the mode identification (via gradient ascent of the parameter log probability) and sensitivity measurement (via the measurement of the eigenvector of the Hessian at a parameter value). These mode identifications and sensitivity measurements are done in Sections 3.1 (Figure 1E), 3.4 (Figure 3E), and 3.5 (Figure 4C). By using this mode identification technique along the ridges of high parameter probability in the SC model, we identify the parameters transitioning between the two regimes. Finally, the sensitivity dimensions of the SC model identified by EPI facilitated regime characterization through perturbation analyses (Figure 4D).

Importantly, we do not claim that these theoretical insights were necessarily dependent on using the techniques in (b). One could have come to these conclusions via various combinations of techniques mentioned in Section 5.1.1 Related Methods. In the case of the V1 model inference, the main point is to indicate that such insight can be afforded by EPI and its related methods, in contrast to the analytic derivations emblematic of practice in theoretical neuroscience.

Lines 297-305

“EPI revealed the quadratic dependence of excitatory variability on input variability to the E- and P-populations, as well as its independence to input from the other two inhibitory populations. […] By pointing out this mathematical complexity, we emphasize the value of EPI for gaining understanding about theoretical models when mathematical analysis becomes onerous or impractical.”

In the case of the SC model inference, random sampling would have taken prohibitively long, and it is unclear how the continuum between the two connectivity regimes would have been identified with alternative techniques:

Lines 412-415

“The probabilistic tools afforded by EPI yielded this insight: we identified two regimes and the continuum of connectivities between them by taking gradients of parameter probabilities in the EPI distribution, we identified sensitivity dimensions by measuring the Hessian of the EPI distribution, and we obtained many parameter samples at each step along the continuum at an efficient rate.”

As the reviewers indicate, the STG model analyzed in our manuscript is not that of Prinz et al. 2004, and thus not the model analyzed by the cosubmission (Macke et al). We chose an alternative model, believing it important to demonstrate inference in biophysically realistic Morris-Lecar models when gradients are tractable, which is the case of the 5-neuron STG model we analyzed from Gutierrez et al. 2013. This 5-neuron model represents the IC neuron (hub) and its coupling to the pyloric (fast) or gastric mill (slow) subcircuit rhythms. In the introductory text, we refer to this model as the “STG subcircuit” model (rather than “STG model”), and we better clarify what aspect of the STG is being modeled in Results section 3.1.

Lines 98-103

“A subcircuit model of the STG [41] is shown schematically in Figure 1A. The fast population (f1 and f2) represents the subnetwork generating the pyloric rhythm and the slow population (s1 and s2) represents the subnetwork of the gastric mill rhythm. […] The hub neuron couples with either the fast or slow population, or both depending on modulatory conditions.”

The difference between this model and the STG model of the pyloric subnetwork is emphasized in Discussion:

Lines 440-443

“However, conditioning on the pyloric rhythm [68] in a model of the pyloric subnetwork model [15] proved to be prohibitive with EPI. The pyloric subnetwork requires many time steps for simulation and many key emergent property statistics (e.g. burst duration and phase gap) are not calculable or easily approximated with differentiable functions.”

5. There is a substantial literature on parameter sensitivity analysis and inference in systems biology, applied dynamical systems and control that has been neglected in this manuscript. The manuscript needs to acknowledge, draw parallels and explain distinctions from current methods (ABC, profile likelihood, deep learning approaches, gaussian processes, etc). The under-referencing of this literature deepened concerns about whether this approach represented an advance. DOIs for a small subset of potentially relevant papers include:https://doi.org/10.1038/nprot.2014.025http://doi.org/10.1085/jgp.201311116http://doi.org/10.1016/j.tcs.2008.07.005http://doi.org/10.3182/20120711-3-BE-2027.00381http://doi.org/10.1093/bioinformatics/btm382http://doi.org/10.1111/j.1467-9868.2010.00765.xhttp://doi.org/10.1098/rsfs.2011.0051https://doi.org/10.1098/rsfs.2011.0047http://doi.org/10.1214/16-BA1017https://doi.org/10.1039/C0MB00107D

Thank you for pointing us to these references on sensitivity analyses, MCMC inference, and applied dynamical systems. We have incorporated most of them into the current manuscript and explain EPI’s relation to each class of these techniques throughout Section 5.1.1 Related approaches.

6. One of the reviewers expressed concern that the work might have had significant input from a senior colleague during its early stages, and that it might be worth discussing with the senior colleague whether their contribution was worthy of authorship. The authors may take this concern into account in revising the manuscript.

Thank you for pointing this out. We take authorship and scientific contribution very seriously (it is always part of our manuscript submission process, as it was here). We have gone back and discussed every researcher who was in any way involved in this work, and to our best estimation, we think this comment references Woods Hole Course Project mentors where this work was discussed and explored in its early stages. We have had explicit follow-up conversations with James Fitzgerald and Dhruva Raman and noted that of course we would be happy to have them involved at any level (great colleagues!). They both reported they are happy with their acknowledgement in the paper and don’t feel that there is a justification for authorship (again we were very welcoming of this possibility and maintain positive enthusiasm for both). Furthermore, Stephen Baccus has requested that we acknowledge the summer course, which we have done. If there is any other “senior colleague” the reviewer had in mind, please let us know and we will of course gladly pursue the matter.

Reviewer #1:General Assessment: The authors introduce a machine learning framework for maximum entropy inference of model parameters which are consistent with certain emergent properties, which they call 'emergent property inference'. I think this is an interesting and direction, and this looks like a useful step towards this program. I think the paper could be improved with a more thorough discussion both of the broad principles their black box approach seeks to optimize, as well as the details of its implementation. I also think the detailed examples should be more self-contained. Finally I find this work to be somewhat misrepresented as a key to all of theoretical neuroscience. This approach may have some things to offer to the interesting problem of finding parameter regions of models, but this is not the entirety of, nor really a major part of theoretical neuroscience as I see it.

We thank the reviewer for their positive comments and thoughtful feedback. We have made serious effort to improve our presentation and explanation of EPI (see response to main concern #1). Furthermore, we have focused on clearly motivating and describing each neural circuit model studied in this manuscript. Sufficient mathematical detail is written in each Results section, while full details are presented in Methods. All code for training EPI on these models and their analysis are available in well-documented scripts and notebooks in our github repository.

https://github.com/cunningham-lab/epi

We modify our writing to clarify that EPI is not a key to all of theoretical neuroscience, but rather a powerful solution to inverse problems in neural circuit modeling. Inverse problems are indeed a major part of theoretical neuroscience, and their solutions are important for the evaluation of neural circuit models. We clarify this point in the introduction:

Lines 26-35

“The fundamental practice of theoretical neuroscience is to use a mathematical model to understand neural computation, whether that computation enables perception, action, or some intermediate processing. […] The inverse problem is crucial for reasoning about likely parameter values, uniquenesses and degeneracies, and predictions made by the model [6, 7, 8].”

Other concerns:(1) Maximizing the entropy of the distribution is not a reparameterization invariant exercise. That is, results depend on whether the model parameters contains rates or time constants, for example. I wonder if this approach is attempting to use a 'flat prior' in some sense which has the same reparameterization issue? Can the authors comment?

The reviewer is correct to point out that maximum entropy solutions are not reparameterization invariant, and indeed the units matter. The reviewer’s suggestion that the method is in some sense using a flat prior is also correct. To clarify, EPI does not execute posterior inference, because there is no explicit dataset or specified prior belief in the EPI framework. However, we derive the relation of EPI to bayesian variational inference in Section 5.1.6, which shows EPI uses a uniform prior when framed as variational inference.

Lines 1027-1031

“We see that EPI is exactly variational inference with an exponential family likelihood defined by sufficient statistics *T*(z^) =^ Ex_∼*p*(x|z)_ [*φ*(x;z)], and where the natural parameter *η*^∗^ is predicated by the mean parameter *µ*_opt_. Equation 40 implies that EPI uses an improper (or uniform) prior, which is easily changed.”

In our examples, we only infer distributions of parameters with the same units, so this issue should not draw concern over the validity of our model analyses. As suggested, the EPI solution will differ according to relative scaling of parameter values under the maximum entropy selection principle. Thus, an important clarification is that sensitivity quantifications are made in the context of the chosen parameter scalings.

(2) I don't think this is a criticism of the work, but instead of the writing about it: I find the introductory paragraphs to give a rather limited overview of theory as finding parameters of models which contain the right phenomenology.

We appreciate this feedback. We have adapted the introduction to clarify that we are focusing on solving inverse problems in theoretical neuroscience.

(3) I am somewhat familiar with the stomatognathic circuit model, and so that is where I think I understand what they have done best. I don't understand what I should take away from their paper with regards to this model. Are there any findings that hadn't been appreciated before? What does this method tell us about the system and or its model?

We clarify in point 4 above that we are not claiming to produce novel, appreciable scientific insight about the STG subcircuit model, which is used as a motivation example. The takeaway is that the conductance parameters producing intermediate hub frequency belong to a complex 2-D distribution, which EPI captures accurately, and that EPI can tell us the parameter changes away from the prototypical configuration that change hub frequency the most or least. For example, for increases in *g*_el_ and *g*_synA_ according to the proportions of the degenerate dimension of parameter space, intermediate hub neuron frequency will be preserved in this model. This suggests that in the real STG neural circuit, the IC neuron will remain at an intermediate frequency between the pyloric and gastric mill rhythms if parameter changes are made along such a dimension.

Lines 171 -173

“The directionality of v_2_ suggests that changes in conductance along this parameter combination will most preserve hub neuron firing between the intrinsic rates of the pyloric and gastric mill rhythms.”

(4) I don't follow the other examples. Ideally more details should be given so that readers like myself who don't already know these systems can understand what's been done.

Thank you for the feedback. We have taken care to give more general context and motivation for each neural circuit model.

(5) In figure 2C, the difference between the confidence interval between linear and nonlinear predictions is huge! How much of this is due to nonlinearities, and how much is due to differences in the way these models are being evaluated?

In the current manuscript, we do not examine the difference between linear and nonlinear predictions of the V1 model.

Reviewer #2:This is a very interesting approach to an extremely important question in theoretical neuroscience, and the mathematics and algorithms appear to be very rigorous. The complexities in using this in practice make me wonder if it will find wide application though: setting up the objective to be differentiable, tweaking hyperparameters for training, and interpreting the results; all seem to require a lot from the user. On the other hand, the authors are to be congratulated on providing high quality open source code including clear tutorials on how to use it.1. Training deep networks is hard. Indeed the authors devote a substantial amount of the manuscript to techniques for training them, and note that different hyperparameters were necessary for each of the different studies. Can the authors be confident that they have found the network which gives maximum entropy or close to it? If so, how. If not, how does that affect the conclusions?

We agree with the reviewer that hyperparameter sensitivity and global optima are important considerations of any optimization algorithm using deep neural networks. To draw a parallel, training deep networks for visual processing used to be considered infeasible, but became easier through iterative improvements in architectural and hyperparameter choices that spread across the field. Similarly, training deep networks via EPI to learn parameter distributions became easier throughout this research project as we learned through trial and error what works well. In fact, there has been extraordinary progress in the field of deep probability distributions (specifically normalizing flows), that have allowed EPI to converge regularly while capturing complex structure (e.g. Dinh et al. 2017 and Kingma et al. 2018). This manuscript’s explanation of hyperparameter choices and the extensive set of examples in the online code will serve as valuable guidelines for future research using this method. Every figure of this paper is reproducible with the jupyter notebooks, and there are several tutorials for understanding the most consequential hyperparameters: the augmented lagrangian constant and deep probability distribution architecture.

In general, we cannot know if we have obtained the global maximum entropy distribution for a given emergent property. The reviewer is correct to point out that multiple distributions may satisfy the emergent property and have different levels of entropy. In the new manuscript, we present the method as an inference technique without focusing very greatly on maximum entropy, since it tends to distract and confuse the reader. We derive an analytic equivalence to variational inference (Section 5.1.6) showing that (a) EPI is a valid inference method, and (b) to emphasize that maximum entropy is the normative selection principle of bayesian inference methods in general. Thus, the concern of not having the globally optimal inferred distribution is the same that applies to all other statistical inference techniques.

Practically, this has scientific implications. It means that we may be missing important structure in the inferred distribution, or we may be missing additional modes in parameter space. To handle this methodologically, we run EPI with multiple random seeds, and select the distribution that has converged with the greatest entropy for scientific analysis. Throughout the manuscript, we compare to analytic, error contour, and brute-force ground truth to ensure we are capturing the correct distribution with EPI (see response to R3 concern 1).

2. Interpreting the results still seems to require quite a lot of work. For example, from inspecting Figure 2 the authors extract four hypotheses. Why these four? Are there other hypotheses that could be extracted and if not how do we know there aren't? Could something systematic be said here?

This analysis is no longer in the manuscript.

3. Scalability. The authors state that the method should in principle be scalable, but does that apply to interpreting the results? For example, for the V1 model it seems that you need to look at 48 figures for 4 variables, and I believe this would scale as O(n^2^) with n variables. This seems to require an unsustainable amount of manual work?

We refer the reviewer to Figure 2 and Section 3.3 for scalability analysis. The scaling analysis addresses the question of the issue of parameter discovery with EPI in high-dimensional parameter spaces.

Another important question the reviewer brings up is how well one can analyze the high-dimensional parameter distributions that EPI produces? Indeed, these distributions become more challenging to understand and visualize in high dimensions. This is where the sensitivity measurements appearing in sections 3.1, 3.4, and 3.5 can be particularly useful. Even in high dimensions, trained deep probability distributions offer tractable quantitative assessments of how parametric combinations affect the emergent property that was conditioned upon.

4. There are some very particular choices made in the applications and I wonder how general the conclusions are as a consequence. For example, in Equation 5 the authors choose an arbitrary amount of variance 0.01^2^ – why? In the same example, why look at y=0.1 and 0.5?

In the current manuscript, we make sure to explain all choices of the emergent property constraints Here, we show the description of each emergent property with equations omitted.

Section 3.2, Lines 136-140

“We stipulate that the hub neuron’s spiking frequency – denoted by statistic *ω*_hub_(**x**)

– is close to a frequency of 0.55Hz, between that of the slow and fast frequencies. Mathematically, we define this emergent property with two constraints: that the mean hub frequency is 0.55Hz, and that the variance of the hub frequency is moderate.”

Section 3.3, Lines 200-206

“Two conditions are necessary and sufficient for RNNs to exhibit stable amplification [51]: real(*λ*_1_) *<* 1 and λ1s>1 .[…] EPI can naturally condition on this emergent property Variance constraints predicate that the majority of the distribution (within two standard deviations) are within the specified ranges.”

Section 3.4, Lines 275-278

“We quantify levels of E-population variability by studying two emergent properties where *s_E_*(x;z) is the standard deviation of the stochastic *E*-population response about its steady state (Figure 3C). In the following analyses, we select 1Hz^2^ variance such that the two emergent properties do not overlap in *s_E_*(z;x).”

Section 3.5, Lines 331-334

“We stipulate that accuracy be on average.75 in each task with variance.075^2^. 75% accuracy is a realistic level of performance in each task, and with the chosen variance, inferred models will not exhibit fully random responses (50%), nor perfect performance (100%).”

Reviewer #3:This paper addresses a major issue in fitting neuroscience models: how to identify the often degenerate, or nearly degenerate, set of parameters that can underlie a set of experimental observations. Whereas previous techniques often depended upon brute force explorations or special parametric forms or local linearizations to generate sets of parameters consistent with measured properties, the authors take advantage of deep generative networks to address this problem. Overall, I think this paper and the complementary submission have the potential to be transformative contributions to model fitting efforts in neuroscience. That being said, since the primary contribution is the methodology, I think the paper requires more systematic comparisons to ground truth examples to demonstrate potential strengths and weaknesses, and more focus on methodology rather than applications.1) The authors only have a single ground-truth example where they compare to a known result (a 2x2 linear dynamical system). It would be good to show how well this method compares to results from, for example, a direct brute force grid search of a system with a strongly non-elliptical (e.g. sharply bent) shaped parameter regime and a reasonably large (e.g. 5?) number of parameters corresponding to a particular property, to see how well the derived probability distribution overlaps the brute force grid search parameters (perhaps shown via several 2-D projections).

We thank the reviewer for pointing out the importance of ground truth comparisons in this manuscript. In this revision, we make ground truth comparisons via analytic derivations, empirical error contours, and brute-force sampling.

Analytic comparisons: The 2x2 linear dynamical system is chosen as a worked example because it has multi-modal non-elliptical structure (Figure 1—figure supplement 1), and its contours can be derived analytically (Figure 1—figure supplement 2). Similarly, in Section 5.4.5, we derive the quadratic relationship between excitatory variability and input noise variability (in a simplified model) suggesting that the quadratic relationship uncovered by EPI (see Section 3.4) is correct.

Error contours: In the motivation example, we compare the EPI inferred distribution of STG conductances to hub frequency contours (Figure 1E), which show that the non-elliptical parametric structure captured by EPI is in agreement with these contours. This general region of parameter space was labeled following grid search analyses in a previous study (Gutierrez et al. 2013, Figure 2, parameter regime G).

Brute-force: The EPI inferred distribution for rapid task switching in the SC model is sharply bent (Figure 4), and matches the parameter set returned from random sampling (Figure 4—figure supplement 5A). We note that the brute-force parameter set is actually not the ground-truth solution, because it does not obey the constraints of the emergent property as the EPI distribution does (Figure 4—figure supplement 5B). This can explain the spurious samples in the brute-force set that are not in the EPI inferred distribution.

All EPI distributions shown in this manuscript are “validated” in the sense that they pass a hypothesis testing criteria for emergent property convergence; all EPI distributions produce their emergent properties. Finally, the underlying maximum entropy flow network (MEFN) algorithm is compared to a ground truth solution (Loaiza Ganem et al. 2017, Figure 2) by deriving ground truth from the duality of maximum entropy distributions and exponential families (see Section 5.1.3).

2) It was not obvious whether EPI actually scales well to higher dimensions and how much computation it would take (there is one claim that it 'should scale reasonably'). While I agree that examples with a small number of parameters is nice for illustration, a major issue is how to develop techniques that can handle large numbers of parameters (brute force, while inelegant, inefficient, and not producing an explicit probability distribution can do a reasonable job for small #'s of parameters). The authors should show some example of extending to larger number of parameters and do some checks to show that it appears to work. As a methodological contribution, the authors should also give some sense of how computationally intensive the method is and some sense of how it scales with size. This seems particularly relevant to, for example, trying to infer uncertainties in a large weight matrix or a non-parametric description of spatial or temporal responses or a sensory neuron (which I'm assuming this technique is not appropriate for? See point #4 below).

The reviewer is right to point out the importance of a scaling analysis. Please see response to Main concern #2.

3) For the STG-like example, this was done for a very simple model that was motivated by the STG but isn't based on experimental recordings. Most of the brute force models of the STG seek to fit various waveform properties of neurons and relative phases. Could the model handle these types of analyses, or would it run into problems due to either needing to specify too many properties or because properties like "number of spikes per burst" are discrete rather than continuous? This isn't fatal, but would be good to consider and/or note explicitly.

The STG subcircuit model of Gutierrez et al. 2013 is certainly less complex than other models of the STG, yet 5 connected Morris-Lecar neurons is certainly a nontrivial system. We clarify why this model is analyzed in Section 3.1 instead of more complex STG models when discussing the differences between EPI and SNPE in Discussion:

Lines 435-447

“A key difference between EPI and SNPE, is that EPI uses gradients of the emergent property throughout optimization. […] In summary, choice of deep inference technique should consider emergent property complexity and differentiability, dimensionality of parameter space, and the importance of constraining the model behavior predicted by the inferred parameter distribution.”

4) The discussion should be expanded to be more specific about what problems the authors think the model is, or is not, appropriate for. Comparisons to the Goncalves article would also be helpful since users will want to know the comparative advantages/disadvantages of each method. (if the authors could coordinate running their methods on a common illustrative example, that would be cool, but not required).

Thank you for this recommendation. We now include substantial text in discussion devoted to this topic in addition to that quoted in the previous response (R3 concern #3).

Lines 427-447

“Methodology for statistical inference in circuit models has evolved considerably in recent years. […] In summary, choice of deep inference technique should consider emergent property complexity and differentiability, dimensionality of parameter space, and the importance of constraining the model behavior predicted by the inferred parameter distribution.”

5) Given that the paper is heavily a (very valuable!) methods paper for a general audience, the method should be better explained both in the main text and the supplement. Some specific ones are below, but the authors should more generally send the paper to naïve readers to check what is/is not well explained.– Figure 1 is somewhat opaque and also has notational issues (e.g. omega is the frequency but also appears to be the random input sample).

Figure 1 has been completely revised.

– For the general audience of eLife, panels C and D are not well described individually or well connected to each other and don't illustrate or describe all of the relevant variables (including what q0 is and what x is).

Agreed. In the new figure, we clearly depict **z** as parameters and **x** as circuit activity. We keep the vertical directionality of panel D exclusively for the deep generative process of the deep probability distribution (where *q*_0_ is replaced and shown explicitly as an isotropic gaussian input to the deep network). The horizontal directionality shared between panels D and E reflect the procedure of theoretical neuroscience described in Section 3.1.

– In Equation 2 (and also in the same equation in the supplement), it was not immediately obvious what the expectation was taken over.

We take care to always indicate the variables over which expectations are taken in the updated manuscript.

– The authors don't specific the distribution of w (it's referred to only as 'a simple random variable', which is not clear).

It is clarified in the text that this simple initial distribution transformed by the deep neural network is an isotropic gaussian.

– It was also sometimes hard to quickly find in the text basic, important quantities like what z was for a given simulation.

We have made sure to make this explicit in each section of the paper.

– The augmented Lagrangian optimization was not well explained or motivated. There is a reference to m=absolute value(mu) but I didn't see m in the above equation.

Lines 767-769

“To run this constrained optimization, we use an augmented lagrangian objective, which is the standard approach for constrained optimization [70], and the approach taken to fit Maximum Entropy Flow Networks (MEFNs) [38]."

– Using mu to describe a vector that includes means and variances is confusing notation since mu often denotes means.

Agreed. Throughout the updated main text, we only describe emergent properties through explicit mean and variance constraints as in Equation 11. We reserve the mean parameterization *µ*_opt_ of the maximum entropy solution of the EPI optimization for technical details in methods. The difference between *µ* and *µ*_opt_ is described in Section 5.1.3.

– It would be helpful to have a pseudo-code 'Algorithm' figure or section of the text.

We provide pseudocode for the EPI optimization in Algorithm 1, which mirrors the pseudocode found in the paper describing the underlying algorithm for MEFN [38].